# Laplace Transform Based Low-Complexity Learning
# of Continuous Markov Semigroups

**Vladimir R. Kostic** [* 1 2]  **Karim Lounici** [* 3]  **Hélène Halconruy** [* 4 5]
**Timothée Devergne** [1]  **Pietro Novelli** [1]  **Massimiliano Pontil** [1 6]

## Abstract

Markov processes serve as universal models for many real-world random processes. This paper presents a data-driven approach to learning these models through the spectral decomposition of the infinitesimal generator (IG) of the Markov semigroup. Its unbounded nature complicates traditional methods such as vector-valued regression and Hilbert-Schmidt operator analysis. Existing techniques, including physics-informed kernel regression, are computationally expensive and limited in scope, with no recovery guarantees for transfer operator methods when the time-lag is small. We propose a novel method leveraging the IG's resolvent, characterized by the Laplace transform of transfer operators. This approach is robust to time-lag variations, ensuring accurate eigenvalue learning even for small time-lags. Our statistical analysis applies to a broader class of Markov processes than current methods while reducing computational complexity from quadratic to linear in the state dimension. Finally, we demonstrate our theoretical findings in several experiments.

## 1. Introduction

Markov semigroups play a critical role in modeling dynamics of complex systems across various fields, including option pricing in finance (Karatzas & Shreve, 1991), molecular dynamics (Schütte & Huisinga, 2003) and climate modeling (Majda et al., 2009), where understanding long-term behavior is essential for accurate forecasting and interpretation.

The central mathematical object for describing Markov semigroups is the Infinitesimal Generator (IG), which governs the evolution of probability distributions over the state space. Its spectra reveal important features such as metastable states, transition statistics, and committor functions, all of which are critical for understanding system dynamics. Accurately learning the spectral decomposition of the IG is thus pivotal for a wide range of applications, including molecular dynamics, time-series clustering, computational neuroscience, and beyond.

The field of molecular dynamics has particularly benefited from spectral decomposition methods of Markov semigroups. Research on AI-augmented molecular dynamics, grounded in statistical mechanics, highlights the importance of accurately identifying spectral gaps (the separation between slow and fast modes of dynamics) in molecular simulations, see (Schütte et al., 2001). Recently, theoretical advancements in (Kostic et al., 2024a) were used in (Devergne et al., 2024) to demonstrate the effectiveness of IG-based methods in accelerating simulations and enabling the practical identification of metastable states. The authors emphasize that IG methods overcome the limitations of more widely used transfer operator (TO) approaches when extracting dynamical information from biased data, and they underline the scalability to larger proteins as a particularly important advantage of IG methods.

On the other hand, Klus & Djurdjevac Conrad (2023) introduced a Transfer Operator (TO)-based spectral clustering method tailored for directed and time-evolving graphs. By leveraging TOs, their approach demonstrates how to identify coherent sets within complex networks, enhancing the analysis of temporal data structures. Furthermore, Cabannes & Bach (2024) emphasize that IG methods open exciting new directions for spectral-based algorithms, be it spectral clustering, spectral embeddings, or spectral distances.

In neuroscience, Marrouch et al. (2020) used TO (also known as the Koopman operator) to analyze brain activity. Their work shows how the operator's spectrum captures the spatiotemporal dynamics of neural signals, providing insights into brain function and possible applications to diagnosing neurological disorders. Ostrow et al. (2023) further

---

*Equal contribution  [1]CSML, Istituto Italiano di Tecnologia, Genova, Italy  [2]Faculty of Science, University of Novi Sad, Serbia  [3]CMAP, École Polytechnique, Paris, France  [4]SAMOVAR, Télécom Sud-Paris, Institut Polytechnique de Paris, France  [5]MODAL'X, Université Paris Nanterre, France  [6]AI Center, University College London, UK. Correspondence to: Vladimir R. Kostic <vladimir.kostic@iit.it>.

*Proceedings of the 42nd International Conference on Machine Learning*, Vancouver, Canada. PMLR 267, 2025. Copyright 2025 by the author(s).

developed a dynamical similarity analysis based on TOs to distinguish learning rules in an unsupervised way, showing that the TO spectrum supports comparative analysis of the temporal structure of computation in neural circuits.

However, the typically unbounded nature of the IG makes designing efficient and reliable estimators challenging. In this paper we address this problem through the lens of Laplace transform. As we shall see, this fully data-driven approach provides a way to bypass both the difficulties of estimating an unbounded operator and overcomes limitations of the transfer operator based algorithms when trajectory data is sampled at high frequency. In particular, we show theoretically and empirically that our method enables accurate and robust estimation of eigenvalues and eigenfunctions even for arbitrarily small time-lags.

**Related work.** A substantial body of research has focused on using transfer operators to learn dynamical systems from data (see the monographs by Brunton et al., 2022; Kutz et al., 2016, and references therein). This has led to the development of two primary approaches: deep learning methods (Bevanda et al., 2021; Fan et al., 2021; Lusch et al., 2018), which excel in capturing complex data representations but often lack rigorous statistical foundations, and kernel methods (Das & Giannakis, 2020; Klus et al., 2019; Kostic et al., 2022; 2023; Williams et al., 2015), which offer strong statistical guarantees for Transfer Operator (TO) estimation but require kernel function selection. A closely related challenge, learning invariant subspaces of the TO, has led to several methodologies (see e.g. Li et al., 2017; Mardt et al., 2018; Tian & Wu, 2021, and references therein), some of which leverage deep canonical correlation analysis (Andrew et al., 2013; Kostic et al., 2024c). Note that TOs share the same eigenfunctions as the IG, which motivates the development of TO methods aimed at learning the spectral properties of the IG. Yet TOs are sensitive to time-lag choice, with their spectral gap deteriorating as the time-lag decreases, making existing spectral recovery guarantees ineffective for small lags—an issue observed in practice, see, e.g., (Bonati et al., 2021). This bottleneck is especially problematic in complex tasks like enhanced sampling (Laio & Parrinello, 2002; Shmilovich & Ferguson, 2023). To address this, research has focused on learning the IG and its eigenstructure directly. As recently shown (Devergne et al., 2024), IG learning can be combined with enhanced sampling methods to efficiently debias data and reveal true dynamics. Still, research on IG learning remains limited and often ad hoc. For instance, (Zhang et al., 2022) developed a deep learning method for Langevin diffusion, while (Klus et al., 2020) extended dynamic mode decomposition to learn the generator, connecting it to Galerkin's approximation. However, neither of these works provides any formal learning guarantees. To the best of our knowledge, most existing works with theoretical guarantees (Cabannes & Bach, 2024; Pillaud-Vivien & Bach, 2023; Hou et al., 2023) either have limited scope or offer only partial or suboptimal analysis, as summarized in Table 1. Crucially, none adequately addresses the challenge posed by the unboundedness of the IG, leading to incomplete frameworks and suboptimal convergence rates, which in some cases depend explicitly on the state space dimension. Moreover, the estimators in these works are susceptible to spurious eigenvalues and do not offer guarantees for accurate estimation of eigenvalues and eigenfunctions. The current state-of-the-art (Kostic et al., 2024a) introduces a physics-informed kernel regression method only for Markov processes admitting a Dirichlet form. This approach leverages the Dirichlet form to define an energy-based metric for learning the model and provides a comprehensive statistical analysis with learning guarantees for the spectral decomposition of the IG while avoiding spurious eigenvalues. That said, their analysis applies only to self-adjoint IGs, assumes partial knowledge and i.i.d. data, and requires gradients of the feature map, leading to a quadratic scaling with the state space dimension $d$, thereby hindering broader applicability.

**Contributions.** We introduce a novel approach for accurately estimating the spectral decomposition of the IG for a broad class of Markov semigroups, encompassing all models considered in prior work. Our method leverages a useful connection between the resolvent of the IG and the semigroup of TOs via the Laplace transform. Unlike TO methods, it estimates the IG directly, avoiding small time-lag issues and preserving a larger spectral gap for more accurate eigenvalue and eigenfunction learning. We provide sharp statistical guarantees, valid for data sampled from a trajectory in the stationary regime, accounting for slow mixing effects. Our results are the first to apply to a wide class of Markov semigroups with sectorial IGs. A key technical contribution is our bound on the Bochner integral approximation error using Crouzeix's bound, which may be of broader interest. Computationally, our method combines multiple TOs at different time-lags through a single matrix product between a Toeplitz matrix and the kernel embedding, reducing complexity to $O(n^2 d)$, making it practical for high-dimensional systems. The complexity can be further reduced while preserving accuracy by utilizing standard scaling techniques such as random Fourier features. Our experiments show a striking performance improvement over TO-based and other IG methods, as predicted by our theory.

## 2. Background

Various physical, biological, and financial systems evolve through stochastic processes $X = (X_t)_{t \in \mathbb{R}^+}$, where $X_t \in \mathcal{X} \subset \mathbb{R}^d$ represents the system's state at time $t$. We focus on continuous-time *Markov processes* with continuous paths, which are essential for modeling these systems. This class includes Itô diffusion processes (see Ex. 2.1 and 2.2), re-

flected or time-changed Brownian motions, and processes with local time. Markov processes model phenomena where the future depends only on the present, not the past, and are described by their laws—measures on the path space. This foundational approach to *Markov theory* defines the process through the *infinitesimal generator* (IG), a key linear (often unbounded) operator on a space of observables (functions defined on the state space).

**Markov theory.** The dynamics of a continuous-time Markov process $X$ is described by a family of probability densities $(p_t)_{t \in \mathbb{R}_+}$

$$\mathbb{P}(X_t \in E | X_0 = x) = \int_E p_t(x, y) dy, \qquad (1)$$

and *transfer operators* (TO) $(A_t)_{t \in \mathbb{R}_+}$ such that for all $t \in \mathbb{R}_+$, $E \in \mathcal{B}(\mathcal{X})$, $x \in \mathcal{X}$ and measurable function $f : \mathcal{X} \to \mathbb{R}$,

$$A_t f = \int_{\mathcal{X}} f(y) p_t(\cdot, y) dy = \mathbb{E}\big[ f(X_t) \, | \, X_0 = \cdot \big]. \quad (2)$$

The transfer operator is essential for understanding the dynamics of $X$. We study its action on $\mathcal{L}^2_\pi(\mathcal{X})$, the space of functions on that are square-integrable with respect to an *invariant measure* $\pi$, which satisfies $A_t^* \pi = \pi$ for all $t \in \mathbb{R}_+$. We assume that the Markov process $X$ meets two key properties regarding $\pi$: **[1]** $\pi$ ensures *long-term stability*, meaning $X$ converges to $\pi$ from any initial state $x$. **[2]** The process exhibits *geometric ergodicity*, meaning it converges exponentially fast to the invariant measure. Finally, the process is characterized by the *infinitesimal generator* $L$, defined for $f \in \mathcal{L}^2_\pi(\mathcal{X})$ by the limit $Lf = \lim_{t \to 0^+} (A_t f - f)/t$, with $L$ being closed on its domain.

The class of **sectorial generators** consists of the operators generating strongly continuous semigroups, analytic in a sector of the complex plane defined by growth conditions in an angular region, i.e., $L$ is a (stable) sectorial operator with angle $\theta \in [0, \pi/2)$,

$$\begin{aligned} \mathrm{F}(L) \subseteq \mathbb{C}_\theta^- := \{ z \in \mathbb{C} \mid \Re(z) \le 0 \; \wedge \\ |\Im(z)| \le -\Re(z) \tan(\theta) \}, \quad (3) \end{aligned}$$

where $\mathrm{F}(L)$ denotes the *numerical range* of $L$. This class covers all time-reversal processes (self-adjoint IG), but also important non-time-reversal processes, such as Advection-Diffusion and underdamped Langevin (Kloeden et al., 1992).

**Spectral decomposition.** When continuous for some $\mu \in \mathbb{C}$, the operator $R_\mu = (\mu I - L)^{-1}$ is the *resolvent* of $L$, and $\rho(L) = \{ \mu \in \mathbb{C} \mid \mu I - L$ is bijective, $R_\mu$ is continuous $\}$ is called the *resolvent set*. For a sectorial operator, the resolvent is uniformly bounded outside a sector containing the spectrum. The spectral decomposition of the IG can be written as

$$L = \sum_{i \in \mathbb{N}} \lambda_i \, g_i \otimes f_i \qquad (4)$$

with eigenvalues $(\lambda_i)_{i \in \mathbb{N}} \subset \mathbb{C}$ and corresponding left and right eigenfunctions $f_i, g_i \in L^2$, respectively.

**Resolvent operator.** Eigenvalues, while informative about long-term behavior, fail to capture transient dynamics of the full time evolution of the process whenever IG is non-normal, that is when $LL^* \ne L^*L$, (Trefethen & Embree, 2020). In contrast, the resolvent of $L$ defined by $R_\mu := (\mu I - L)^{-1}$, $\mu \in \rho(L)$ being a shift parameter $\mu \in \rho(L)$, provides a more comprehensive view of the dynamics, making it the core object of spectral theory of IG. Through its characterization via the Laplace transform, (see for instance (Bakry et al., 2014), equation (A.1.3)) as

$$R_\mu = \int_0^\infty A_t e^{-\mu t} dt, \qquad (5)$$

it is intrinsically connected to the TO defined for the time-lag $t$ by $A_t = e^{tL}$. Moreover, the resolvent encodes both the spectrum of $L$ (eigenvalues via its poles and the continuous spectrum) and transient phenomena, such as the system's approach to equilibrium. Last, the resolvent is essential for analyzing stability under perturbations and understanding how changes in the IG affect the dynamics.

**Link with SDEs.** *Itô diffusion processes* are a key example of Markov processes, governed by stochastic differential equations (SDEs) of the form:

$$dX_t = a(X_t) dt + b(X_t) dW_t, \quad X_0 = x, \qquad (6)$$

where $x \in \mathcal{X}$, $W = (W_t^1, \ldots, W_t^p)_{t \in \mathbb{R}^+}$ is a standard $p$-dimensional Brownian motion, the drift $a : \mathcal{X} \to \mathbb{R}^d$ and diffusion $b : \mathcal{X} \to \mathbb{R}^{d \times p}$ are globally Lipschitz and sub-linear. This ensures a unique solution $X = (X_t)_{t \ge 0}$ in $(\mathcal{X}, \mathcal{B}(\mathcal{X}))$. SDEs like (6) include Langevin dynamics and Ornstein-Uhlenbeck processes. The IG $L$ associated with (6) is a second-order differential operator, defined for $f \in \mathcal{L}^2_\pi(\mathcal{X})$ and $x \in \mathcal{X}$, as:

$$Lf(x) = \nabla f(x)^\top a(x) + \tfrac{1}{2} \mathrm{Tr}\big[ b(x)^\top (\nabla^2 f(x)) b(x) \big], \quad (7)$$

where $\nabla^2 f = (\partial_{ij}^2 f)_{i,j \in [d]}$ is the Hessian of $f$. Its domain is the Sobolev space $\mathcal{W}_\pi^{1,2}(\mathcal{X}) = \{ f \in \mathcal{L}^2_\pi(\mathcal{X}) \mid \|f\|_{\mathcal{L}^2_\pi} + \|\nabla f\|_{\mathcal{L}^2_\pi} < \infty \}$. Its spectral decomposition allows one to solve SDE (6), that is

$$\mathbb{E}[f(X_t) \, | \, X_0 = x] = \sum_{i \in \mathbb{N}} e^{\lambda_i t} \langle g_i, f \rangle_{\mathcal{L}^2_\pi} f_i(x). \qquad (8)$$

*Example* 2.1 (Overdamped Langevin). Let $\sigma$, $k_b$, and $T \in \mathbb{R}_+^*$. The *overdamped Langevin* dynamics of a particle in a potential $V : \mathbb{R}^d \to \mathbb{R}$ satisfies (6) with $a = -\gamma^{-1} \nabla V$ and $b \equiv \sqrt{2(k_b T/\gamma)}$, where $\gamma$, $k_b$, and $T$ are the friction coefficient, Boltzmann constant, and system temperature, respectively. The invariant measure is the *Boltzmann distribution* $\pi(dx) \propto e^{-V(x)/(k_b T)} dx$. In dissipative systems, the IG $L$ is sectorial, with its spectrum usually in the left half-plane.

| Aspect | (Cabannes & Bach, 2024) | (Hou et al., 2023) | (Kostic et al., 2024a) | Our work |
|---|---|---|---|---|
| Many Markov Processes | ✗ (only Laplacian) | ✗ (only Itô) | ✗ (only Dirichlet form) | ✓ |
| Risk metric | $\mathcal{L}_\pi^2$ norm | $\mathcal{L}_\pi^2$ norm | Weighted Sobolev norm | $\mathcal{L}_\pi^2$ norm |
| Required prior knowledge | ✗ | ✓(full info. needed) | ✓ (partial info. needed) | ✗ |
| Data | iid | trajectory | iid | trajectory |
| IG error bound | $\mathcal{O}(n^{-\frac{d}{2(d+1)}})$ | $\mathrm{Var} = \mathcal{O}(\frac{d^2}{\gamma^2\sqrt{n}})$ | $\mathcal{O}(n^{-\frac{\alpha}{2(\alpha+\beta)}})$ | $\mathcal{O}(n^{-\frac{\alpha}{2(\alpha+\beta)}})$ |
| Spectral rates | ✗ (spuriousness) | ✗ (spuriousness) | ✓ (self-adjoint) | ✓ (sectorial) |
| Computational complexity | $\mathcal{O}(n^2+n^{3/2}d)$ | $\mathcal{O}(n^3d^3)$ | $\mathcal{O}(r\,n^2d^2)$ | $\mathcal{O}((r\vee d)\,n^2)$ $\mathcal{O}(rn(\sqrt{n}\vee N)\vee dnN)$ |

*Table 1.* Comparison to previous kernel-based works on generator learning. State-space dimension is $d$, $N$ is the number of features (possibly $N=\infty$), $n$ is a sample size and $r \ll \max(n,N)$ is estimator's rank. Our learning bounds are derived in Thm. 6.2 where parameters $\alpha \in [1,2]$ and $\beta \in (0,1]$ quantify the intrinsic difficulty of the problem and the impact of kernel choice on IG learning.

*Example* 2.2 (Ornstein-Uhlenbeck (OU) processes). The OU process with drift is governed by the SDE (6) with $a(x) = Ax$ and $b \equiv B$, where $A \in \mathbb{R}^{d\times d}$ is the drift matrix and $B \in \mathbb{R}^{d\times d}$ is the diffusion matrix. This models systems like the Vasicek interest rate and neural dynamics, where fluctuations return to equilibrium. If the real parts of $A$'s eigenvalues are negative, the OU process has an invariant Gaussian distribution with covariance $\Sigma_\infty$ satisfying Lyapunov's equation: $A\Sigma_\infty + \Sigma_\infty A^\top = -BB^\top$.

## 3. Problem formulation

Let $\mathcal{H}$ be an RKHS with kernel $k : \mathcal{X} \times \mathcal{X} \to \mathbb{R}$, and $\phi : \mathcal{X} \to \mathcal{H}$ be a *feature map* such that $k(x,x') = \langle\phi(x),\phi(x')\rangle$ for all $x,x' \in \mathcal{X}$. We assume that $\mathcal{H} \subset \mathcal{L}_\pi^2(\mathcal{X})$, enabling us to approximate $L : \mathcal{L}_\pi^2(\mathcal{X}) \to \mathcal{L}_\pi^2(\mathcal{X})$ with an operator $G : \mathcal{H} \to \mathcal{H}$ (Kostic et al., 2022). Although $\mathcal{H}$ is a subset of $\mathcal{L}_\pi^2(\mathcal{X})$, they have different metric structures, so for $f,g \in \mathcal{H}$, $\langle f,g\rangle_\mathcal{H} \neq \langle f,g\rangle_{\mathcal{L}_\pi^2}$. To resolve this, we introduce the *injection operator* $S_\pi : \mathcal{H} \to \mathcal{L}_\pi^2(\mathcal{X})$, which maps each $f \in \mathcal{H}$ to its pointwise equivalent in $\mathcal{L}_\pi^2(\mathcal{X})$ with the appropriate $\mathcal{L}_\pi^2$ norm. For bounded kernels this operator is Hilbert-Schmidt, allowing one to efficiently learn bounded operators on $\mathcal{L}_\pi^2$ via finite rank approximations.

While regressing directly the generator might lead to learning spurious spectra due to its unbounded nature, the resolvent operator is bounded for $\mu > 0$, and, hence, recalling (5), the standard regression risk is well defined via

$$\mathcal{R}(G)=\sum_{k\in\mathbb{N}} \mathbb{E}_{X_0\sim\pi} \left| \int_0^\infty h_k(X_t)e^{-\mu t}dt-[Gh_k](X_0)\right|^2,$$

where $(h_k)_{k\in\mathbb{N}}$ is any orthonormal system of $\mathcal{H}$. Next, if we define the target feature via Bochner integral as

$$\psi(X_0) = \int_0^\infty \phi(X_t)\, e^{-\mu t}dt, \qquad (9)$$

the risk can be equivalently written as the mean square error

(MSE) w.r.t. stationary distribution $\pi$

$$\mathcal{R}(G) = \mathbb{E}_{X_0\sim\pi} \|\psi(X_0) - G^*\phi(X_0)\|_\mathcal{H}^2, \qquad (10)$$

and we can show, c.f. App. A.2, the universal approximation result for its minimizers over bounded operators in $\mathcal{H}$. Namely, since $\mathcal{R}_{\mathrm{ex}}(\widehat{G}) = \mathcal{R}(\widehat{G}) - \min_G \mathcal{R}(G) = \left\|R_\mu S_\pi - S_\pi\widehat{G}\right\|_{\mathrm{HS}}^2$, if $\mathcal{H}$ is dense in $\mathcal{L}_\pi^2(\mathcal{X})$ and the injection operator is Hilbert-Schmidt, then one can find arbitrarily good finite-rank approximations of $R_\mu S_\pi$. However, learning the IG alone is insufficient for forecasting the process, and estimating the spectral decomposition of $L$ is of greater interest. But, as noted in (Kostic et al., 2023), as the estimator's rank increases, metric distortion between $\mathcal{H}$ and $\mathcal{L}_\pi^2(\mathcal{X})$ hinders learning.

## 4. Approach and main results

The main bottleneck in the risk functional (10) is the integral computation in (9), which hinders standard operator regression methods. In this section, starting from a simple idea to approximate (9) with numerical integration schemes, we present a novel fully data-driven method that addresses this difficulty. Namely, consider

$$\psi_m(X_0) = \sum_{j=0}^\ell m_j\phi(X_{t_j}), \qquad (11)$$

where $m = (m_j)_{j=0}^\ell$ are real weights given by the famous trapezoid rule with $\ell \geq 1$ points and time-discretization $\Delta t > 0$, that is

$$t_j = j\Delta t \text{ and } m_j = \begin{cases} \frac{\Delta t}{2}\,e^{-\mu\,t_j} & \text{if } j\in\{0,\ell\}, \\ \Delta t\,e^{-\mu\,t_j} & \text{if } 1 \leq j \leq \ell-1. \end{cases} \qquad (12)$$

So, we estimate $R_\mu$ by learning

$$\widetilde{R}_m := \sum_{j=0}^\ell m_j A_{t_j}, \qquad (13)$$

in which case the problem of minimizing the risk (10) over $G \colon \mathcal{H} \to \mathcal{H}$ transforms to

$$\min_{G \colon \mathcal{H} \to \mathcal{H}} \widetilde{\mathcal{R}}(G) = \mathbb{E}_{X_0 \sim \pi} \left\| \psi_m(X_0) - G^* \phi(X_0) \right\|_{\mathcal{H}}^2, \quad (14)$$

which can be solved by *Reduced Rank Regression* (RRR) estimator proposed in (Kostic et al., 2022). Namely, to learn (5), we constrain (14) to rank-$r$-RKHS operators $G \in \mathrm{B}_r(\mathcal{H}) := \{G \colon \mathcal{H} \to \mathcal{H} \,|\, \mathrm{rank}(G) \leq r\}$, and obtain the solution

$$\widetilde{G}_{m,\gamma}^r = C_\gamma^{-1/2} [\![ C_\gamma^{-1/2} \widetilde{H}_m ]\!]_r, \quad (15)$$

where $\widetilde{H}_m = S_\pi^* \widetilde{R}_m S_\pi = \sum_{j=0}^{\ell} m_j T_{t_j}$ is the aggregated cross-covariance obtained by combining $T_{t_j} = S_\pi^* e^{t_j L} S_\pi = \mathbb{E}_{X_0 \sim \pi}[\phi(X_0) \otimes \phi(X_{t_j})]$ being the cross-covariance operators in RKHS $\mathcal{H}$ w.r.t. invariant measure $\pi$, and $C_\gamma = T_0 + \gamma I$ is the regularized covariance, since $C = T_0 = S_\pi^* S_\pi = \mathbb{E}_{X \sim \pi}[\phi(X) \otimes \phi(X)]$.

While the computational details for deriving the empirical version of (15), denoted by $\widehat{G}_{m,\gamma}^r$, are presented in Sec. 5, the main challenge in the statistical analysis of the risk/error bounds, compared to the standard TO case, lies in addressing both the bias from approximating the integral and the variance from non-iid data collected along a single trajectory sampled at frequency $1/\Delta t$. While the general case is discussed in Sec. 6, we focus here on well-specified learning problems using any universal bounded kernel, specifically for eigenvalue estimation of self-adjoint operators.

Due to the unbounded nature of the generator, (13) with the choice of (12) may not always provide a good approximation of the integral transform (5). However, for a large class of problems with sectorial IG, such as Examples 2.1 and 2.2, we are able to prove, see App. C.2.4, that $\|R_\mu - \widetilde{R}_m\| \leq c\,\Delta t$, where $c$ is a constant when $\ell \mu \Delta t$ is sufficiently large, and, consequently, obtain that the difference between the true risk and its approximation is bounded in terms of the time-lag parameter. Concerning the variance, the main challenge is accounting for the unavoidable data dependence by aggregating concentration inequalities at multiples of the initial time-lag, leveraging the mixing time from geometric ergodicity. Our approach reveals the impact of key parameters (shift $\mu > 0$, regularization $\gamma > 0$, $L$ eigenvalues, and time-lag $\Delta t$) on the variance.

Putting both analyses together, we bound the excess risk of $\widehat{G} = \widehat{G}_{m,\gamma}^r$ in the operator norm. That is, for the the operator norm error $\mathcal{E}(\widehat{G}) = \left\| R_\mu S_\pi - S_\pi \widehat{G} \right\|_{\mathcal{H} \to \mathcal{L}_\pi^2}$ with probability at least $1 - \delta$ over samples drawn at frequency $1/\Delta t$, we obtain that

$$\mathcal{E}(\widehat{G}_{m,\gamma}^r) \lesssim \frac{1}{\mu - \lambda_{r+1}} + \Delta t + \frac{\ln^{3/2}(n/\delta)}{\mu \sqrt{n \Delta t |\lambda_2|}} \quad (16)$$

holds for large enough $n = 2\ell$, where the regularization parameter is chosen as $\gamma \asymp 1/(n\Delta t)$.

Analyzing (16), when the sampling frequency is $1/\Delta t \asymp n^{1/2}$ and the hyperparameters are chosen such that $|\lambda_{r+1}| \geq n^{1/2}$, $\gamma \asymp n^{-1/4}$ and $\mu \asymp \sqrt{1/\Delta t}$, the learning rate for the operator norm error is $n^{-1/2} \ln^{3/2}(n)$. This matches the learning rates in (Kostic et al., 2023; 2024a) for TO and $R_\mu$, respectively.

Further, the error/risk analysis led to spectral learning rates, specifically for estimating the eigenvalues and eigenfunctions of $L$. The key difference in the analysis is that the hypothetical domain $\mathcal{H}$ typically has a different geometry (norm) than the true domain $\mathcal{L}_\pi^2$ (Kostic et al., 2022; 2023). To control the potential deterioration of spectral learning rates relative to the risk/error, one must analyze the *metric distortion* of the estimator's eigenfunctions, defined as $\eta(h) = \|h\|_{\mathcal{H}} / \|h\|_{\mathcal{L}_\pi^2}$, for $h \in \mathcal{H}$. When the metric distortion is uniformly bounded (as can occur in well-specified settings), the eigenvalue bound becomes

$$\frac{|\lambda_i - \widehat{\lambda}_i|}{|\mu - \lambda_i| |\mu - \widehat{\lambda}_i|} - \frac{\sigma_{r+1}(R_\mu S_\pi)}{\sigma_r(R_\mu S_\pi)} \lesssim \Delta t + \frac{\ln^{3/2}(n^2/\delta)}{\mu \sqrt{n \Delta t}}, \quad (17)$$

where $\widehat{\lambda}_i = \mu - 1/\widehat{\nu}_i$ are estimates of the generator's eigenvalues $\lambda_i$ for $i \in [r]$, with $\widehat{G}_{m,\gamma}^r = \sum_{i \in [r]} \widehat{\eta}_i \, \widehat{h}_i \otimes \widehat{g}_i$ being the spectral decomposition. Note that (17) can be transformed into eigenfunction bounds using standard arguments, see App. C.5 for technical details.

Investigating the spectral learning rate in (17), we find that tuning $\mu$ in an unbounded manner is prohibitive; if $\mu$ is too large, the resolvent's eigenvalues collapse to zero. Thus, for spectral estimation, we must fix a small $\mu > 0$, which influences the rate and the optimal relationship between $n$ and $\Delta t$. Specifically, the spectral learning rate becomes $n^{-1/3} \ln^{1/2}(n^2/\delta)$, corresponding to a sampling frequency of $1/\Delta t \asymp n^{1/3}$. As expected, the estimation bias depends on the singular value gap of the resolvent operator restricted to the RKHS, given by $R_{\mu|_{\mathcal{H}}} = R_\mu S_\pi$. Finally, note that metric distortion of eigenfunctions can be estimated from data, allowing for spectral bounds with empirical biases for each eigenpair (see Sec. 6).

In conclusion, while TO methods can learn the IG's eigenfunctions by learning $A_{\Delta t} = e^{\Delta t\, L}$, there are currently no theoretical guarantees for small $\Delta t$. This motivated methods that learn the IG directly. Table 1 contrasts these methods with our contribution, which is applicable to more general settings and guarantees learning the leading eigenvalues with lower computational complexity under mild conditions. Compared to (Kostic et al., 2024a), which relies on the Dirichlet form, our estimator is more general (e.g., applicable to underdamped Langevin diffusion) and offers linear complexity in terms of state dimension, albeit with lower bias and higher variance. For details, see Sec. 6.

## 5. Learning algorithms

In this section we assume the access to the dataset $\mathcal{D}_n = (x_{i-1})_{i\in[n]}$ obtained by sampling the process $(X_t)_{t\geq 0}$ at some sampling frequency $1/\Delta t$ for $\Delta t > 0$ being typically small in order to observe all the relevant time-scales of the process. To simplify analysis, we will assume stationarity, i.e. $X_0 \sim \pi$, and, hence, $X_{j\Delta t} \sim \pi$. Clearly, to derive empirical risk we need to replace the expectation in (10) with the empirical mean, which leads to estimating cross-covariance operators by their empirical counterparts

$$\widehat{H}_m = \sum_{j=0}^{\ell} \frac{m_j}{n-j} \sum_{i=0}^{n-j-1} \phi(x_i) \otimes \phi(x_{i+j}), \qquad (18)$$

noting that for the time-lag $j\Delta t$ we can only observe $n-j$ pairs from the joint distribution $\rho_{j\Delta t}$, that is $\widehat{T}_{j\Delta t} = \frac{1}{n-j} \sum_{i=0}^{n-j-1} \phi(x_i) \otimes \phi(x_{i+j})$, $j=0,\ldots,\ell$.

Therefore, we can construct the empirical RRR estimator of $\widetilde{R}_m$ as $\widehat{G}_{m,\gamma}^r = \widehat{C}_\gamma^{-1/2} [\![ \widehat{C}_\gamma^{-1/2} \widehat{H}_m ]\!]_r$. In particular, when $r = n$, it coincides with the standard ridge regression estimator.

In the reminder of this section we show how to compute the estimator and its eigenvalue decomposition in both settings, when the finite dictionary of $N$ features spans $\mathcal{H}$ (Algorithm 1) and when $\mathcal{H}$ is infinite dimensional RKHS (Algorithm 2). To derive them we follow general construction of vector-valued RRR estimator developed in (Turri et al., 2023), detailed in App. B. To that end, recall the definition of the sampling operator $\widehat{S} \colon \mathcal{H} \to \mathbb{R}^n$ and its adjoint $\widehat{S}^* \colon \mathbb{R}^n \to \mathcal{H}$

$$\widehat{S}h = \tfrac{1}{\sqrt{n}}(h(x_{i-1}))_{i\in[n]} \text{ and } \widehat{S}^*v = \tfrac{1}{\sqrt{n}}\sum_{i\in[n]} v_i \phi(x_{i-1}),$$

implying that $\widehat{T}_{j\Delta t} = \frac{n}{n-j} \widehat{S}^* (\sum_{i\in[n]} \mathbb{1}_i \mathbb{1}_{i+j}^\top) \widehat{S}$, where $(\mathbb{1}_i)_{i\in[n]} \subset \mathbb{R}^n$ is the standard basis. Then, using (18), we obtain that $\widehat{H}_m = \widehat{S}^* M \widehat{S}$, where $M \in \mathbb{R}^{n\times n}$ is a Toeplitz matrix (i.e. has constant diagonals) given by

$$M_{i,i+j} := \begin{cases} (nm_j)/(n-j) & , i\in[n], 0\leq j\leq \ell, \\ 0 & , \text{otherwise.} \end{cases} \qquad (19)$$

When the process is time-reversal invariant, meaning that $T_t$, and consequently $\widetilde{R}_m$, are self-adjoint, we can enforce symmetry in the empirical objects by estimating $T_{j\Delta t} \approx \frac{1}{2}(\widehat{T}_{j\Delta t} + \widehat{T}_{j\Delta t}^*)$, which can be done at no cost by replacing $M$ with $\frac{1}{2}(M+M^\top)$. Consequently, both formulations of the algorithm solve a symmetric eigenvalue problems, resulting in real eigenvalues and avoiding additional numerical errors.

In finite-dimensional $\mathcal{H}$, we have that $\phi(x) = z(x)^\top z(\cdot)$, where $z = [z_1, \ldots, z_N]^\top$ is a vector of features that span $\mathcal{H}$, that is $\mathcal{H} = \{h = v^\top z \mid v \in \mathbb{R}^N\}$. Thus, operator (18) becomes isometrically isomorphic to a matrix computed by replacing $\phi$ by $z$. Thus, estimator $\widehat{G}_{m,\gamma}^r$ can be expressed as a $N \times N$ matrix in basis $(z_i)_{i\in[N]}$.

---

**Algorithm 1** Primal LaRRR

**Require:** dictionary of functions $(z_i)_{i\in[N]}$; hyperparameters $\mu > 0$, $\gamma > 0$ and $r \in [n]$.
1: Compute $Z = [z(x_0) \mid \ldots \mid z(x_{n-1})] \in \mathbb{R}^{N\times n}$
2: **if** self-adjoint **then** {using Toeplitz matrix (19)}
3:     Symmetrize $M \leftarrow (M+M^\top)/2$
4: **end if**
5: Solve eigenvalue problem $HH^\top v_i = \widehat{\sigma}_i^2 C_\gamma v_i$, $i \in [r]$, where $Z_f = ZM$, $H = \frac{1}{n} Z_f Z^\top$ and $C_\gamma = \frac{1}{n} ZZ^\top + \gamma I$
6: Normalize $v_i \leftarrow v_i/(v_i^\top C_\gamma v_i)^{1/2}$, $i \in [r]$
7: Form $V_r = [v_1 \mid \ldots \mid v_r] \in \mathbb{R}^{N\times r}$
8: Compute eigentriples $(\nu_i, w_i^l, w_i^r)$ of $V_r^\top H V_r$
9: Construct $\widehat{g}_i = z^\top H V_r w_i^l$ and $\widehat{h}_i = z^\top V_r w_i^r$
10: Compute eigenvalues $\widehat{\lambda}_i = \mu - 1/\nu_i$
**Ensure:** Estimated eigentriples $(\widehat{\lambda}_i, \widehat{g}_i, \widehat{h}_i)_{i\in[r]}$ of $L$

---

Alternatively, to derive dual Alg. 2, applicable to infinite-dimensional $\mathcal{H}$, we perform computations in "sample" space, relying on the reproducing property $h(x) = \langle h, \phi(x) \rangle_{\mathcal{H}}$ and kernel Gram matrices $K = n^{-1}[k(x_i, x_j)]_{i,j\in[n]} \in \mathbb{R}^{n\times n}$ and $K_\gamma = K + \gamma I$.

---

**Algorithm 2** Dual LaRRR

**Require:** kernel $k$; hyperparameters $\mu > 0$, $\gamma > 0$ and $r \in [n]$.
1: Compute $K = n^{-1}[k(x_i, x_j)]_{i,j\in[n]} \in \mathbb{R}^{n\times n}$
2: **if** self-adjoint **then** {using Toeplitz matrix (19)}
3:     Symmetrize $M \leftarrow (M+M^\top)/2$
4: **end if**
5: Solve eigenvalue problem $K_f K u_i = \widehat{\sigma}_i^2 K_\gamma u_i$, $i \in [r]$, where $K_f = MKM^\top$
6: Normalize $u_i \leftarrow u_i/(u_i^\top KK_\gamma u_i)^{1/2}$, $i \in [r]$
7: Form $U_r = [u_1 \mid \ldots \mid u_r] \in \mathbb{R}^{n\times r}$ and $V_r = KU_r$
8: Compute eigentriples $(\widehat{\nu}_i, w_i^l, w_i^r)$ of $V_r^\top M V_r$
9: Construct $\widehat{g}_i = \widehat{S}^* M^\top V_r w_i^\ell/\overline{\nu}_i$ and $\widehat{h}_i = \widehat{S}^* U_r w_i^r$
10: Compute eigenvalues $\widehat{\lambda}_i = \mu - 1/\nu_i$
**Ensure:** Estimated eigentriples $(\widehat{\lambda}_i, \widehat{g}_i, \widehat{h}_i)_{i\in[r]}$ of $L$

---

In both algorithms, the most expensive computation is in line 5. Naive computations result in cubic complexity w.r.t feature dimension $N$ (primal) or sample size $n$ (dual). However, using classical iterative solvers, like Lanczos or the generalized Davidson method to compute the leading eigenvalues of the generalized eigenvalue problem, when $r \ll n$ this cost can significantly be reduced, c.f. (Hogben, 2006). Namely, assuming that complexity of computing a kernel is linear in $d$, we obtain the complexity of primal LaRRR to be $\mathcal{O}((r \vee d)\, nN \vee r(n\ell \vee N^2))$, while the complexity of dual one is $\mathcal{O}((r \vee d)\, n^2)$.

In light of (8), even when the SDE in (6) is unknown, Algorithms 1 and 2 enable the construction of approximate

solutions from a single (long) simulated trajectory by estimating the dominant spectrum of $L$. Specifically, they allow approximating conditional expectations as

$$\mathbb{E}[h(X_t) \mid X_0 = x] \approx \sum_{i \in [r]} e^{\widehat{\lambda}_i t} \langle \widehat{g}_i, h \rangle_{\mathcal{H}} \widehat{h}_i(x), \quad (20)$$

where $\langle \widehat{g}_i, h \rangle_{\mathcal{H}}$ can be computed on the training set via the kernel trick, see (Kostic et al., 2022). This approach is particularly interesting for high-dimensional state spaces, where classical numerical methods become unfeasible due to the curse of dimensionality, making data-driven methods a key tool in fields like molecular dynamics, (Schütte et al., 2023). We prove in Sec. 6 that the precision of our method does not depend on the state dimension, but on intrinsic effective dimension of the process, which, together with its linear complexity w.r.t. $d$, makes it an attractive approach in such problems. Finally, we remark that (20) enables forecasting of full state distributions and not just the mean (e.g. $f$ can be an indicator function), noting that this formula extends to all $\mathcal{L}_\pi^2$ functions, at the price of an additional projection error, which leads to predicting the evolution of distributions, c.f. (Kostic et al., 2024b).

## 6. Statistical learning guarantees

We derive now statistical bounds for estimating IG's resolvent using the Laplace transform-based Reduced Rank Regression algorithm (LaRRR). We then derive learning rates for IG's eigenvalues and eigenfunctions, assuming the RKHS is generated by an universal kernel $k$, hence using Alg. 2 to compute $\widehat{G}_{m,\gamma}^r$.

We start with the following auxiliary result, essentially proven in (Kostic et al., 2022). It shows that estimated eigenvalues in the operator regression are guaranteed to lie in the $\epsilon$-pseudospectrum $\mathrm{Sp}_\epsilon$ of the true operator (union of all spectra of $\epsilon$ perturbed operators), where $\epsilon$ depends on the operator norm error $\mathcal{E}(\widehat{G}) = \|R_\mu S_\pi - S_\pi \widehat{G}\|_{\mathcal{H} \to \mathcal{L}_\pi^2}$ and the metric distortion $\eta(h) := \|h\|_{\mathcal{H}} / \|h\|_{\mathcal{L}_\pi^2}$, $h \in \mathcal{H}$, of estimated eigenfunctions. To obtain the result, the latter is either uniformly bounded or empirically estimated.

**Proposition 6.1.** *Let $\widehat{G} = \sum_{i \in [r]} \widehat{\nu}_i \, \widehat{h}_i \otimes \widehat{g}_i$ be the spectral decomposition of $\widehat{G} \colon \mathcal{H} \to \mathcal{H}$, and denote the empirical metric distortions as $\widehat{\eta}_i = \|\widehat{h}_i\| / \|\widehat{S}\widehat{h}_i\|$, $i \in [r]$. Then for every $\mu > 0$, $\Delta t > 0$, $\ell \geq 1$ and $i \in [r]$,*

$$\frac{1}{\|(\widehat{\nu}_i I - R_\mu)^{-1}\|} \leq \varepsilon_i = \max\left(\mathcal{E}(\widehat{G})\widehat{\eta}(\widehat{h}_i), \frac{\mathcal{E}(\widehat{G})\|\widehat{G}\|}{\sigma_r(R_\mu S_\pi) - \mathcal{E}(\widehat{G})}\right),$$

*implying that $\widehat{\nu}_i$ belongs to $\varepsilon_i$-pseudospectrum of $R_\mu$.*

To bound the operator norm version of the excess risk, $\mathcal{E}(G) = \|R_\mu S_\pi - S_\pi G\|_{\mathcal{H} \to \mathcal{L}_\pi^2}$, we make the following assumptions that quantify the complexity of learning problem and suitability of the chosen RKHS:

**(BK)** *Boundedness.* There exists $c_{\mathcal{H}} > 0$ such that $\mathrm{ess\,sup}_{x \sim \pi} \|\phi(x)\|^2 \leq c_{\mathcal{H}}$, i.e. $\phi \in \mathcal{L}_\pi^\infty(\mathcal{X}, \mathcal{H})$;

**(RC)** *Regularity.* For some $\alpha \in (0, 2]$ there exists $c_\alpha > 0$ such that $H_\mu H_\mu^* \preceq (c_\alpha/\mu)^2 C^{1+\alpha}$, with $H_\mu = S_\pi^* R_\mu S_\pi = \int_0^\infty T_t e^{-\mu t} dt$.

**(SD)** *Spectral Decay.* There exists $\beta \in (0, 1]$ and $c_\beta > 0$ s.t. $\lambda_j(C) \leq c_\beta \, j^{-1/\beta}$, for all $j \in J$.

These assumptions, which originate from the state-of-the-art statistical learning theory for regression in RKHS (Fischer & Steinwart, 2020), have been extended to TO regression (Li et al., 2022) and to learning self-adjoint IG of diffusions (Kostic et al., 2024a). Condition **(BK)** ensures that $\mathcal{H} \subseteq \mathcal{L}_\pi^2$, while **(SD)** quantifies the regularity of $\mathcal{H}$. Similar to the regularity condition in (Kostic et al., 2023), **(RC)** quantifies the relationship between the hypothesis class (bounded operators in $\mathcal{H}$) and the object of interest, $R_\mu$. Specifically, **(RC)** holds if $L$ has eigenfunctions in the $\alpha$-interpolation space between $\mathcal{H}$ and $\mathcal{L}_\pi^2(\mathcal{X})$. If $f_i \in \mathcal{H}$ for all $i \in \mathbb{N}$, then $\alpha \geq 1$ (see App. C.1). Since the worst-case bound on metric distortion in Prop. 6.1 depends on the estimator's norm, $\alpha$ must be restricted to $[1, 2]$ to avoid vacuous bounds, though this restriction isn't needed for empirical metric distortions. We present $\alpha \in [1, 2]$ here and analyze $\alpha < 1$ in App. C.6.

**Theorem 6.2.** *Let $L$ be sectorial operator such that $w_\star = -\lambda_2(L + L^*)/2 > 0$. Let **(BK)**, **(RC)** and **(SD)** hold for some $\alpha \in [1, 2]$ and $\beta \in (0, 1]$, respectively, and $\mathrm{cl}(\mathrm{Im}(S_\pi)) = \mathcal{L}_\pi^2(\mathcal{X})$. Given $\delta \in (0, 1)$ and $r \in [n]$, let*

$$\gamma \asymp \left(\frac{\ln^3(n/\delta)}{n \, \Delta t \, w_\star}\right)^{\frac{1}{\alpha+\beta}}, \quad \varepsilon_n^\star(\delta) = \left(\frac{\ln^3(n/\delta)}{\mu^{\frac{2(\alpha+\beta)}{\alpha}} n \, w_\star}\right)^{\frac{\alpha}{2\beta+3\alpha}} \quad (21)$$

*$\Delta t = \varepsilon_n^\star$ and $1/\ell = o(\varepsilon_n^\star)$, then there exists a constant $c > 0$, depending only on $\mathcal{H}$ and $\sigma_r(R_\mu S_\pi) - \sigma_{r+1}(R_\mu S_\pi) > 0$, such that for large enough $n \geq r$ with probability at least $1 - \delta$ in the draw of $\mathcal{D}_n$ it holds that*

$$\mathcal{E}(\widehat{G}_{m,\gamma}^r) \lesssim \max\left(\widehat{\sigma}_{r+1}, \sigma_{r+1}(R_\mu S_\pi) + c \, \varepsilon_n^\star(\delta)\right). \quad (22)$$

*Proof sketch.* Let $G_{\mu,\gamma}^r = C_\gamma^{-1/2} [\![ C_\gamma^{-1/2} H_\mu ]\!]_r$ and $\widetilde{G}_{m,\gamma}^r = C_\gamma^{-1/2} [\![ C_\gamma^{-1/2} \widetilde{H}_m ]\!]_r$, and set for brevity $\|\cdot\| := \|\cdot\|_{\mathcal{H} \to \mathcal{L}_\pi^2}$. Then we have

$$\mathcal{E}(\widehat{G}_{m,\gamma}^r) \leq \underbrace{\|R_\mu S_\pi - S_\pi G_{\mu,\gamma}\|}_{\textbf{(I) regularization bias}} + \underbrace{\|S_\pi(G_{\mu,\gamma} - G_{\mu,\gamma}^r)\|}_{\textbf{(II) rank reduction bias}}$$
$$+ \underbrace{\|S_\pi(G_{\mu,\gamma}^r - \widetilde{G}_{m,\gamma}^r)\|}_{\textbf{(III) integration bias}} + \underbrace{\|S_\pi(\widetilde{G}_{m,\gamma}^r - \widehat{G}_{m,\gamma}^r)\|}_{\textbf{(IV) estimator variance}}.$$

Regularity assumption **(RC)** guarantees that **(I)** $\leq \frac{c_\alpha}{\mu} \gamma^{\alpha/2}$. From definition of RRR, we immediately get **(II)** $\lesssim \sigma_{r+1}(\widetilde{R}_m S_\pi)$. Applying proposition in App. C.2.4 yields

**(III)** $\lesssim \Delta t$ . Results of App. C.3 gives the control on **(IV)**. Hence we get w.p.a.l. $1 - \delta$,

$$\mathcal{E}(\widehat{G}^r_{m,\gamma}) \lesssim \frac{\gamma^{\alpha/2}}{\mu} + \Delta t + \sigma_{r+1}(R_\mu S_\pi) + \frac{\ln^{3/2}\left(\frac{n\ell}{\delta}\right)}{\mu\sqrt{(n-\ell)\,\Delta t\, w_\star\, \gamma^\beta}},$$

noting that the same result holds when $\sigma_{r+1}(R_\mu S_\pi)$ is replaced by $\widehat{\sigma}_{r+1}$ computed in line 5 of Algorithms 1-2. By balancing with respect to $\gamma$ first and then with respect to $\Delta t$, we derive the final bound. $\square$

Since Prop. 6.1 transforms the estimation error via metric distortion into the pseudospectral perturbation level, it is a starting point to derive estimation of eigenvalues and eigenvectors of $R_\mu$, and consequently of $L$. The quality of such bounds depends on the properties of $L$, in particular on the conditioning of eigenvalues, that is on the angles between its eigenfunctions, or more generally its spectral projectors. The nicest case is for normal generators, where we derive the following.

**Theorem 6.3.** *Under the assumptions of Thm. 6.2, let $(\widehat{\lambda}_i, \widehat{g}_i, \widehat{h}_i)_{i\in[r]}$ be the output of Algorithm 2. If $L^*L=LL^*$, then for large enough $n \geq r$ with probability at least $1 - \delta$ in the draw of $\mathcal{D}_n$,*

$$\frac{|\lambda_i - \widehat{\lambda}_i|}{|\mu - \lambda_i||\mu - \widehat{\lambda}_i|} \leq \epsilon^\delta_{n,i} \text{ and } \left\|\widehat{f}_i - f_i\right\|^2_{\mathcal{L}^2_\pi} \leq \frac{2\epsilon^\delta_{n,i}}{[\text{gap}_i - \epsilon^\delta_{n,i}]_+},$$

*where $\epsilon^\delta_{n,i} = (\widehat{\sigma}_{r+1}\widehat{\eta}_i \wedge \sigma_{r+1}(R_\mu S_\pi)/\sigma_r(R_\mu S_\pi)) + \varepsilon^\star_n(\delta)$, $\widehat{f}_i = S_\pi \widehat{h}_i / \|S_\pi \widehat{h}_i\|_{\mathcal{L}^2_\pi}$ and $\text{gap}_i$ is the difference between $i$-th and $(i+1)$-th eigenvalue of $R_\mu$, $i \in [r]$.*

Without further details on the rich theory of spectral perturbations, note that in non-normal setting, the conditioning of eigenvalues typically comes as multiplicative factor, e.g. Bauer-Fike theorem (App. A.3).

**Comparison to other approaches.** Assume for simplicity that the RKHS aligns perfectly with $\mathcal{L}^2_\pi(\mathcal{X})$ and that $L=L^*$, meaning that $\sigma_j(A_{\Delta t}S_\pi)=\sigma_j(A_{\Delta t})$ and $\sigma_j(R_\mu S_\pi)=\sigma_{r+1}(R_\mu)$ for all $j \leq r+1$. TO methods learn $A_{\Delta t} = e^{\Delta t L}$, which shares the same eigenfunctions as the generator $L$. Analysis from (Kostic et al., 2023, Thm. 3 and Eq. 8) for TO ensures that $\|\widehat{f}_i - f_i\|^2 \leq 2|\widehat{e^{\lambda_i \Delta t}} - e^{\lambda_i \Delta t}|/[\text{gap}_i(A_{\Delta t}) - |\widehat{e^{\lambda_i \Delta t}} - e^{\lambda_i \Delta t}|]_+$, $\widehat{\lambda_i \Delta t}$ denoting the empirical estimate of this product from data. However this bound becomes vacuous as $\Delta t \to 0$ since $\text{gap}_i(A_{\Delta t}) \to 0$ but $|\widehat{e^{\lambda_i \Delta t}} - e^{\lambda_i \Delta t}| > 0$ for finite sample size. In contrast, our generator-based approach is not sensitive to time-lag, as $\text{gap}_j(R_\mu)$ is independent of $\Delta t$, guaranteeing recovery of eigenfunctions even as $\Delta t \to 0$, which is crucial for capturing fast dynamics. We note that this analysis of the bounds indeed captures the reality of the both estimators in practice, see Fig. 2 bellow and Fig. 4 in App. D.

We now compare our method to the physics-informed approach of (Kostic et al., 2024a), using the same simplifying assumptions as above for clarity. In their equation (24), they obtained:

$$\frac{|\lambda_i - \widehat{\lambda}_i|}{|\mu - \lambda_i||\mu - \widehat{\lambda}_i|} - \sqrt{\frac{\sigma_{r+1}(R_\mu)}{\sigma_r(R_\mu)}} \lesssim n^{-\frac{\alpha}{2(\alpha+\beta)}} \ln(\delta^{-1}).$$

Note that their bias term ($\sqrt{\sigma_{r+1}(R_\mu)/\sigma_r(R_\mu)}$) is larger than ours ($\sigma_{r+1}(R_\mu)/\sigma_r(R_\mu)$), while their variance term ($\propto n^{-\alpha/2(\alpha+\beta)}$) is smaller than ours ($\propto n^{-\alpha/(3\alpha+2\beta)}$). This is expected, as their method is restricted to self-adjoint IGs exploiting their structure, while our results are structure agnostic and apply to the broader class of sectorial IGs.

# 7. Experiments

In this section, we demonstrate that our LaRRR Algorithms 1 and 2 successfully recover the eigenvalues and eigenfunctions of the process's IG. We present both a 1D and 2D experiments, where we measure error against the ground truth, as well as a higher dimensional molecular dynamics experiment, where we obtain results consistent with the independent study of Mardt et al. (2018). Finally, to empirically support our claims in Tab. 1, in Fig. 5 in App. D we provide an additional experiment, showcasing that our method, despite not being physics-informed as (Kostic et al., 2024a), does not suffer from spurious estimation of IG's eigenvalues as baselines (Cabannes & Bach, 2024; Hou et al., 2023) do. The implementation of the methods, as well as all experiments are available in the GitHub repository.

**Non-normal process in 2D.** We evaluate the performance of our primal LaRRR algorithm on a one-dimensional Ornstein-Uhlenbeck with a (non-normal) $2 \times 2$ matrix. As features we use 1000 random Fourier features, and test if we can recover eigenvalues of the non-normal linear drift. In Fig. 1 we compare to TO RRR estimator over ten trials and for two different time-discretizations. While high variability in the estimation for both methods is expected due to sensitiveness of the eigenvalues to perturbations due to non-normality, we can observe that LaRRR achieves better estimation.

**Overdamped Langevin dynamics in 1D.** We evaluate the performance of our dual LaRRR algorithm on a one-dimensional Langevin process with a triple-well potential (Schwantes & Pande, 2015). Specifically, we simulate the dynamics with inverse temperature $\beta=1$ and potential $U(x)=4(x^8+0.8e^{-80x^2}+0.2e^{-80(x-0.5)^2}+0.5e^{-40(x+0.5)^2})$ that consists of three Gaussian-like wells located at $x \in \{-0.5, 0, 0.5\}$, with a bounding term proportional to $x^8$ to confine the equilibrium distribution primarily within the interval $[-1, 1]$. We generate 10 independent trajectories from this process and apply LaRRR to each trajectory in order to estimate the leading eigenvalues of the generator. By fitting 10 separate models, we assess the distribution of the relative errors in the eigenvalue

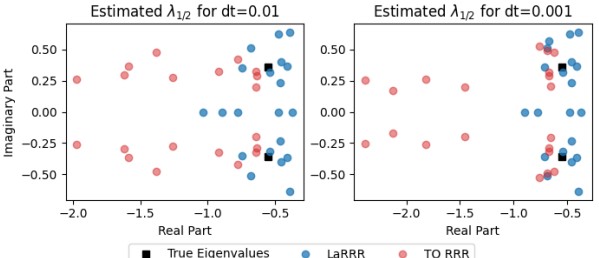

*Figure 1.* The eigenvalues of the (non-normal) $2 \times 2$ drift matrix in Ornstein-Uhlenbeck process are estimated via primal LaRRR and TO RRR methods with 1000 random Fourier features in ten trials for two different time-discretizations. Number of samples for $\Delta t = 1e-2$ is $n = 1e4$, while for $\Delta t = 1e-3$ is $n = 1e5$. This large number of samples is anticipated due to additional sensitivity of the true eigenvalues to perturbations.

estimates, defined as $|\hat{\lambda}_i - \lambda_i|/|\lambda_i|$ for each eigenvalue $\lambda_i$. Dual algorithm with RBF kernel is assessed in Fig. 2. Similar results obtained by the primal version using random Fourier features is shown in Fig. 4 in App. D.

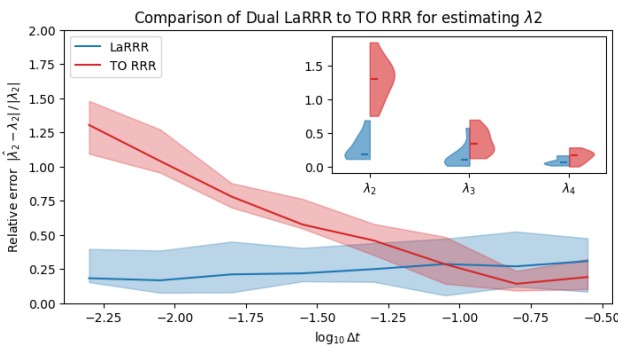

*Figure 2.* Comparison of the LaRRR dual algorithm to the TO RRR one on the task of estimating the slowest timescales of the process. As predicted by our theory, while LaRRR (blue) is not affected by decreasing the time-lag $\Delta t$, the error of TO method (red) explodes as $\Delta t \to 0$. The main figure shows three quartiles of relative error for $\lambda_2$ across 10 independent trajectories of a 1D Langevin process on a triple-well potential, while the inset figure shows distributions for the leading three eigenvalues for $\Delta t = 0.005$.

**Alanine Dipeptide.** Alanine dipeptide in water serves as a benchmark for studying dynamics due to its simplicity and metastability, where the system predominantly occupies several metastable states, that can be identified using the dihedral angles $\phi$ and $\psi$. Under reasonable assumptions, the long-term dynamics can be treated as Markovian by integrating out the water degrees of freedom. In this study, we apply our method to alanine dipeptide, using the inter-atomic distances between heavy atoms as input. Notably, the state space dimension is $d = 45$, which is intractable for the kernel based generator learning method of (Kostic et al., 2024a). We use two independent trajectories from (Nüske

et al., 2017) to train and validate the model, respectively. The recovery of metastable states corresponding to the leading two (non-trivial) eigenfunctions of IG is shown in Fig. 3: the eigenfunctions' values (color) are represented in the 2D plane of dihedral angles. Note that approximately constant values (red and blue) reveal two metastable states that align with the state-of-the-art expert knowledge in molecular dynamics, (Wehmeyer & Noé, 2018).

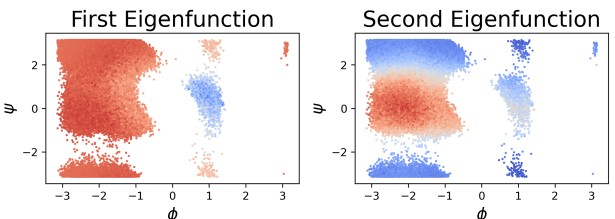

*Figure 3.* The first two non trivial eigenfunctions of the alanine dipeptide in water trained on a 250 ns simulation with $\Delta t = 50$ ps and displayed on an independent 250 ns test simulation. The color of the points corresponds to the values of the eigenfunctions.

# 8. Conclusion

We presented a first-of-its-kind method to learn continuous Markov semigroups, offering both theoretical guarantees at any time-lag and linear computational complexity in the state dimension, enabling efficient exploration of high-dimensional complex systems. Notably, our method applies to a broad range of Markov processes previously unaddressed, and overcomes the problem of TO-based methods failing to capture slow dynamics when trained on data with high sampling frequency. The main limitation of the current results lies in the assumption of uniform sampling of the (full) state of the system. While the theory can be seamlessly adapted to multiple observations sharing the same non-uniform sampling, an important challenge is to extend it to non-uniformly sampled single trajectory data, as well as to the case of partially observed systems.

**Acknowledgments.** This work was supported by the EU Project ELIAS (grant No. 101120237), and by the European Union – NextGenerationEU and the Italian National Recovery and Resilience Plan through the Ministry of University and Research (MUR), under Project PE0000013 CUP J53C22003010006. KL acknowledges support from the French National Research Agency for the DECATTLON project (ANR-24-CE40-3341). We thank the anonymous reviewers for their insightful and valuable feedback.

# Impact Statement

This paper aims to advance the field of Machine Learning. While the work may have societal implications, none require specific mention here.

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

# Supplementary Material

| notation | meaning | notation | meaning |
|---|---|---|---|
| $\wedge$ | minimum | $\vee$ | maximum |
| $\Re$ | real part of a complex number | $\Im$ | imaginary part of a complex number |
| $[\![\cdot]\!]_r$ | $r$-truncated SVD of an operator | $I$ | identity operator |
| $\mathrm{HS}\,(\mathcal{H},\mathcal{G})$ | space of Hilbert-Schmidt operators $\mathcal{H}\to\mathcal{G}$ | $\mathrm{B}_r(\mathcal{H})$ | set of finite rank-$r$ operators on $\mathcal{H}$ |
| $\|A\|$ | operator norm of an operator $A$ | $\|A\|_{\mathrm{HS}}$ | Hilbert-Schmidt norm of an operator $A$ |
| $\mathrm{F}(A)$ | numerical range an operator $A$ | $\nabla^2 f$ | Hessian of a function $f$ |
| $\sigma_i(\cdot)$ | $i$-th singular value of an operator | $\lambda_i(\cdot)$ | $i$-th eigenvalue of an operator |
| $\mathcal{X}$ | state space of the Markov process | $(X_t)_{t\geq0}$ | time-homogeneous Markov process |
| $p$ | transition kernel of the Markov process | $\pi$ | invariant measure of the Markov process |
| $a$ | drift of the Itô process | $b$ | diffusion of the Itô process |
| $\mathcal{L}^2_\pi(\mathcal{X})$ | $\mathrm{L}^2$ space of functions on $\mathcal{X}$ w.r.t. measure $\pi$ | $A_t$ | transfer operator on $\mathcal{L}^2_\pi(\mathcal{X})$ for time-step $t$ |
| $\mathcal{W}^{1,2}_\pi(\mathcal{X})$ | Sobolev space w.r.t. measure $\pi$ on $\mathcal{X}$ | $L$ | generator of the semigroup on $\mathcal{W}^{1,2}_\pi(\mathcal{X})$ |
| $\mu$ | shift parameter | $\Delta t$ | time discretization |
| $R_\mu$ | resolvent of $L$ | $\widetilde{R}_m$ | approximated resolvent of $L$ |
| $R_{\mu\mid_\mathcal{H}}$ | resolvent operator restricted to $\mathcal{H}$ | $R_\mu(L)$ | resolvent set of $L$ |
| $k(x,y)$ | kernel | $\phi$ | canonical feature map |
| $\mathcal{H}$ | reproducing kernel Hilbert space | $S_\pi$ | canonical injection $\mathcal{H}\hookrightarrow\mathcal{L}^2_\pi(\mathcal{X})$ |
| $\mathbb{1}$ | function in $\mathcal{L}^2_\pi(\mathcal{X})$ constantly equal to 1 | $\gamma$ | regularization parameter |
| $\mathcal{R}$ | true risk | $\mathcal{E}$ | operator norm error |
| $\mathcal{R}_{\mathrm{ex}}$ | excess risk, i.e. HS norm error | $\mathcal{R}_0$ | irreducible risk |
| $\widetilde{\mathcal{R}}$ | approximated risk | $\widehat{\mathcal{R}}$ | empirical risk |
| $\psi$ | Bochner integral | $\psi_m$ | approximated Bochner integral |
| $\widehat{S}$ | sampling operator w.r.t. $\mathcal{L}^2_\pi(\mathcal{X})$ | $\rho$ | joint invariant measure of the Markov process |
| $C$ | covariance operator on $\mathcal{L}^2_\pi(\mathcal{X})$ | $\widehat{C}$ | empirical covariance operator on $\mathcal{L}^2_\pi(\mathcal{X})$ |
| $C_\gamma$ | regularized covariance operator on $\mathcal{L}^2_\pi(\mathcal{X})$ | $\widehat{C}_\gamma$ | regularized emp. covariance operator on $\mathcal{L}^2_\pi(\mathcal{X})$ |
| $T_t$ | cross-covariance operator at time step $t$ | $\widehat{T}_t$ | emp. cross-covariance operator at time step $t$ |
| $H_\mu$ | integrated cross-covariance | $\widehat{H}_m$ | empirical aggregated cross-covariance |
| $\widetilde{H}_m$ | aggregated cross-covariance | $\mathrm{M}$ | Toeplitz matrix associated to $\widehat{H}_m$ |
| $\mathrm{K}$ | kernel Gram matrix | $\mathrm{K}_\gamma$ | regularized kernel Gram matrix |
| $G$ | population estimator of $R_\mu$ on $\mathcal{H}$ | $\widehat{G}$ | empirical estimator of $R_\mu$ on $\mathcal{H}$ |
| $G_{\mu,\gamma}$ | population KRR estimator | $\widehat{G}_{m,\gamma}$ | empirical KRR estimator |
| $G^r_{\mu,\gamma}$ | population RRR estimator | $\widehat{G}^r_{m,\gamma}$ | empirical RRR estimator |
| $\widetilde{G}^r_{m,\gamma}$ | approximated population RRR estimator | $\widetilde{P}$ | approximated projector |
| $P$ | spectral projector | $\widehat{P}$ | empirical spectral projector |
| $\eta$ | metric distortion | $\widehat{\eta}$ | empirical metric distortion |
| $\lambda_i$ | $i$-th generator eigenvalue | $\widehat{\lambda}_i$ | $i$-th eigenvalue of the empirical estimator |
| $f_i$ | $i$-th generator right eigenfunction in $\mathcal{L}^2_\pi(\mathcal{X})$ | $\widehat{f}_i$ | $i$-th empirical right eigenfunction in $\mathcal{L}^2_\pi(\mathcal{X})$ |
| $g_i$ | $i$-th generator left eigenfunction in $\mathcal{L}^2_\pi(\mathcal{X})$ | $\widehat{g}_i$ | $i$-th empirical left eigenfunction in $\mathcal{L}^2_\pi(\mathcal{X})$ |
| $\widehat{h}_i$ | $i$-th generator right eigenfunction in $\mathcal{L}^2_\pi(\mathcal{X})$ | $\widehat{h}_i$ | $i$-th right empirical eigenfunction |
| $\widehat{\nu}_i$ | $i$-th empirical eigenvalue | $c_\mathcal{H}$ | boundedness constant |
| $\alpha$ | regularity parameter | $c_\alpha$ | regularity constant |
| $\beta$ | spectral decay parameter | $c_\beta$ | spectral decay constant |
| $\tau$ | embedding parameter | $c_\tau$ | embedding constant |

The supplementary material is organized as follows. App. A reviews key results on Markov semigroups, RKHS operator regression, and spectral perturbation theory. App. B covers prior reduced rank regression methods and extends LaRRR to non-uniformly sampled data from multiple trajectories. App. C presents proofs of Theorems 6.2 and 6.3, including guarantees for LaRRR with non-uniform sampling. The table above summarizes the notation.

## A. Background

### A.1. Markov semigroups

Markov processes, where the future depends only on the present, are key to modeling physical, biological, and financial systems. This class includes Itô diffusions, time-dependent or singular drift processes with continuous paths, and jump processes like Poisson, Lévy, and Hawkes. Here, we focus on continuous-path Markov processes, primarily Itô diffusions. All Markov processes can be defined both as time-dependent random functions and by their laws, via measures on path space. A key tool is the infinitesimal generator (IG), a linear operator on observables.

We provide here some basics on operator theory for Markov processes. Let $\mathcal{X} \subset \mathbb{R}^d$ ($d \in \mathbb{N}$) and $(X_t)_{t\in\mathbb{R}_+}$ be a $\mathcal{X}$-valued time-homogeneous Markov process defined on a filtered probability space $(\Omega, \mathcal{F}, (\mathcal{F}_t)_{t\in\mathbb{R}_+}, \mathbb{P})$ where $\mathcal{F}_t = \sigma(X_s, s \le t)$ is the natural filtration of $(X_t)_{t\in\mathbb{R}_+}$. The dynamics of a continuous-time Markov process $X$ is described by a family of probability densities $(p_t)_{t\in\mathbb{R}_+}$

$$\mathbb{P}(X_t \in E | X_0 = x) = \int_E p_t(x,y) dy, \tag{23}$$

and *transfer operators* (TO) $(A_t)_{t\in\mathbb{R}_+}$ such that for all $t \in \mathbb{R}_+$, $E \in \mathcal{B}(\mathcal{X})$, $x \in \mathcal{X}$ and measurable function $f : \mathcal{X} \to \mathbb{R}$,

$$A_t f = \int_{\mathcal{X}} f(y) p_t(\cdot, y) dy = \mathbb{E}\big[f(X_t) \,|\, X_0 = \cdot\big]. \tag{24}$$

The transfer operator is essential for understanding the dynamics of $X$. We examine its action on $\mathcal{L}^2_\pi(\mathcal{X})$, noting the presence of an *invariant measure* $\pi$, which satisfies $A_t^* \pi = \pi$ for all $t \in \mathbb{R}_+$. In theory of Markov processes, the family $(A_t)_{t\in\mathbb{R}_+}$ is referred to as the *Markov semigroup* associated to the process $X$. The process $X$ is then characterized by the *infinitesimal generator* (IG) $L : \mathcal{L}^2_\pi(\mathcal{X}) \to \mathcal{L}^2_\pi(\mathcal{X})$ of the family $(A_t)_{t\in\mathbb{R}_+}$ defined by

$$L = \lim_{t\to0^+} \frac{A_t - I}{t}. \tag{25}$$

In other words, $L$ characterizes the linear differential equation $\partial_t A_t f = L A_t f$ satisfied by the transfer operator. The spectrum of the IG can be difficult to capture due to the potential unboundedness of $L$. To circumvent this problem, one can focus on an auxiliary operator, the resolvent, which shares the same eigenfunctions as $L$ and becomes compact under certain conditions. The following result can be found in Yosida's book ((Yosida, 2012), Chap. IX) : For every $\mu > 0$, the operator $(\mu I - L)$ admits an inverse $R_\mu = (\mu I - L)^{-1}$ that is a continuous operator on $\mathcal{X}$ and

$$(\mu I - L)^{-1} = \int_0^\infty e^{-\mu t} A_t dt.$$

The operator $L_\mu$ is the *resolvent* of $L$ and the corresponding *resolvent set* of $L$ is defined by

$$\rho(L) = \big\{\mu \in \mathbb{C} \,|\, (\mu I - L) \text{ is bijective and } L_\mu \text{ is continuous}\big\}.$$

In fact, $\rho(L)$ contains all real positive numbers and $(\mu I - L)^{-1}$ is bounded. In particular, the resolvent of a *sectorial operator* (see (38) below) is uniformly bounded outside a sector containing the spectrum. Its spectral decomposition writes

$$L = \sum_{i\in\mathbb{N}} \lambda_i \, g_i \otimes f_i \tag{26}$$

with eigenvalues $(\lambda_i)_{i\in\mathbb{N}} \subset \mathbb{C}$ and corresponding left and right eigenfunctions $f_i, g_i \in L^2$, respectively.

We detail the two examples of processes with sectorial IG discussed in the paper: the overdamped Langevin process and the Ornstein-Uhlenbeck process.

**Example 2.1 (Overdamped Lagenvin - detailed)** Let $\sigma$, $k_b$, and $T \in \mathbb{R}_+^*$. The *overdamped Langevin* dynamics of a particle in a potential $V : \mathbb{R}^d \to \mathbb{R}$ is given by the SDE: $dX_t = -\gamma^{-1}\nabla V(X_t)dt + \sqrt{2(k_bT/\gamma)}dW_t$ and $X_0 = x$,

where $\gamma$, $k_b$, and $T$ are the friction coefficient, Boltzmann constant, and system temperature, respectively. The invariant measure is the *Boltzmann distribution* $\pi(dx) = Z^{-1}e^{-V(x)/(k_bT)}dx$, where $Z$ is a normalizing constant. Its infinitesimal generator is $Lf = -\gamma^{-1}\nabla V^\top \nabla f + (k_bT/\gamma)\Delta f$, for $f \in \mathcal{W}_\pi^{1,2}(\mathcal{X})$. Since $\int(-Lf)g\,d\pi = -\int\left[\nabla\left((k_bT/\gamma)\nabla f(x)\frac{e^{-(k_bT/\gamma)^{-1}V(x)}}{Z}\right)\right]g(x)dx = (k_bT/\gamma)\int\nabla f^\top\nabla g\,d\pi = \int f(-Lg)\,d\pi$, the generator $L$ is self-adjoint. In dissipative systems, the IG $L$ is sectorial, with its spectrum usually in the left half-plane. For confining potentials, the spectrum is discrete, featuring 0 as the largest eigenvalue, which corresponds to the distribution $\pi$.

**Example 2.2 (Ornstein-Uhlenbeck - detailed)** The Ornstein-Uhlenbeck (OU) process with drift is governed by the SDE $dX_t = AX_t + BdW_t$, with $X_0 = x$, where $A \in \mathbb{R}^{d\times d}$ is the drift matrix and $B \in \mathbb{R}^{d\times d}$ is the diffusion matrix. This models systems like the Vasicek interest rate, damped harmonic oscillators, and neural dynamics, where fluctuations return to equilibrium. If the real parts of $A$'s eigenvalues are negative, the OU process has an invariant Gaussian distribution with covariance $\Sigma_\infty$ satisfying Lyapunov's equation: $A\Sigma_\infty + \Sigma_\infty A^\top = -BB^\top$. Its infinitesimal generator is defined, for $f \in L^2$, by $Lf(x) = \nabla f(x)^\top Ax + \frac{1}{2}\mathrm{Tr}[B^\top(\nabla^2 f(x))B]$. The IG has a discrete spectrum with eigenvalues related to the drift matrix $A$, where 0 corresponds to the distribution $\pi$, and the rest are negative, reflecting relaxation rates.

## A.2. Operator Regression in RKHS spaces

Recalling the operator regression problem for learning the resolvent of the generator, for the reader's convenience we state that the learning problem is well posed, the proof of which is essentially the same as in (Kostic et al., 2022).

**Proposition A.1.** *Given $\mu > 0$, let $\mathcal{H} \subseteq \mathcal{L}_\pi^2(\mathcal{X})$ be the RKHS associated to kernel $k : \mathcal{X}\times\mathcal{X} \to \mathbb{R}$ such that $S_\pi \in \mathrm{HS}\left(\mathcal{H}, \mathcal{L}_\pi^2\right)$, and let $P_\mathcal{H}$ be the orthogonal projector onto the closure of $\mathrm{Im}(S_\pi) \subseteq \mathcal{L}_\pi^2(\mathcal{X})$. Then for every $\varepsilon > 0$ there exists a finite rank operator $G : \mathcal{H} \to \mathcal{H}$ such that*

$$\mathcal{R}(G) \leq \underbrace{\|S_\pi\|_{\mathrm{HS}}^2 - \|R_\mu S_\pi\|_{\mathrm{HS}}^2}_{\mathcal{R}_0} + \underbrace{\|(I - P_\mathcal{H})R_\mu S_\pi\|_{\mathrm{HS}(\mathcal{H},\mathcal{L}_\pi^2)}^2}_{\mathcal{R}_{\mathrm{ex}}(G)} + \varepsilon.$$

*Consequently, when $k$ is universal, the excess risk can be made arbitrarily small.*

*Proof.* First, note that we can decompose

$$\mathcal{R}(G) \leq \underbrace{\|S_\pi\|_{\mathrm{HS}}^2 - \|R_\mu S_\pi\|_{\mathrm{HS}}^2}_{\mathcal{R}_0} + \underbrace{\|R_\mu S_\pi - S_\pi G\|_{\mathrm{HS}(\mathcal{H},\mathcal{L}_\pi^2)}^2}_{\mathcal{R}_{\mathrm{ex}}(G)},$$

as in (Kostic et al., 2022, Proposition 4) but now applied with additional integration. Next, since $S_\pi \in \mathrm{HS}\left(\mathcal{H}, \mathcal{W}_\pi^\mu(\mathcal{X})\right)$, according to the spectral theorem for positive self-adjoint operators, $S_\pi$ admits an SVD $S_\pi = \sum_{j\in\mathbb{N}}\sigma_j\ell_j\otimes h_j$. Recalling that $[\![\cdot]\!]_r$ denotes the $r$-truncated SVD, i.e. $[\![S_\pi]\!]_r = \sum_{j\in[r]}\sigma_j\ell_j\otimes h_j$, since $\|S_\pi - [\![S_\pi]\!]_r\|_{\mathrm{HS}}^2 = \sum_{j>r}\sigma_j^2$, for every $\delta > 0$ there exists $r \in \mathbb{N}$ such that $\|S_\pi - [\![S_\pi]\!]_r\|_{\mathrm{HS}} < \mu\delta/3$. Consequently since all the eigenvalues of $L$ have non-positive real part, $\|R_\mu(S_\pi - [\![S_\pi]\!]_r)\|_{\mathrm{HS}} \leq \|S_\pi - [\![S_\pi]\!]_r\|_{\mathrm{HS}}/\mu \leq \delta/3$. Next since $\mathrm{Im}(P_\mathcal{H}R_\mu S_\pi) \subseteq \mathrm{cl}(\mathrm{Im}(S_\pi))$, for any $j \in [r]$, there exists $g_j \in \mathcal{H}$ s.t. $\|P_\mathcal{H}R_\mu\ell_j - Z_\mu g_j\| \leq \frac{\delta}{3r}$, and, denoting $B_r := \sum_{j\in[r]}\sigma_jg_j\otimes h_j$ we conclude $\|P_\mathcal{H}R_\mu[\![S_\pi]\!]_r - S_\pi B_r\|_{\mathrm{HS}} \leq \delta/3$. Finally we recall that the set of non-defective matrices is dense in the space of matrices (Trefethen & Embree, 2020), implying that the set of non-defective rank-$r$ linear operators is dense in the space of rank-$r$ linear operators on a Hilbert space. Therefore, there exists a non-defective $G \in \mathrm{B}_r(\mathcal{H})$ such that $\|G - B_r\|_{\mathrm{HS}} < \delta/(3\sigma_1(S_\pi))$. So, we conclude

$$\|R_\mu S_\pi - S_\pi G\|_{\mathrm{HS}} \leq \|(I-P_\mathcal{H})R_\mu S_\pi\|_{\mathrm{HS}} + \|R_\mu S_\pi - [\![R_\mu S_\pi]\!]_r\|_{\mathrm{HS}} + \|[\![R_\mu S_\pi]\!]_r - S_\pi B_r\|_{\mathrm{HS}} + \|S_\pi(G - B_r)\|_{\mathrm{HS}}$$
$$\leq \|(I-P_\mathcal{H})R_\mu S_\pi\|_{\mathrm{HS}} + \delta.$$

$\square$

## A.3. Spectral perturbation theory

Recalling that for a bounded linear operator $A$ on some Hilbert space $\mathcal{H}$ the *resolvent set* of the operator $A$ is defined as $\mathrm{Res}(A) := \{\lambda \in \mathbb{C} : A - \lambda I \text{ is bijective}\}$, and its *spectrum* $\mathrm{Sp}(A) := \mathbb{C} \setminus \{\mathrm{Res}(A)\}$, let $\lambda \subseteq \mathrm{Sp}(A)$ be isolated part of spectra, i.e. both $\lambda$ and $\mu := \mathrm{Sp}(A) \setminus \lambda$ are closed in $\mathrm{Sp}(A)$. Than, the *Riesz spectral projector* $P_\lambda : \mathcal{H} \to \mathcal{H}$ is defined by

$$P_\lambda := \frac{1}{2\pi}\int_\Gamma (zI - A)^{-1}dz, \tag{27}$$

where $\Gamma$ is any contour in the resolvent set $\text{Res}(A)$ with $\lambda$ in its interior and separating $\lambda$ from $\mu$. Indeed, we have that $P_\lambda^2 = P_\lambda$ and $\mathcal{H} = \text{Im}(P_\lambda) \oplus \text{Ker}(P_\lambda)$ where $\text{Im}(P_\lambda)$ and $\text{Ker}(P_\lambda)$ are both invariant under $A$ and $\text{Sp}(A_{|\text{Im}(P_\lambda)}) = \lambda$, $\text{Sp}(A_{|\text{Ker}(P_\lambda)}) = \mu$. Moreover, $P_\lambda + P_\mu = I$ and $P_\lambda P_\mu = P_\mu P_\lambda = 0$.

Finally if $A$ is *compact* operator, then the Riesz-Schauder theorem, see e.g. (Reed & Simon, 1980), assures that $\text{Sp}(T)$ is a discrete set having no limit points except possibly $\lambda = 0$. Moreover, for any nonzero $\lambda \in \text{Sp}(T)$, then $\lambda$ is an *eigenvalue* (i.e. it belongs to the point spectrum) of finite multiplicity, and, hence, we can deduce the spectral decomposition in the form

$$A = \sum_{\lambda \in \text{Sp}(A)} \lambda P_\lambda, \tag{28}$$

where geometric multiplicity of $\lambda$, $r_\lambda := \text{rank}(P_\lambda)$, is bounded by the algebraic multiplicity of $\lambda$. If additionally $A$ is normal operator, i.e. $AA^* = A^*A$, then $P_\lambda = P_\lambda^*$ is orthogonal projector for each $\lambda \in \text{Sp}(A)$ and $P_\lambda = \sum_{i=1}^{r_\lambda} \psi_i \otimes \psi_i$, where $\psi_i$ are normalized eigenfunctions of $A$ corresponding to $\lambda$ and $r_\lambda$ is both algebraic and geometric multiplicity of $\lambda$.

Next we review well-known perturbation bounds for eigenfunctions and spectral projectors of normal compact operators, that is when $AA^* = A^*A$.

**Proposition A.2** ((Davis & Kahan, 1970)). *Let $A$ be compact self-adjoint operator on a separable Hilbert space $\mathcal{H}$. Given a pair $(\widehat{\lambda}, \widehat{f}) \in \mathbb{C} \times \mathcal{H}$ such that $\left\| \widehat{f} \right\| = 1$, let $\lambda$ be the eigenvalue of $A$ that is closest to $\widehat{\lambda}$ and let $f$ be its normalized eigenfunction. If $\widehat{g} := \min\{|\widehat{\lambda} - \lambda| \mid \lambda \in \text{Sp}(A) \setminus \{\lambda\}\} > 0$, then $\sin(\sphericalangle(\widehat{f}, f)) \leq \left\| A\widehat{f} - \widehat{\lambda}\widehat{f} \right\| / \widehat{g}$.*

**Proposition A.3** (Sinclair's theorem, see (Zwald & Blanchard, 2005)). *Let $A$ and $\widehat{A}$ be two compact operators on a separable Hilbert space. For nonempty index set $J \subset \mathbb{N}$ let*

$$\text{gap}_J(A) := \min\{|\lambda_i(A) - \lambda_j(A)| \mid i \in \mathbb{N} \setminus J, \, j \in J\}$$

*denote the spectral gap w.r.t $J$ and let $P_J$ and $\widehat{P}_J$ be the corresponding spectral projectors of $A$ and $\widehat{A}$, respectively. If $A$ is normal and for some $\|A - \widehat{A}\| < \text{gap}_J(A)$, then*

$$\|P_J - \widehat{P}_J\| \leq \frac{\|A - \widehat{A}\|}{\text{gap}_J(A)}.$$

For non-normal operators, spectral perturbation becomes much more involved. Two core objects in it are the pseudospectrum and the numerical range, which we review next.

**Definition A.4** (Pseudospectrum). Given $\epsilon > 0$, the $\epsilon$-*pseudospectrum* of a bounded operator $A$ on $\mathbb{H}$ (w.r.t. the spectral norm) is a set in the complex plane that consists of all eigenvalues of all $\epsilon$-perturbations of $A$ (w.r.t. spectral norm). Formally, it is defined as

$$\text{Sp}_\epsilon(A) = \{z \in \mathbb{C} \mid z \in \text{Sp}(A + E), \text{ for some operator } E \text{ s.t. } \|E\| \leq \epsilon\}.$$

Since, it provides insights on the sensitivity of eigenvalues, pseudospectrum is the tool of choice in the study of non-normal operators. Equivalently, it can be expressed via resolvent

$$\text{Sp}_\epsilon(A) = \left\{z \in \mathbb{C} \mid \left\|(zI - A)^{-1}\right\|^{-1} \leq \epsilon\right\},$$

implying that the lower bounds on the resolvent's norm lead to bounds on the eigenvalue sensitivity to the perturbations. Whenever the operator is normal, that is it commutes with its adjoint, indeed we have that $\text{dist}(z, \text{Sp}(A)) = \min_{\lambda \in \text{Sp}(A)} |z - \lambda| = \epsilon$ for all $z \in \partial \text{Sp}_\epsilon(A)$. On the other hand, when operators are not normal, distance of the eigenvalues of the perturbed operator to the spectrum is typically amplified. A general result showing this is the following.

**Proposition A.5** (Bauer-Fike theorem, see (Trefethen & Embree, 2020)). *Let $A$ be diagonalizable bounded operator on a separable Hilbert space, that is there exists a bounded operator $X$ with bounded inverse $X^{-1}$ such that $X^{-1}AX$ is diagonal operator. If $B_\epsilon := \{z \in \mathbb{C} \mid |z| \leq \epsilon\}$ denotes the $\epsilon$-ball in $\mathbb{C}$, then*

$$\text{Sp}(A) + B_\epsilon \subseteq \text{Sp}_\epsilon(A) \subseteq \text{Sp}(A) + B_{\epsilon\kappa(X)},$$

*where $\kappa(X) = \|X\|\|X^{-1}\|$ is the condition number of $X$. Consequently, $\text{dist}(z, \text{Sp}(A)) \leq \epsilon\kappa(X)$ for all $z \in \partial \text{Sp}_\epsilon(A)$.*

We recommend a book by (Trefethen & Embree, 2020) for an in depth study of nonnormality, pseudospectra and related quantities.

**Definition A.6** (Numerical Range). The *field of values*, known also as *numerical range*, of a bounded operator $A$ on $\mathbb{H}$ is a set in the complex plane that is closely related to the spectrum of $A$. Formally, it is defined as

$$\mathrm{F}(A) = \left\{ \frac{\langle Av, v \rangle}{\langle v, v \rangle} : v \in \mathbb{H}, v \neq 0 \right\}.$$

Due to Hausdorff's theorem, numerical range is a convex subset of $\mathbb{C}$ and that its closure contains the spectrum of $A$. It is one of the key objects in the study of operator semigroups, since it is known that

$$e^{t \max\{\Re(\lambda) \,|\, \lambda \in \mathrm{Sp}(A)\}} \leq \left\| e^{tA} \right\| \leq e^{t \max\{\Re(z) \,|\, z \in \mathrm{F}(A)\}}.$$

Finally, we conclude this review with a key result on bounding analytic functions of an operator using its numerical range.

**Theorem A.7** (Crouzeix's Theorem, see (Crouzeix & Palencia, 2017)). *Let $A$ be a bounded linear operator on a Hilbert space $\mathbb{H}$, and let $f$ be any analytic function. Then*

$$\|f(A)\| \leq (1 + \sqrt{2}) \max_{z \in \mathrm{cl}\, \mathrm{F}(A)} |f(z)|,$$

*where* $\mathrm{cl}\, \mathrm{F}(A)$ *denotes the closure of numerical range of $A$.*

# B. Methods and their extension to non-uniform sampling

In this section we discuss the Algorithms 1 and 2, together with their simple extensions to the case of learning from multiple independent copies of the process over a finite time-horizon.

For computationally efficient primal and dual form of the solution of general vector-valued reduced rank regression problem we refer to (Turri et al., 2023, Proposition 2.2). Coupling this result with classical result on solving low rank eigenvalue problems, see e.g. (Kostic et al., 2022, Theorem 2) for application in the context of TO, we obtain the final form of two algorithms.

Now, let us discuss the setting where dataset $\mathcal{D}_n = (x_{i,t_{j-1}})_{i \in [n], j \in [\ell+1]}$ consisting of $n$ trajectories sampled at the same (possibly non-uniform) times $0 = t_0 < t_1 < \ldots < t_\ell$. This is typical when we observe particle systems where $n$ copies of the same process are tracked.

In this setting, learning transfer operators becomes difficult task since, due to diverse time-lags one ends up with the non-convex optimization problems. Instead, learning IG of the process by LaRRR is direct. Namely, we simply have that $\widehat{G}^r_{m,\gamma} = \widehat{C}^{-1/2}_\gamma [\![\widehat{C}^{-1/2}_\gamma \widehat{H}_m]\!]_r$ where in this setting

$$\widehat{C}_\gamma = \frac{1}{n(\ell+1)} \sum_{i \in [n]} \sum_{j \in [\ell+1]} \phi(x_{i,t_{j-1}}) \otimes \phi(x_{i,t_{j-1}}) \quad \text{and} \quad \widehat{H}_m = \sum_{j \in [\ell+1]} m_{j-1} \Big( \frac{1}{n} \sum_{i \in [n]} \phi(x_{i,t_0}) \otimes \phi(x_{i,t_{j-1}}) \Big)$$

where now $m_0 = \frac{t_1 - t_0}{2} e^{-\mu t_0}$, $m_\ell = \frac{t_\ell - t_{\ell-1}}{2} e^{-\mu t_\ell}$ and $m_j = \frac{t_{j+1} - t_{j-1}}{2} e^{-\mu t_j}$ otherwise for $1 \leq j \leq \ell - 1$.

In this case, the sampling operator becomes

$$\widehat{S}h = (h(x_{i,t_{j-1}}))_{j \in [\ell+1], i \in [n]} = \frac{1}{\sqrt{n(\ell+1)}} \left[ h(x_{1,t_0}), \ldots, h(x_{n,t_0}), h(x_{1,t_1}), \ldots, h(x_{n,t_\ell}) \right]^\top,$$

implying that the matrix $M \in \mathbb{R}^{n(\ell+1) \times n(\ell+1)}$ is defined as

$$M_{i,jn+i} := \begin{cases} (\ell+1) m_j & , i \in [n], 0 \leq j \leq \ell, \\ 0 & , \text{otherwise}. \end{cases} \tag{29}$$

Therefore, we can readily implement Algorithms 1 and 2 also in this setting of non-uniformly sampled data. Moreover, analyzing such an estimator, as we discuss in App. C.7, from the statistical perspective, becomes an easier task than a single trajectory analysis. This is due to the fact that we observe $n$ iid copies allows the use of classical concentration inequalities without the need to asses the impact of mixing.

## C. Statistical learning theory

In this section, we provide the proofs of the main results from Section 6, along with additional discussions for the reader's convenience. We begin by stating the main assumptions, then present results on controlling the approximation error and the estimator's variance, and finally prove the main results on operator norm error and spectral learning bounds. To this end, we first decompose the operator norm error $\mathcal{E}(\widehat{G}^r_{m,\gamma}) = \|R_\mu S_\pi - S_\pi \widehat{G}^r_{m,\gamma}\|_{\mathcal{H}\to\mathcal{L}^2_\pi}$ of the Laplace transform-based Reduced Rank Regression algorithm (LaRRR) $\widehat{G}^r_{m,\gamma}$ as

$$\mathcal{E}(\widehat{G}^r_{m,\gamma}) \leq \underbrace{\|(I-P_\mathcal{H})R_\mu S_\pi\|}_{\textbf{(0) representation bias}} + \underbrace{\|P_\mathcal{H} R_\mu S_\pi - S_\pi G_{\mu,\gamma}\|}_{\textbf{(I) regularization bias}} + \underbrace{\left\|S_\pi(G_{\mu,\gamma}-G^r_{\mu,\gamma})\right\|}_{\textbf{(II) rank reduction bias}} + \underbrace{\left\|S_\pi(G^r_{\mu,\gamma}-\widetilde{G}^r_{m,\gamma})\right\|}_{\textbf{(III) integration bias}} +$$

$$\underbrace{\left\|S_\pi(\widetilde{G}^r_{m,\gamma}-\widehat{G}^r_{m,\gamma})\right\|}_{\textbf{(IV) estimator variance}}, \tag{30}$$

where $P_\mathcal{H}$ is the orthogonal projection in $\mathcal{L}^2_\pi(\mathcal{X})$ onto the $\mathrm{cl}(\mathrm{Im}(S_\pi))$, $G_{\mu,\gamma} = C^{-1}_\gamma H_\mu$ and $G^r_{\mu,\gamma}=C^{-1/2}_\gamma[\![C^{-1/2}_\gamma H_\mu]\!]_r$ are the population KRR and RRR models for the risk (10), respectively, while $\widetilde{G}_{m,\gamma} = C^{-1}_\gamma \widetilde{H}_m$ and $\widetilde{G}^r_{m,\gamma}=C^{-1/2}_\gamma[\![C^{-1/2}_\gamma \widetilde{H}_m]\!]_r$ are the population KRR and RRR models for the approximated risk (14), and, for simplicity, we abbreviate $\|\cdot\| := \|\cdot\|_{\mathcal{H}\to\mathcal{L}^2_\pi}$.

### C.1. Assumptions

In this section we discuss the assumptions that are necessary to study the learning problem. They concern the interplay between the resolvent of IG and the chosen RKHS space, which is encoded in the injection operator $S_\pi$ and the restriction of the resolvent to the RKHS $Z_\mu = R_\mu S_\pi$.

We start by observing that $S_\pi \in \mathrm{HS}\left(\mathcal{H}, \mathcal{L}^2_\pi(\mathcal{X})\right)$, according to the spectral theorem for positive self-adjoint operators, has an SVD, i.e. there exists at most countable positive sequence $(\sigma_j)_{j\in J}$, where $J := \{1, 2, \dots,\} \subseteq \mathbb{N}$, and ortho-normal systems $(\ell_j)_{j\in J}$ and $(h_j)_{j\in J}$ of $\mathrm{cl}(\mathrm{Im}(S_\pi))$ and $\mathrm{Ker}(S_\pi)^\perp$, respectively, such that $S_\pi h_j = \sigma_j \ell_j$ and $S^*_\pi \ell_j = \sigma_j h_j$, $j \in J$. Now, given $\alpha \geq 0$, let us define scaled injection operator $S_{\pi,\alpha} \colon \mathcal{H} \to \mathcal{L}^2_\pi(\mathcal{X})$ as $S_{\pi,\alpha} := \sum_{j\in J} \sigma^\alpha_j \ell_j \otimes h_j$. Notice that $S_\pi = S_{\pi,1}$, while $\mathrm{Im}\, S_{\pi,0} = \mathrm{cl}(\mathrm{Im}(S_\pi))$. Next, we equip $\mathrm{Im}(S_{\pi,\alpha})$ with a norm $\|\cdot\|_\alpha$ to build an interpolation space.

$$[\mathcal{H}]_\alpha := \left\{ f \in \mathrm{Im}(S_{\pi,\alpha}) \mid \|f\|^2_\alpha := \sum_{j\in J} \sigma^{-2\alpha}_j \langle f, \ell_j\rangle^2 < \infty \right\}.$$

We remark that for $\alpha = 1$ the space $[\mathcal{H}]_\alpha$ is just an RKHS $\mathcal{H}$ seen as a subspace of $\mathcal{L}^2_\pi(\mathcal{X})$. Moreover, we have the following injections

$$[\mathcal{H}]_{\alpha_1} \hookrightarrow [\mathcal{H}]_1 \hookrightarrow [\mathcal{H}]_{\alpha_2} \hookrightarrow [\mathcal{H}]_0 = \mathcal{L}^2_\pi(\mathcal{X}),$$

where $\alpha_1 \geq 1 \geq \alpha_2 \geq 0$.

In addition, from **(BK)** we also have that RKHS $\mathcal{H}$ can be embedded into $L^\infty_\pi(\mathcal{X})$, i.e. for some $\tau \in (0, 1]$

$$[\mathcal{H}]_1 \hookrightarrow [\mathcal{H}]_\tau \hookrightarrow L^\infty_\pi(\mathcal{X}) \hookrightarrow \mathcal{L}^2_\pi(\mathcal{X}).$$

According to (Fischer & Steinwart, 2020), if $S_{\pi,\tau,\infty} \colon [\mathcal{H}]_\tau \hookrightarrow L^\infty_\pi(\mathcal{X})$ denotes the injection operator, its boundedness implies the polynomial decay of the singular values of $S_\pi$, i.e. $\sigma^2_j(S_\pi) \lesssim j^{-1/\tau}$, $j \in J$, and the following condition holds

**(KE)** *Kernel embedding property*: there exists $\tau \in [\beta, 1]$ such that

$$c_\tau := \|S_{\pi,\tau,\infty}\|^2 = \operatorname*{ess\,sup}_{x\sim\pi} \sum_{j\in J} \sigma^{2\tau}_j |\ell_j(x)|^2 < +\infty. \tag{31}$$

Assumption **(SD)** allows one to quantify the effective dimension of $\mathcal{H}$ in ambient space $\mathcal{L}^2_\pi(\mathcal{X})$, while the kernel embedding property **(KE)** allows one to estimate the norms of whitened feature maps

$$\xi(x) := C^{-1/2}_\gamma \phi(x) \in \mathcal{H}, \tag{32}$$

as the following result states.

**Lemma C.1** ((Fischer & Steinwart, 2020)). *Let* **(KE)** *hold for some* $\tau \in [\beta, 1]$ *and* $c_\tau \in (0, \infty)$. *Then,*

$$\mathbb{E}_{x \sim \pi} \|\xi(x)\|_{\mathcal{H}}^2 \leq \begin{cases} \frac{c_\beta^\beta}{1-\beta} \gamma^{-\beta} &, \beta < 1, \\ c_\tau \gamma^{-1} &, \beta = 1. \end{cases} \quad and \quad \|\xi\|_\infty^2 = \operatorname*{ess\,sup}_{x \sim \pi} \|\xi(x)\|_{\mathcal{H}}^2 \leq c_\tau \gamma^{-\tau}. \tag{33}$$

Next, we discuss assumption **(RC)** that quantifies the regularity of the problem w.r.t. the chosen RKHS. Typical form in which well-specifiedness of the operator regression problem is expressed, see e.g. (Li et al., 2022), is to ask that $\mathcal{H}$ is invariant under the action of the operator. In our setting, this reads as $\operatorname{Im}(Z_\mu) \subseteq \operatorname{Im}(S_\pi)$. This can be relaxed (or straightened) by using interpolation spaces $[\mathcal{H}]_\alpha = \operatorname{Im}(S_{\pi,\alpha})$, which leads to **(RC)**. Indeed, according to (Zabczyk, 2020)[Theorem 2.2], the condition **(RC)** is equivalent to $\operatorname{Im}(Z_\mu) = \mu \operatorname{Im}(R_\mu S_\pi) \subseteq \operatorname{Im}(S_{\pi,\alpha})$, i.e. $G_{\mathcal{H}}^\alpha := \mu S_{\pi,\alpha}^\dagger S_\pi^* R_\mu S_\pi = \mu S_{\pi,\alpha}^\dagger H_\mu$ is bounded operator on $\mathcal{H}$ and $\mu R_\mu S_\pi = S_{\pi,\alpha} G_{\mathcal{H}}^\alpha$. In this case we have that $\mu H_\mu = S_\pi^* (\mu R_\mu) S_\pi = S_\pi^* S_{\pi,\alpha} G_{\mathcal{H}}^\alpha$, and, thus, $H_\mu H_\mu^* \preceq (\|G_{\mathcal{H}}^\alpha\|/\mu)^2 C^{1+\alpha}$. We note that re-scaling with $\mu$ is motivated by the fact that $\|\mu R_\mu\| \leq 1$.

We conclude this section discussing the assumptions for the case when RKHS is build as a span of some finite dictionary of functions $(z_j)_{j \in [N]}$, $z_j : \mathcal{X} \to \mathbb{R}$, $j \in [N]$,

$$\mathcal{H} := \left\{ h_u = \sum_{j \in [N]} u_j z_j \,\middle|\, u = (u_1, \ldots, u_N) \in \mathbb{R}^N \right\}. \tag{34}$$

The choice of the dictionary, instrumental in designing successful learning algorithms, may be based on prior knowledge on the process or learned from data (Kostic et al., 2024c; Mardt et al., 2018). The space $\mathcal{H}$ is naturally equipped with the geometry induced by the norm $\|h_u\|_{\mathcal{H}}^2 := \sum_{j=1}^N u_j^2$. Moreover, every operator $A : \mathcal{H} \to \mathcal{H}$ can be identified with matrix $\mathrm{A} \in \mathbb{R}^{N \times N}$ by $Ah_u = z(\cdot)^\top \mathrm{A}u$. In the following, we will refer to $A$ and A as the same object, explicitly stating the difference when necessary.

In this setting all spaces $[\mathcal{H}]_\alpha$ are finite dimensional, and, hence, $\operatorname{Im}(Z_\mu) \subset \operatorname{Im}(S_\pi)$ implies also $\operatorname{Im}(Z_\mu) \subset \operatorname{Im}(S_{\pi,\alpha})$ for every $\alpha > 0$. This choice makes **(RC)** for different $\alpha \in (0, 2]$ all equivalent to asking for $\mathcal{H}$ to contain *all* eigenfunctions of the generator. Moreover, in this setting we can set $\beta = \tau = 0$ and observe that in the limit $\gamma \to 0$ we have

$$\operatorname{tr}(C_\gamma^{-1} C) \to N \quad \text{and} \quad \|\xi\|_\infty^2 \to \operatorname*{ess\,sup}_{x \sim \pi} \langle \phi(x), C^\dagger \phi(x) \rangle_{\mathcal{H}} < \infty.$$

## C.2. Approximation error analysis

In this section we study the approximation errors, i.e. bias terms in (30).

### C.2.1. REPRESENTATION BIAS

The first term is the representation error that one incurs only when the hypothesis space $\mathcal{H}$ is not dense in the true domain $\mathcal{L}_\pi^2(\mathcal{X})$. That is, if one uses universal kernels, such as RBF Gaussian kernel $k(x, y) = e^{-\|x-y\|^2/l^2}$ with some length-scale $l > 0$, then $\|(I - P_{\mathcal{H}})R_\mu S_\pi\| = 0$. On the other hand, when one uses the finite-dimensional RKHS (34), this term quantifies the loos of information due to the restriction of the model to $\mathcal{H}$. In recent work by (Kostic et al., 2024c) it has been shown how to learn dictionary of functions with neural networks so that the minimal representation error is incurred from perspective of TOs. While, one can use the same method for learning an appropriate kernel for the estimation of the resolvent, an interesting future directing would be to develop representation learning based on the Laplace transform.

### C.2.2. REGULARIZATION BIAS

We control this term using the regularity assumption. The result is stated in the following proposition whose proof we omit since it follows exactly the same lines as (Kostic et al., 2023, Proposition 5) with the only difference that **(RC)** assumption holds for $R_\mu$ instead of TO with fixed lag-time.

**Proposition C.2.** *Let* $G_{\mu,\gamma} = C_\gamma^{-1} H_\mu$ *for* $\gamma > 0$, *and* $P_{\mathcal{H}} : \mathcal{L}_\pi^2(\mathcal{X}) \to \mathcal{L}_\pi^2(\mathcal{X})$ *be the orthogonal projector onto* $\operatorname{cl}(\operatorname{Im}(S_\pi))$. *If the assumptions* **(BK)**, **(SD)** *and* **(RC)** *hold, then*

$$\|G_{\mu,\gamma}\| \leq \begin{cases} (c_\alpha/\mu) c_{\mathcal{H}}^{(\alpha-1)/2} &, \alpha \in [1, 2], \\ (c_\alpha/\mu) \gamma^{(\alpha-1)/2} &, \alpha \in (0, 1], \end{cases} \quad and \quad \|P_{\mathcal{H}} R_\mu S_\pi - S_\pi G_{\mu,\gamma}\| \leq (c_\alpha/\mu) \gamma^{\frac{\alpha}{2}}. \tag{35}$$

While the previous result gives us the means to study the learning bounds when $\mathcal{H}$ is dense in $\mathcal{L}^2_\pi(\mathcal{X})$, whenever the RKHS is finite-dimensional and not learned, the regularization bias is usually coupled with the representation bias and one studies the decay of $\|R_\mu S_\pi - S_\pi G_{\mu,\gamma}\|$ as a function of $\gamma > 0$ w.r.t. $N \to \infty$ for the choice of dictionary that forms basis of $\mathcal{L}^2_\pi(\mathcal{X})$ in the limit.

### C.2.3. RANK REDUCTION BIAS

To control of this term, observe that

$$\left\| S_\pi(G_{\mu,\gamma} - G^r_{\mu,\gamma}) \right\| = \left\| C^{1/2} C^{-1}_\gamma H_\mu(I - P_r) \right\| \le \left\| C^{-1/2}_\gamma H_\mu(I - P_r) \right\| \le \sigma_{r+1}(B), \tag{36}$$

where $P_r$ is the orthogonal projector onto the leading right singular space of operator $B = C^{-1/2}_\gamma H_\mu$.

This bound on the bias can be further analyzed using the **(RC)** assumption, as indicated by the following proposition essentially proven in (Kostic et al., 2023, Propositions 6 and 7).

**Proposition C.3.** *Let $B := C^{-1/2}_\gamma H_\mu$, let **(RC)** hold for some $\alpha \in (0, 2]$. Then for every $j \in J$,*

$$\sigma^2_j(Z_\mu) - (c_\alpha/\mu)^2 \, c^{\alpha/2}_{\mathcal{H}} \, \gamma^{\alpha/2} \le \sigma^2_j(B) \le \sigma^2_j(Z_\mu). \tag{37}$$

### C.2.4. INTEGRATION BIAS

Diversely to the previous terms analyzed up to now, the control of the integration bias term is truly novel technical contribution in the study of TO and IG data-driven methods. This necessary step allows one to connect empirical objects to the population ones by approximating the integral with the finite sum.

We start with the key result showing that, for the class of sectorial IGs, we can control the difference between the resolvent and its finite-sum approximation via TOs of different times-lags using Crouzeix's theorem A.7.

**Proposition C.4.** *Let $L$ be a (stable) sectorial operator with angle $\theta \in [0, \pi/2)$, that is*

$$\mathrm{F}(L) \subseteq \mathbb{C}^-_\theta := \{z \in \mathbb{C} \mid \Re(z) \le 0 \ \wedge \ |\Im(z)| \le -\Re(z)\tan(\theta)\}, \tag{38}$$

*where $\mathrm{F}(L) = \left\{ \langle Lf, f \rangle_{\mathcal{L}^2_\pi} / \|f\|^2_{\mathcal{L}^2_\pi} : f \in \mathrm{dom}(L) \setminus \{0\} \right\}$ denotes the numerical range of $L$, and we let $(m_j)^\ell_{j=0}$ be given by the trapezoid rule*

$$m_j = \begin{cases} \frac{t_1 - t_0}{2} e^{-\mu t_0} & , j = 0, \\ \frac{t_{j+1} - t_{j-1}}{2} e^{-\mu t_j} & , 1 \le j \le \ell - 1, \\ \frac{t_\ell - t_{\ell-1}}{2} e^{-\mu t_\ell} & , j = \ell, \end{cases} \tag{39}$$

*where $0 = t_0 < t_1 < \ldots < t_\ell$ is (possibly non-uniform) discretization in time. Then, for every $\mu > 0$,*

$$\left\| R_\mu - \widetilde{R}_m \right\| \le 2t_1 + \frac{(1+\sqrt{2})\pi^2 \kappa^2}{18e^2 \cos^2(\theta)} \Delta t + \frac{e^{-\mu t_\ell}}{\mu}, \tag{40}$$

*where $\Delta t = \max_{j \in [\ell+1]}(t_j - t_{j-1})$ is the maximal time-step, $t_\ell$ is time-horizon and $\kappa = \max_{j \in [\ell+1]}(t_j - t_{j-1})/\min_{j \in [\ell+1]}(t_j - t_{j-1}) \in [1, +\infty)$ is the conditioning of discretization that measures the amount of non-uniformity. Consequently, for uniform discretization $t_j = j\Delta t$, $j = 0, \ldots, \ell$, it holds*

$$\left\| R_\mu - \widetilde{R}_m \right\| \le \left( 2 + \frac{(1+\sqrt{2})\pi^2}{18e^2 \cos^2(\theta)} + \frac{e^{-\ell\mu\Delta t}}{\mu\Delta t} \right) \Delta t = s_{\mu,\ell}(\Delta t). \tag{41}$$

*Proof.* Given $0 \le j \le \ell - 1$, remark that the functions

$$f^j_\mu(z) := \int^{t_{j+1}}_{t_j} e^{-(\mu-z)t} dt - \frac{t_{j+1} - t_j}{2}\left( e^{-(\mu-z)\,t_j} + e^{-(\mu-z)\,t_{j+1}} \right) \text{ and } g_\mu(z) := \int^\infty_{t_\ell} e^{-(\mu-z)t} dt$$

are analytic whenever $\mu = \Re(\mu) \ge \Re(z)$. Thus, since $L$ is stable, we have that

$$\left\| R_\mu - \widetilde{R}_m \right\| = \left\| \sum^{\ell-1}_{j=0} f^j_\mu(L) + g_\mu(L) \right\| \le \left\| f^0_\mu(L) \right\| + \sum^{\ell-1}_{j=1} \left\| f^j_\mu(L) \right\| + \left\| g_\mu(L) \right\|.$$

Next, using that $\|A_t\| \leq 1$ for all $t \geq 0$, we bound the first and last term as

$$\left\| f_\mu^0(L) \right\| \leq \int_0^{t_1} e^{-\mu t} dt + \tfrac{t_1}{2}\left(1 + e^{-\mu t_1}\right) = \left(\frac{1 - e^{-\mu t_1}}{\mu t_1} + \frac{1 + e^{-\mu t_1}}{2}\right) t_1 \leq 2\, t_1.$$

and

$$\|g_\mu(L)\| \leq \int_{t_\ell}^\infty e^{-\mu t} dt = \frac{e^{-\mu\, t_\ell}}{\mu}.$$

Finally, we bound $f_\mu^j(L)$ using Crouzeix's theorem A.7. To that end, we need to replace the unbounded operator $L$ with its bounded Yoshida approximation $L_s := s^2(sI - L)^{-1}L$, $s > 0$, for which we know that for every $f \in \mathrm{dom}(L)$ $L_s f \to Lf$, as $s \to \infty$, (Kato, 2012). This implies that for every $\varepsilon > 0$, there exists large enough $s > 0$ so that $\mathrm{F}(L_s) \subseteq \mathbb{C}_\theta^- + \varepsilon$. Now, since $f_{\mu+\varepsilon}^j$ is analytic over $\mathbb{C}_\theta^- + \varepsilon$, applying the Crouzeix's theorem A.7 we conclude that

$$\left\| f_{\mu+\varepsilon}^j(L_s) \right\| \leq \tfrac{1+\sqrt{2}}{2} \sup_{z \in \mathrm{cl}\, \mathrm{F}(L_s)} |f_{\mu+\varepsilon}^j(z)| \leq \tfrac{1+\sqrt{2}}{2} \sup_{z \in \mathbb{C}_\theta^-} |f_\mu^j(z)|.$$

So, since $f_\mu^j$ is the error of approximating the integral of complex valued function $t \mapsto e^{-(\mu-z)t}$ on the real interval $[j\Delta t, (j+1)\Delta t]$ by the trapezoid rule with two points, we have that (see for instance (Quarteroni et al., 2010), equation (9.12))

$$f_\mu^j(z) = -\frac{(t_{j+1} - t_j)^3}{6} \partial_t^2 [e^{-(\mu-z)t}](t_j(1-\xi) + \xi\, t_{j+1}) = -\frac{(t_{j+1} - t_j)^3}{6}(\mu - z)^2 e^{-(\mu-z)(t_j(1-\xi)+\xi\, t_{j+1})},$$

for some $\xi \in (0,1)$. Hence, for $z \in \mathbb{C}_\mu^-$,

$$|f(z)| \leq \frac{(t_{j+1}-t_j)^3}{6}|\mu - z|^2 e^{-(\mu-\Re(z))t_j} \leq \frac{(t_{j+1}-t_j)^3}{6}(1 + \tan^2\theta)(\mu - \Re(z))^2 e^{-(\mu-\Re(z))t_j} =$$

$$\leq \frac{(1+\tan^2\theta)\Delta t^3}{6}\frac{4e^{-2}}{t_j^2} = \frac{2}{3\,e^2\cos^2\theta}\frac{(t_{j+1}-t_j)^3}{t_j^2} \leq \frac{2\kappa^2}{3\,e^2\cos^2\theta}\frac{\Delta t}{j^2},$$

Therefore,

$$\sum_{j=1}^{\ell-1} \left\| f_\mu^j(L_s) \right\| \leq \Delta t \frac{(1+\sqrt{2})}{3e^2\cos^2\theta} \sum_{j=1}^{\ell-1} \frac{1}{j^2} \leq \frac{\pi^2(1+\sqrt{2})}{18e^2\cos^2\theta}\Delta t,$$

and the proof follows by letting $s \to \infty$ and noting that sequence of bounded operators $(f_\mu^j(L_s))_s$ converges to a bounded operator $f_\mu^j(L)$. $\qquad\square$

Now, appying the above result on the integration bias in (30) for the case without the rank reduction (reasonable when $r = n$), we obtain

$$\left\| S_\pi(G_{\mu,\gamma} - \widetilde{G}_{m,\gamma}) \right\| = \left\| S_\pi C_\gamma^{-1} S_\pi (R_\mu - \widetilde{R}_m) S_\pi \right\| \leq \sqrt{c_\mathcal{H}}\left(2 + \tfrac{(1+\sqrt{2})\pi^2\kappa^2}{18e^2\cos^2(\theta)}\right)\Delta t + \sqrt{c_\mathcal{H}}\,\frac{e^{-\mu t_\ell}}{\mu}, \qquad (42)$$

which becomes arbitrarily small for $\Delta t \to 0$ and $t_\ell \to \infty$.

On the other hand for the rank reduction case,

$$\left\| S_\pi(G_{\mu,\gamma}^r - \widetilde{G}_{m,\gamma}^r) \right\| = \left\| C^{1/2}C_\gamma^{-1}S_\pi^*(R_\mu S_\pi B_r - \widetilde{R}_m S_\pi \widetilde{P}_r) \right\| \leq \left\| (R_\mu - \widetilde{R}_m)S_\pi \widetilde{P}_r \right\| + \left\| R_\mu S_\pi(P_r - \widetilde{P}_r) \right\| \qquad (43)$$

that is, $\left\| S_\pi(G_{\mu,\gamma}^r - \widetilde{G}_{m,\gamma}^r) \right\| \leq \sqrt{c_\mathcal{H}}\left(2 + \tfrac{(1+\sqrt{2})\pi^2\kappa^2}{18e^2\cos^2(\theta)}\right)\Delta t + \sqrt{c_\mathcal{H}}\,\frac{e^{-\mu t_\ell}}{\mu} + \frac{\left\| P_r - \widetilde{P}_r \right\|}{\mu}$, which can be controlled by bounding $\left\| P_r - \widetilde{P}_r \right\|$ via $\left\| B^*B - \widetilde{B}^*\widetilde{B} \right\|$ using Theorem A.3. Since in the analysis of the variance term similar construction should be performed for the orthogonal projector onto the leading singular subspace of the empirical operator $\widehat{B} = \widehat{C}_\gamma^{-1/2}\widehat{H}_m$, in the following we jointly bound last to terms in (30).

## C.3. Estimator's variance

In this section, with the exception of App. C.7, we assume the single trajectory setting with uniform time-discretization, and analyze the last term in (30) developing concentration inequalities in Hilbert spaces for non-iid variables based on the notion of beta-mixing and the method of blocks introduced in (Yu, 1994). Before we begin, let us summarize the reminder of the terms, assuming the case of universal kernel together with **(BK)**, **(SD)** and **(RC)**

$$\mathcal{E}(\widehat{G}^r_{m,\gamma}) \leq (c_\alpha/\mu)\gamma^{\alpha/2} + \sigma_{r+1}(B) + \left\| S_\pi(G^r_{\mu,\gamma} - \widehat{G}^r_{m,\gamma}) \right\|. \tag{44}$$

### C.3.1. REMINDER ON ERGODIC AND EXPONENTIALLY MIXING MARKOV PROCESSES

We recall some fundamental results on ergodic, exponentially mixing Markov Processes.

**Definition C.5** (Strict stationarity)**.** A Markov process $\mathbf{X} = (X_i)_{i \in \mathbb{N}}$ with values in $\mathcal{X}$ is strictly stationary if for every $m, l \in \mathbb{N}$ the marginal distribution of $(X_{1+l}, \dots, X_{m+l})$ is the same as $(X_1, \dots, X_m)$.

For a set $I \subseteq \mathbb{N}$ and a strictly stationary process $\mathbf{X} = (X_i)_{i \in \mathbb{N}}$ we let $\Sigma_I$ for the $\sigma$-algebra generated by $\{X_i\}_{i \in I}$ and $\mu_I$ for the joint distribution of $\{X_i\}_{i \in I}$. Notice that $\mu_{I+i} = \mu_I$. In this notation $\pi = \mu_{\{1\}}$ and $\rho_\tau = \mu_{\{1,1+\tau\}}$. We can now introduce the $\beta$-mixing coefficients

$$\beta_{\mathbf{X}}(\tau) = \sup_{B \in \Sigma \otimes \Sigma} \left| \mu_{\{1,1+\tau\}}(B) - \mu_{\{1\}} \times \mu_{\{1+\tau\}}(B) \right|$$

which by the Markov property is equivalent to

$$\beta_{\mathbf{X}}(\tau) = \sup_{B \in \Sigma^I \otimes \Sigma^J} \left| \mu_{I \cup J}(B) - \mu_I \times \mu_J(B) \right|,$$

where $I, J \subset \mathbb{N}$ with $j > i + \tau$ for all $i \in I$ and $j \in J$. The latter is the definition of the mixing coefficients for general strictly stationary processes.

It is well-known that a stationary, geometrically ergodic Markov process is $\beta$-mixing with $\beta_{\mathbf{X}}(\tau) \to 0$ exponentially fast as $\bar{\tau} \to \infty$, that is $\beta_{\mathbf{X}}(\bar{\tau}) \leq \eta e^{-\gamma\bar{\tau}}$, for some $\eta, \gamma \in (0, \infty)$ for all $\bar{\tau} \in \mathbb{N}$. See e.g. (Bradley, 2007, vol. 2 Theorem 21.19 pp 325).

The following result exploits a block-process argument to extend concentration bounds from the iid setting to strictly stationary $\beta$-mixing Markov processes.

**Lemma C.6** (cf. (Kostic et al., 2022), Lemma 1)**.** *Let* $\mathbf{X}$ *be strictly stationary with values in a normed space* $(\mathcal{X}, \|\cdot\|)$, *and let* $\bar{\tau}, m \in \mathbb{N}$ *such that* $m$ *is the largest integer satisfyong* $n \geq 2m\bar{\tau}$. *Moreover, let* $Z_1, \dots, Z_{m+1}$ *be* $m+1$ *independent copies of* $Z = \sum_{i=1}^\tau X_i$. *Then for* $s > 0$,

$$\mathbb{P}\left\{ \left\| \sum_{i=1}^n X_i \right\| > s \right\} \leq \mathbb{P}\left\{ \left\| \sum_{j=1}^m Z_j \right\| > \frac{s}{2} \right\} + \mathbb{P}\left\{ \left\| \sum_{j=1}^m Z_j + \sum_{i=1}^{k'} X'_i \right\| > \frac{s}{2} \right\} + 2m\beta_{\mathbf{X}}(\bar{\tau} - 1).$$

*where we define* $k \in [\![0, 2\bar{\tau}-1]\!]$ *such that* $k = n - 2m\bar{\tau} = l\bar{\tau} + k'$ *with* $l \in \{0, 1\}$ *and* $k' \in [\![0, \bar{\tau}-1]\!]$, *and* $X'_i = X_{(2m+l)\bar{\tau}+i}$ *for all* $i \in [\![1, k']\!]$.

*Proof.* Recall first the definition of the blocked variables

$$Y_j = \sum_{i=2(j-1)\bar{\tau}+1}^{(2j-1)\bar{\tau}} X_i \quad \text{and} \quad Y'_j = \sum_{i=(2j-1)\bar{\tau}+1}^{2j\bar{\tau}} X_i. \tag{45}$$

Then, we have,

$$\left\| \sum_{i=1}^n X_i \right\| = \left\| \sum_{j=1}^m Y_j + \sum_{j=1}^m Y'_j + \sum_{i=1}^k X_{2m\bar{\tau}+i} \right\| \leq \left\| \sum_{j=1}^m Y_j + \sum_{i=1}^{l\bar{\tau}} X_{2m\bar{\tau}+i} \right\| + \left\| \sum_{j=1}^m Y'_j + \sum_{i=1}^{k'} X_{2m\bar{\tau}+l\bar{\tau}+i} \right\|$$

where we let $k = n - 2m\bar{\tau} = l\bar{\tau} + k'$ with $l \in \{0, 1\}$ and $k' \in [\![0, \bar{\tau} - 1]\!]$, and we use the convention $\sum_{i=1}^0 \cdot = 0$. Thus, for $s > 0$,

$$
\mathbb{P}\Big\{\Big\|\sum_{i=1}^n X_i\Big\| > s\Big\} \le \mathbb{P}\Big\{\Big\|\sum_{j=1}^{m+l} Z_j\Big\| > \frac{s}{2}\Big\} + (m - 1 + l)\beta_{\mathbf{X}}(\bar{\tau} - 1) + \mathbb{P}\Big\{\Big\|\sum_{j=1}^m Y_j' + \sum_{i=1}^{k'} X_i'\Big\| > \frac{s}{2}\Big\} + m\beta_{\mathbf{X}}(\bar{\tau} - 1)
$$

$$
\le \mathbb{P}\Big\{\Big\|\sum_{j=1}^{m+l} Z_j\Big\| > \frac{s}{2}\Big\} + \mathbb{P}\Big\{\Big\|\sum_{j=1}^m Z_j + \sum_{i=1}^{k'} X_i'\Big\| > \frac{s}{2}\Big\} + 2m\beta_{\mathbf{X}}(\bar{\tau} - 1),
$$

where we have defined $X_i' = X_{(2m+l)\bar{\tau}+i}$ for all $i \in [\![1, k']\!]$ and we have used (Kostic et al., 2022), Lemma 3 to get the first inequality. $\qquad\square$

### C.3.2. CONCENTRATION FOR TRANSFER OPERATORS

We consider the transfer operators $T_{j\Delta t} = S_\pi^* e^{j\Delta t\, L} S_\pi$ for any $j \in \{0, \dots, \ell\}$ and their empirical versions

$$
\widehat{T}_{j\Delta t} = \frac{1}{n-j} \sum_{i=0}^{n-j-1} \phi(X_{i\Delta t}) \otimes \phi(X_{(i+j)\Delta t}).
$$

We prove several concentration results on operators $C$ and $T_{j\Delta t}$, $j \in [l]$ and $\widetilde{H}_m$.

**Proposition C.7.** *Assume that $n - j \ge 2m\bar{\tau}$. Let $\delta > (m-1)\beta_{\mathbf{X}\cdot\Delta t}(\bar{\tau} - 1)$. Let Assumption (KE) be satisfied. With probability at least $1 - \delta$ in the draw $X_0 \sim \pi, X_{i\Delta t} \sim p(X_{(i-1)\Delta t}, \cdot), i \in [n]$,*

$$
\Big\|C_\gamma^{-1/2}(\widehat{T}_{j\Delta t} - T_{j\Delta t})\Big\| \le 8\sqrt{2c_{\mathcal{H}}} \ln\left(\frac{4}{\delta - 2(m-1)\beta_{\mathbf{X}\cdot\Delta t}(\bar{\tau} - 1)}\right) \sqrt{\frac{c_\beta^\beta}{(1-\beta)\gamma^\beta m} + \frac{c_\tau}{m^2\gamma^\tau}}. \tag{46}
$$

*Proof of Proposition* (C.7). Set $\xi(x) := C_\gamma^{-1/2}\phi(x)$. Assume first for simplicity that $n - j = 2m\bar{\tau}$. We have

$$
C_\gamma^{-1/2}(\widehat{T}_{j\Delta t} - T_{j\Delta t}) := \frac{1}{n-j} \sum_{i=0}^{n-j-1} \xi(X_{i\Delta t}) \otimes \phi(X_{(i+j)\Delta t}) - \mathbb{E}\big[\xi(X_{i\Delta t}) \otimes \phi(X_{(i+j)\Delta t})\big].
$$

Set $A_i := \xi(X_{(i-1)\Delta t}) \otimes \phi(X_{(i-1+j)\Delta t})$. Let $Z_1, \dots, Z_m$ be $m$ iid copies of $Z := \frac{1}{\bar{\tau}}\sum_{i=1}^{\bar{\tau}} A_i$. For any $k \ge 2$, by triangular inequality and convexity of $x \mapsto x^k$, we have that $\|Z\|_{\mathrm{HS}}^k \le \big(\frac{1}{\bar{\tau}}\sum_{i=1}^{\bar{\tau}}\|A_i\|_{\mathrm{HS}}\big)^k \le \frac{1}{\bar{\tau}}\sum_{i=1}^{\bar{\tau}}\|A_i\|_{\mathrm{HS}}^k$. Observe next that

$$
\mathbb{E}[\|A_i\|_{\mathrm{HS}}^k] = \mathbb{E}\big[\big\|\xi(X_{(i-1)\Delta t})\big\|^k \big\|\phi(X_{(i-1+j)\Delta t})\big\|^k\big] \le \|\xi\|_\infty^{k-2}\|\phi\|_\infty^k\, \mathbb{E}\big[\big\|\xi(X_{(i-1)\Delta t})\big\|^2\big]
$$

$$
= \|\xi\|_\infty^{k-2}\|\phi\|_\infty^k\, \mathrm{tr}(C_\gamma^{-1}C) \le \frac{1}{2}k!\left(\gamma^{-\tau/2}\sqrt{c_\tau\, c_{\mathcal{H}}}\right)^{k-2}\left(\sqrt{c_{\mathcal{H}}\, \mathrm{tr}(C_\gamma^{-1}C)}\right)^2.
$$

In view of (Fischer & Steinwart, 2020, Lemma 11), we have under Condition (SD) when $\beta < 1$ that

$$
\mathrm{tr}(C_\gamma^{-1}C) \le \frac{c_\beta^\beta}{1-\beta}\gamma^{-\beta}, \quad \forall\gamma > 0.
$$

Consequently

$$
\mathbb{E}[\|Z\|_{\mathrm{HS}}^k] \le \frac{1}{\bar{\tau}}\sum_{i=1}^{\bar{\tau}}\mathbb{E}[\|A_i\|_{\mathrm{HS}}^k] \le \frac{1}{2}k!\left(\gamma^{-\tau/2}\sqrt{c_\tau\, c_{\mathcal{H}}}\right)^{k-2}\left(\sqrt{c_{\mathcal{H}}\frac{c_\beta^\beta}{1-\beta}\gamma^{-\beta}}\right)^2. \tag{47}
$$

We apply (Kostic et al., 2023, Proposition 9) to get with probability at least $1 - \delta$,

$$\left\| \frac{1}{m} \sum_{i=1}^{m} Z_i \right\| \leq 4\sqrt{2\,c_{\mathcal{H}}} \ln\left(\frac{2}{\delta}\right) \sqrt{\frac{c_\beta^\beta}{(1-\beta)\gamma^\beta m} + \frac{c_\tau}{m^2 \gamma^\tau}}.$$

Replacing $\delta$ by $\frac{\delta}{2} - (m-1)\beta_{\mathbf{X}_{\cdot\Delta t}}(\bar\tau - 1)$ and an union bound combining the last display with Lemma C.6 gives the result.

$\square$

We assume now that the Markov process $\mathbf{X}_{\cdot\Delta t} = (X_{i\Delta t})_{i\in\mathbb{N}}$ is ergodic, exponentially mixing, that is, there exists $c_{\mathrm{mix}} \in (0, \infty)$, such that for all $\tau \in \mathbb{N}$,

$$\beta_{\mathbf{X}_{\cdot\Delta t}}(\tau) \leq c_{\mathrm{mix}}\, e^{-\Delta t\, w_\star\, \bar\tau}, \quad \forall \bar\tau \geq 1, \tag{48}$$

where we recall that $w_\star = -\lambda_2(L + L^*)/2 > 0$.

For any $n \geq 1$ and $\delta \in (0, 1)$, define the rate

$$\bar\varepsilon_n(\delta) := c\left(\frac{\ln^2\left(\frac{n}{\delta}\right)}{\Delta t\, w_\star\, n} + \frac{\ln\left(\frac{n}{\delta}\right)}{\sqrt{\Delta t\, w_\star\, n}}\right), \tag{49}$$

where $c = c(c_{\mathrm{mix}}) > 0$ is a large enough numerical constant.

**Proposition C.8.** *Let $\delta \in (0, 1)$. Assume that $(n - j)\Delta t\, w_\star \geq 4\ln\left(\frac{2e^2 c_{\mathrm{mix}}(n-j)}{\delta}\right)$. Let Condition 48 be satisfied. With probability at least $1 - \delta$ in the draw $X_0 \sim \pi, X_{i\Delta t} \sim p(X_{(i-1)\Delta t}, \cdot), i \in [n]$,*

$$\left\| \widehat{T}_{j\Delta t} - T_{j\Delta t} \right\| \leq \bar\varepsilon_{n-j}(\delta). \tag{50}$$

*Proof.* We start from the following result (Kostic et al., 2022, Proposition 3) assuming $n - j \geq 2m\bar\tau$

$$\mathbb{P}\left(\left\| \widehat{T}_{j\Delta t} - T_{j\Delta t} \right\| \leq \frac{48}{m} \ln\left(\frac{4m\bar\tau}{\delta - (m-1)\beta_{\mathbf{X}_{\cdot\Delta t}}(\bar\tau - 1)}\right) + 12\sqrt{\frac{2\|C\|}{m}\ln\frac{4m}{\delta - (m-1)\beta_{\mathbf{X}_{\cdot\Delta t}}(\bar\tau - 1)}}\right) \geq 1 - \delta. \tag{51}$$

For any $j \in [l]$, we take integers $m_j, \bar\tau_j \geq 1$ such that $n - j \geq 2m_j\bar\tau_j$ and $\delta \geq 2(m_j - 1)\beta_{\mathbf{X}_{\cdot\Delta t}}(\bar\tau_j - 1)$. Hence we can pick

$$m_j := \left\lfloor \frac{(n-j)\Delta t\, w_\star}{2\ln\left(\frac{2e^2 c_{\mathrm{mix}}(n-j)}{\delta}\right)} \right\rfloor \quad \text{and} \quad \bar\tau_j := \left\lfloor \frac{1}{\Delta t\, w_\star} \ln\left(\frac{2e^2 c_{\mathrm{mix}}(m_j - 1)}{\delta}\right) \right\rfloor. \tag{52}$$

Assuming that $n$ is large enough such that $(n-j)\Delta t\, w_\star \geq 4\ln\left(\frac{2e^2 c_{\mathrm{mix}}(n-j)}{\delta}\right)$, we get

$$m_j \geq \frac{1}{4}(n-j)\Delta t\, w_\star / \ln\left(\frac{4e^2 c_{\mathrm{mix}}(n-j)}{\delta}\right).$$

We also have in view of (48) that $(m_j - 1)\beta_{\mathbf{X}_{\cdot\Delta t}}(\bar\tau - 1) \leq \delta/2$. Replacing these quantities in (51), we get the result.

$\square$

Define for any $n \geq 1$

$$\bar\varepsilon_n^{(1)}(\gamma, \delta) := c\left(\frac{c_\tau\, \ln(n\delta^{-1})}{n\, \Delta t\, w_\star\, \gamma^\tau}\mathcal{L}^1(\gamma, \delta) + \sqrt{\frac{c_\tau\, \ln(n\delta^{-1})}{n\, \Delta t\, w_\star\, \gamma^\tau}\mathcal{L}^1(\gamma, \delta)}\right), \tag{53}$$

and

$$\mathcal{L}^1(\gamma, \delta) := \ln\left(\frac{4}{\delta}\right) + \ln\left(\frac{\mathrm{tr}(C_\gamma^{-1}C)}{\|C_\gamma^{-1}C\|}\right),$$

where the numerical constant $c > 0$ only depends on $c_{\mathrm{mix}}$.

**Proposition C.9.** *Let Assumption* (KE) *and Condition* (48) *be satisfied. Then, with probability at least* $1 - \delta$ *in the draw* $X_0 \sim \pi$, $X_{i\Delta t} \sim p(X_{(i-1)\Delta t}, \cdot)$, $i \in [n]$,

$$\left\| C_\gamma^{-1/2}(\widehat{C} - C)C_\gamma^{-1/2} \right\| \leq \bar{\varepsilon}_n^{(1)}(\gamma, \delta).$$

*In addition, for* $\delta \in (0, 1)$ *and* $n$ *large enough such that* $\bar{\varepsilon}_n^{(1)}(\gamma, \delta) \in (0, 1)$,

$$\left\| C_\gamma^{1/2}\widehat{C}_\gamma^{-1}C_\gamma^{1/2} \right\| \leq \frac{1}{1 - \bar{\varepsilon}_n^{(1)}(\gamma, \delta)}.$$

*Finally, for any* $j \in [l]$ *such that* $n - j \geq 2m\bar{\tau}$, *we have with probability at least* $1 - \delta$ *in the draw* $X_0 \sim \pi$, $X_{i\Delta t} \sim p(X_{(i-1)\Delta t}, \cdot)$, $i \in [n]$,

$$\left\| C_\gamma^{-1/2}(\widehat{T}_{j\Delta t} - T_{j\Delta t})C_\gamma^{-1/2} \right\| \leq \bar{\varepsilon}_{n-j}^{(1)}(\gamma, \delta).$$

*Proof of Proposition C.9.* Define for any $m \geq 1$ and $\delta \in (0, 1)$

$$\varepsilon_m^{(1)}(\gamma, \delta) := \frac{4c_\tau}{3m\gamma^\tau} \mathcal{L}^1(\gamma, \delta) + \sqrt{\frac{2\,c_\tau}{m\,\gamma^\tau} \mathcal{L}^1(\gamma, \delta)}, \tag{54}$$

where

$$\mathcal{L}^1(\gamma, \delta) := \ln\left(\frac{4}{\delta}\right) + \ln\left(\frac{\operatorname{tr}(C_\gamma^{-1}C)}{\|C_\gamma^{-1}C\|}\right).$$

Combining Lemma C.6 with (Kostic et al., 2023, Proposition 13), we obtain with probability at least $1 - \delta$,

$$\left\| C_\gamma^{-1/2}(C - \widehat{C})C_\gamma^{-1/2} \right\| \leq 2\varepsilon_m^{(1)}\big(\gamma, \delta/2 - (m-1)\,\beta_{\mathbf{X}_{\cdot\Delta t}}(\bar{\tau} - 1)\big). \tag{55}$$

Exploiting again the $\beta$-mixing assumption in (48), we take $\bar{\tau}$ as the smallest integer such that

$$\bar{\tau} \geq 1 + \frac{1}{\Delta t\, w_\star} \ln\left(\frac{4c_{\mathrm{mix}}\, n}{\delta}\right). \tag{56}$$

With this choice of $\bar{\tau}$ and picking $m$ as the largest integer such that $n \geq 2m\bar{\tau}$, then we get

$$2\varepsilon_m^{(1)}\big(\gamma, \delta/2 - (m-1)\,\beta_{\mathbf{X}_{\cdot\Delta t}}(\bar{\tau} - 1)\big) \leq 2\varepsilon_m^{(1)}(\gamma, \delta/4) \leq \bar{\varepsilon}_n^{(1)}(\gamma, \delta),$$

provided that the numerical constant $c = c(c_{\mathrm{mix}}) > 0$ is large enough.

Finally, for $\delta \in (0, 1)$ and $n$ large enough such that $\bar{\varepsilon}_n^{(1)}(\gamma, \delta) \in (0, 1)$ and observing that

$$1 - \left\| I - C_\gamma^{-1/2}\widehat{C}_\gamma C_\gamma^{-1/2} \right\| = \left\| C_\gamma^{-1/2}(C - \widehat{C})C_\gamma^{-1/2} \right\|,$$

we get

$$\left\| C_\gamma^{1/2}\widehat{C}_\gamma^{-1}C_\gamma^{1/2} \right\| = \left\| (C_\gamma^{-1/2}\widehat{C}_\gamma C_\gamma^{-1/2})^{-1} \right\| \leq \frac{1}{1 - \left\| I - C_\gamma^{-1/2}\widehat{C}_\gamma C_\gamma^{-1/2} \right\|} \leq \frac{1}{1 - \bar{\varepsilon}_n^{(1)}(\gamma, \delta)}.$$

The proof for the last result on $\left\| C_\gamma^{-1/2}(\widehat{T}_{j\Delta t} - T_{j\Delta t})C_\gamma^{-1/2} \right\|$ follows from the same argument. □

**Proposition C.10.** *Let the assumptions of Proposition C.7 and Condition* (48) *be satisfied. Assume in addition that* $\mu\,\Delta t \in (0, 1)$ *and* $w_\star \Delta t \leq 1$. *Then with probability at least* $1 - \delta$ *in the draw* $X_0 \sim \pi$, $X_{i\Delta t} \sim p(X_{(i-1)\Delta t}, \cdot)$, $i \in [n]$,

$$\left\| C_\gamma^{-1/2}(\widetilde{H}_m - \widehat{H}_m) \right\| \leq \frac{\bar{\varepsilon}_{n,l}^{(2)}(\gamma, \delta)}{\mu}, \tag{57}$$

*where*

$$\bar{\varepsilon}_{n,l}^{(2)}(\gamma, \delta) := c\left(\frac{\ln^2\left(\frac{n(l+1)}{\delta}\right)}{\Delta t\, w_\star \gamma^{\tau/2}} \frac{1}{n-l} + \frac{\ln^{3/2}\left(\frac{n(l+1)}{\delta}\right)}{\sqrt{\Delta t\, w_\star}} \sqrt{\frac{1}{\gamma^\beta(n-l)}}\right), \tag{58}$$

*where* $c = c(c_{\mathrm{mix}}, c_{\mathcal{H}}, c_\tau, c_\beta, \beta) > 0$ *is a large enough numerical constant.*

*Proof of Proposition C.10.* For any $j \in [l]$, we take integers $m_j, \bar{\tau}_j \geq 1$ such that $n - j \geq 2m_j\bar{\tau}_j$ and $\delta \geq 4(m_j - 1)\beta_{\mathbf{X}_{\cdot \Delta t}}(\bar{\tau}_j - 1)$. We use the same choices as in (52).

Replacing these quantities in (46), we get with probability at least $1 - \delta$,

$$\left\| C_\gamma^{-1/2}(\widehat{T}_{j\Delta t} - T_{j\Delta t}) \right\| \leq 8\sqrt{2c_{\mathcal{H}}} \ln\left(\frac{8}{\delta}\right) \left( \sqrt{\frac{2c_\beta^\beta}{(1-\beta)\gamma^\beta(n-j)\Delta t\, w_\star}} \ln\left(\frac{2e^2 c_{\text{mix}}(n-j)}{\delta}\right) \right.$$
$$\left. + \frac{\sqrt{2c_\tau}}{(n-j)\Delta t\, w_\star\, \gamma^{\tau/2}} \ln\left(\frac{2e^2 c_{\text{mix}}(n-j)}{\delta}\right) \right) =: \varepsilon_{n,j}^{(2)}(\delta). \tag{59}$$

Next, by definition of $\widetilde{H}_m, \widehat{H}_m$ and an elementary union bound, we get with probability at least $1 - \delta$,

$$\left\| C_\gamma^{-1/2}(\widetilde{H}_m - \widehat{H}_m) \right\| \leq \Delta t \sum_{j=0}^l \omega_j \left\| C_\gamma^{-1/2}(\widehat{T}_{j\Delta t} - T_{j\Delta t}) \right\| e^{-\mu j \Delta t}$$

$$\leq \Delta t \sum_{j=0}^l \omega_j\, \varepsilon_{n,j}^{(2)}(\delta/(l+1))\, e^{-\mu j \Delta t}$$

$$\leq C \left( \sqrt{\Delta t}\, \frac{\ln^{3/2}\left(\frac{n(l+1)}{\delta}\right)}{\sqrt{w_\star\, \gamma^\beta}} \sum_{j=0}^l \frac{e^{-\mu j \Delta t}}{\sqrt{n-j}} + \frac{\ln^2\left(\frac{n(l+1)}{\delta}\right)}{w_\star \gamma^{\tau/2}} \sum_{j=0}^l \frac{e^{-\mu j \Delta t}}{n-j} \right),$$

for some large enough numerical constant $C = C(c_{\mathcal{H}}, c_\tau, c_{\text{mix}}, c_\beta, \beta) > 0$ that can depend only on $c_{\mathcal{H}}, c_\tau, c_{\text{mix}}, c_\beta, \beta$.

We apply the well-known Abel transform formula

$$\sum_{k=0}^l f_k g_k = f_l \sum_{k=0}^l g_k - \sum_{j=0}^{l-1} (f_{j+1} - f_j) \sum_{k=0}^j g_k,$$

to series $\sum_{j=0}^l \frac{e^{-\mu j \Delta t}}{(n-j)^\alpha}$, with $\alpha \in \{1/2, 1\}$. Since we assumed that $\mu\Delta t \in (0,1)$, elementary computations give

$$\sum_{j=0}^l \frac{e^{-\mu j \Delta t}}{(n-j)^\alpha} \leq \frac{1}{(n-l)^\alpha} \frac{1}{1 - e^{-\mu\Delta t}} - \sum_{j=0}^{l-1} \left( \frac{1}{(n-j-1)^\alpha} - \frac{1}{(n-j)^\alpha} \right) \sum_{k=0}^j e^{-\mu k \Delta t}$$

$$\leq \frac{1}{(n-l)^\alpha} \frac{1}{1 - e^{-\mu\Delta t}} \leq \frac{2}{\mu\Delta t\, (n-l)^\alpha},$$

where we have used the inequality $e^{-x} \leq 1 - x + x^2/2$ true for any $x \in (0,1)$ in the last line. $\qquad \square$

**Proposition C.11.** *Let Assumption (KE) and Condition (48) be satisfied. Fix $\delta \in (0,1)$ and assume in addition that $n$ is large enough such that $n\Delta t\, w_\star \geq 4\ln\left(\frac{2e^2 c_{\text{mix}} n}{\delta}\right)$. Then, with probability at least $1 - \delta$ in the draw $X_0 \sim \pi$, $X_{i\Delta t} \sim p(X_{(i-1)\Delta t}, \cdot), i \in [n]$,*

$$\left\| C_\gamma^{-1/2}(\widehat{C} - C) \right\| \leq \bar{\varepsilon}_{n,0}^{(2)}(\gamma, \delta) \tag{60}$$

*Proof of Proposition C.11.* Similarly to the proof of Proposition C.7, we obtain, assuming that $n \geq 2m\bar{\tau}$, with probability at least $1 - \delta$ in the draw $X_0 \sim \pi, X_{i\Delta t} \sim p(X_{(i-1)\Delta t}, \cdot), i \in [n]$,

$$\left\| C_\gamma^{-1/2}(\widehat{C} - C) \right\| \leq 8\sqrt{2c_{\mathcal{H}}} \ln\left(\frac{4}{\delta - 2(m-1)\beta_{\mathbf{X}_{\cdot \Delta t}}(\bar{\tau} - 1)}\right) \sqrt{\frac{c_\beta^\beta}{(1-\beta)\gamma^\beta m} + \frac{c_\tau}{m^2 \gamma^\tau}}. \tag{61}$$

We pick integers $m, \bar{\tau} \geq 1$ such that $n \geq 2m\bar{\tau}$ and $\delta \geq 2(m-1)\beta_{\mathbf{X}_{\cdot \Delta t}}(\bar{\tau} - 1)$. Hence we pick

$$m := \left\lfloor \frac{n\Delta t\, w_\star}{2\ln\left(\frac{2e^2 c_{\text{mix}} n}{\delta}\right)} \right\rfloor \quad \text{and} \quad \bar{\tau} := \left\lfloor \frac{1}{\Delta t\, w_\star} \ln\left(\frac{2e^2 c_{\text{mix}}(m-1)}{\delta}\right) \right\rfloor, \tag{62}$$

Assuming that $n$ is large enough such that $n\,\Delta t\,w_\star \geq 4\ln\left(\frac{2e^2 c_{\mathrm{mix}} n}{\delta}\right)$, we get $m \geq \frac{1}{4}n\,\Delta t\,w_\star/\ln\left(\frac{4e^2 c_{\mathrm{mix}} n}{\delta}\right)$. We also have in view of (48) that $(m-1))\beta_{\mathbf{X}_{\cdot\Delta t}}(\bar{\tau}-1) \leq \delta/2$. Replacing these quantities in (61), we get the result. □

### C.3.3. VARIANCE OF SINGULAR VALUES

We introduce the approximated and empirical KRR models as $\widetilde{G}_{m,\gamma} = C_\gamma^{-1}\widetilde{H}_m$ and $\widehat{G}_{m,\gamma} = \widehat{C}_\gamma^{-1}\widehat{H}_m$, respectively. We recall that $\widetilde{G}_{m,\gamma}^r = C_\gamma^{-1/2}[\![C_\gamma^{-1/2}\widetilde{H}_m]\!]_r$ and $\widehat{G}_{m,\gamma}^r = \widehat{C}_\gamma^{-1/2}[\![\widehat{C}_\gamma^{-1/2}\widehat{H}_m]\!]_r$. We also introduce $B := C_\gamma^{-1/2}H_\mu$ and $\widehat{B} := \widehat{C}_\gamma^{-1/2}\widehat{H}_m$, let denote $P_r$ and $\widehat{P}_r$ denote the orthogonal projector onto the subspace of leading $r$ right singular vectors of $B$ and $\widehat{B}$, respectively. Then we have $[\![B]\!]_r = BP_r$ and $[\![\widehat{B}]\!]_r = \widehat{B}\widehat{P}_r$, and, hence $\widetilde{G}_{m,\gamma}^r = \widetilde{G}_{m,\gamma}\widetilde{P}_r$ and $\widehat{G}_{m,\gamma}^r = \widehat{G}_{m,\gamma}\widehat{P}_r$. Let us recall that $G_{\mu,\gamma}^r = C_\gamma^{-1/2}[\![C_\gamma^{-1/2}H_\mu]\!]_r$.

**Proposition C.12.** *Let* **(RC)**, **(SD)** *and* **(KE)** *hold for some* $\alpha \in [1,2]$, $\beta \in (0,1]$ *and* $\tau \in [\beta,1]$. *Let* $B := C_\gamma^{-1/2}H_\mu$ *and* $\widehat{B} := \widehat{C}_\gamma^{-1/2}\widehat{H}_m$. *Given* $\delta > 0$ *if* $\bar{\varepsilon}_n^{(1)}(\gamma,\delta/3) < 1/2$, *then with probability at least* $1-\delta$ *in the draw* $X_0 \sim \pi$, $X_{i\Delta t} \sim p(X_{(i-1)\Delta t},\cdot)$, $i \in [n]$,

$$\left\|\widehat{B}^*\widehat{B} - B^*B\right\| \leq \frac{2}{\mu^2}\left(\sqrt{c_{\mathcal{H}}} + \bar{\varepsilon}_{n,\ell}^{(3)}(\gamma,\delta/3) + \sqrt{c_{\mathcal{H}}}\,\mu\,s_{\mu,\ell}(\Delta t)\right)\left(\bar{\varepsilon}_n^{(3)}(\gamma,\delta/3) + \sqrt{c_{\mathcal{H}}}\,\mu\,s_{\mu,\ell}(\Delta t)\right) \tag{63}$$

*where* $\bar{\varepsilon}_{n,\ell}^{(3)}(\gamma,\delta/3) = \bar{\varepsilon}_{n,l}^{(2)}(\gamma,\delta/3) + \bar{\varepsilon}_{n,0}^{(2)}(\gamma,\delta/3)c_\alpha c_{\mathcal{H}}^{(\alpha-1)/2}$, $\bar{\varepsilon}_{n,l}^{(2)}(\gamma,\delta)$ *is defined in* (58), *and* $s_{\mu,\ell}(\Delta t)$ *in* (41). *Consequently, for every* $i \in [n]$, *when* $\bar{\varepsilon}_n^{(3)}(\gamma,\delta/3)/\mu + \sqrt{c_{\mathcal{H}}}\,s_{\mu,\ell}(\Delta t) \leq \sqrt{c_{\mathcal{H}}}$ *it holds that*

$$|\sigma_i^2(\widehat{B}) - \sigma_i^2(B)| \leq \frac{2\sqrt{c_{\mathcal{H}}}}{\mu}\left(\bar{\varepsilon}_n^{(3)}(\gamma,\delta/3) + \sqrt{c_{\mathcal{H}}}\,\mu\,s_{\mu,\ell}(\Delta t)\right). \tag{64}$$

*Proof.* We start from the Weyl's inequalities for the square of singular values

$$|\sigma_i^2(\widehat{B}) - \sigma_i^2(B)| \leq \left\|\widehat{B}^*\widehat{B} - B^*B\right\|, \ i \in [n].$$

But, since,

$$\widehat{B}^*\widehat{B} - B^*B = \widehat{H}_m^*\widehat{C}_\gamma^{-1}\widehat{H}_m - H_\mu^*C_\gamma^{-1}H_\mu = (\widehat{H}_m - H_\mu)^*\widehat{C}_\gamma^{-1}\widehat{H}_m + H_\mu^*C_\gamma^{-1}(\widehat{H}_m - H_\mu) + H_\mu^*(\widehat{C}_\gamma^{-1} - C_\gamma^{-1})\widehat{H}_m,$$

denoting $M = C_\gamma^{-1/2}(\widehat{H}_m - H_\mu)$, $N = C_\gamma^{-1/2}(\widehat{C} - C)$ and $R = C_\gamma^{1/2}(\widehat{G}_{m,\gamma} - G_{\mu,\gamma})$, we have

$$\begin{aligned}
\widehat{B}^*\widehat{B} - B^*B &= M^*C_\gamma^{1/2}\widehat{G}_{m,\gamma} + B^*M - B^*N\widehat{G}_{m,\gamma} = B^*M + (M^*C_\gamma^{1/2} - B^*N)(\widehat{G}_{m,\gamma} \pm G_{\mu,\gamma})\\
&= B^*M + M^*B - B^*N\widetilde{G}_{m,\gamma} + (M^* - B^*NC_\gamma^{-1/2})R\\
&= (G_{\mu,\gamma})^*(\widehat{H}_m - H_\mu) + (\widehat{H}_m - H_\mu)G_{\mu,\gamma} - (G_{\mu,\gamma})^*(\widehat{C} - C)G_{\mu,\gamma} + (M^* + (G_{\mu,\gamma})^*N^*)R.
\end{aligned}$$

Note next by definition of $G_{\mu,\gamma}$ and $\widehat{G}_{m,\gamma}$, we have

$$\widehat{G}_{m,\gamma} - G_{\mu,\gamma} = C_\gamma^{-1/2}\left[C_\gamma^{-1/2}(\widehat{H}_m - H_\mu) - C_\gamma^{-1/2}(\widehat{C} - C)C_\gamma^{-1/2}[C_\gamma^{1/2}\widehat{C}_\gamma^{-1}C_\gamma^{1/2}][C_\gamma^{-1/2}\widehat{H}_m]\right]. \tag{65}$$

Therefore, due to (65), $R = C_\gamma^{1/2}\widehat{C}_\gamma^{-1}C_\gamma^{1/2}(M - NG_{\mu,\gamma})$, we conclude

$$\begin{aligned}
\widehat{B}^*\widehat{B} - B^*B &= (G_{\mu,\gamma})^*(\widehat{H}_m - H_\mu) + (\widehat{H}_m - H_\mu)^*G_{\mu,\gamma} - (G_{\mu,\gamma})^*(\widehat{C} - C)G_{\mu,\gamma}\\
&\quad + (M - NG_{\mu,\gamma})^*C_\gamma^{1/2}\widehat{C}_\gamma^{-1}C_\gamma^{1/2}(M - NG_{\mu,\gamma}) \tag{66}
\end{aligned}$$

Hence,

$$\begin{aligned}
\left\|\widehat{B}^*\widehat{B} - B^*B\right\| &\leq 2\|B\|\|M\| + \|B\|\|N\|\|G_{\mu,\gamma}\| + \left\|C_\gamma^{1/2}\widehat{C}_\gamma^{-1}C_\gamma^{1/2}\right\|[\|M\| + \|N\|\|G_{\mu,\gamma}\|]^2\\
&\leq \left[2\|B\| + \left\|C_\gamma^{1/2}\widehat{C}_\gamma^{-1}C_\gamma^{1/2}\right\|(\|M\| + \|N\|\|G_{\mu,\gamma}\|)\right][\|M\| + \|N\|\|G_{\mu,\gamma}\|]
\end{aligned}$$

Now, noting that

$$\|M\| \leq \left\|C_\gamma^{-1/2}(\widehat{H}_m - \widetilde{H}_m)\right\| + \left\|C_\gamma^{-1/2}(\widetilde{H}_m - H_\mu)\right\| \leq \left\|C_\gamma^{-1/2}(\widehat{H}_m - \widetilde{H}_m)\right\| + \sqrt{c_{\mathcal{H}}}\left\|\widetilde{R}_m - R_\mu\right\|,$$

and applying Propositions C.4, C.9, C.10 and C.11 we obtain (63), and, therefore, (64) follows. □

We remark that to bound singular values we can rely on the fact

$$|\sigma_i(\widehat{B}) - \sigma_i(B)| = \frac{|\sigma_i^2(\widehat{B}) - \sigma_i^2(B)|}{\sigma_i(\widehat{B}) + \sigma_i(B)} \leq \frac{|\sigma_i^2(\widehat{B}) - \sigma_i^2(B)|}{\sigma_i(\widehat{B}) \vee \sigma_i(B)}. \tag{67}$$

### C.3.4. VARIANCE OF RRR ESTIMATOR

**Proposition C.13.** *Let* (RC)*,* (SD) *and* (KE) *hold for some* $\alpha \in [1,2]$*,* $\beta \in (0,1]$ *and* $\tau \in [\beta, 1]$*. Given* $\delta > 0$ *and* $\gamma > 0$*, if* $\overline{\varepsilon}_n^{(1)}(\gamma, \delta) < 1/2$*, then with probability at least* $1 - \delta$ *in the draw* $X_0 \sim \pi$*,* $X_{i\Delta t} \sim p(X_{(i-1)\Delta t}, \cdot)$*,* $i \in [n]$*,*

$$\left\| S_\pi(\widehat{G}_{m,\gamma}^r - G_{\mu,\gamma}^r) \right\| \leq \frac{2}{\mu} \left( \overline{\varepsilon}_{n,\ell}^{(3)}(\gamma, \delta/3) + \sqrt{c_{\mathcal{H}}} \, \mu \, s_{\mu,\ell}(\Delta t) \right) \left[ 1 + \frac{c_{\mathcal{H}}/\mu + (\sqrt{c_{\mathcal{H}}}/\mu) \, \overline{\varepsilon}_{n,\ell}^{(3)}(\gamma, \delta/3) + s_{\mu,\ell}(\Delta t)}{\sigma_r^2(Z_\mu) - \sigma_{r+1}^2(Z_\mu) - (c_\alpha/\mu)^2 \, c_{\mathcal{H}}^{\alpha/2} \, \gamma^{\alpha/2}} \right]. \tag{68}$$

*Proof.* Start by observing that $\left\| S_\pi(\widehat{G}_{m,\gamma}^r - G_{\mu,\gamma}^r) \right\| \leq \left\| C_\gamma^{1/2}(\widehat{G}_{m,\gamma}^r - G_{\mu,\gamma}^r) \right\|$ and

$$C_\gamma^{1/2}(\widehat{G}_{m,\gamma}^r - G_{\mu,\gamma}^r) = (C_\gamma^{1/2}\widehat{C}_\gamma^{-1}C_\gamma^{1/2}) \cdot \left[ C_\gamma^{-1/2}(\widehat{C} - C)G_{\mu,\gamma} + C_\gamma^{-1/2}(\widehat{H}_m - H_\mu) \right] \widehat{P}_r + B(\widehat{P}_r - P_r). \tag{69}$$

Taking the norm, using that the norm of orthogonal projector $\widehat{P}_r$ is bounded by one and applying Proposition A.3, we obtain

$$\left\| S_\pi(\widehat{G}_{m,\gamma}^r - G_{\mu,\gamma}^r) \right\| \leq \left\| C_\gamma^{\frac{1}{2}}\widehat{C}_\gamma^{-1}C_\gamma^{\frac{1}{2}} \right\| \left[ \left\| C_\gamma^{-\frac{1}{2}}(\widehat{C} - C) \right\| \|G_{\mu,\gamma}^r\| + \left\| C_\gamma^{-\frac{1}{2}}(\widehat{H}_m - H_\mu \pm \widetilde{H}_m) \right\| \right] + \|B\| \frac{\left\| B^*B - \widehat{B}^*\widehat{B} \right\|}{\sigma_r^2(B) - \sigma_{r+1}^2(B)}.$$

To complete the proof it suffices to apply Propositions C.2, C.3, C.9, C.10, C.11 A.3 and C.12. □

We remark that the previous proof simplifies when we do not have the rank reduction, that is we can estimate

$$\left\| S_\pi(\widehat{G}_{m,\gamma} - G_{\mu,\gamma}) \right\| \leq \left\| C_\gamma^{\frac{1}{2}}\widehat{C}_\gamma^{-1}C_\gamma^{\frac{1}{2}} \right\| \left[ \left\| C_\gamma^{-\frac{1}{2}}(\widehat{C} - C) \right\| \|G_{\mu,\gamma}\| + \left\| C_\gamma^{-\frac{1}{2}}(\widehat{H}_m - H_\mu \pm \widetilde{H}_m) \right\| \right]$$

$$\leq \frac{2 \, \overline{\varepsilon}_{n,\ell}^{(3)}(\gamma, \delta/3)}{\mu} + 2\sqrt{c_{\mathcal{H}}} \, s_{\mu,\ell}(\Delta t), \tag{70}$$

and further obtain that

$$\left\| \widehat{G}_{m,\gamma} \right\| \leq \|G_{\mu,\gamma}\| + \frac{1}{\sqrt{\gamma}} \left\| C_\gamma^{1/2}(\widehat{G}_{m,\gamma} - G_{\mu,\gamma}) \right\| \leq \frac{c_\alpha \, c_{\mathcal{H}}^{(\alpha-1)/2}}{\mu} + \frac{1}{\sqrt{\gamma}} \left( \frac{2 \, \overline{\varepsilon}_{n,\ell}^{(3)}(\gamma, \delta/3)}{\mu} + 2\sqrt{c_{\mathcal{H}}} \, s_{\mu,\ell}(\Delta t) \right). \tag{71}$$

### C.4. LaRRR operator norm error bounds

**Theorem 6.2.** *Let* $L$ *be sectorial operator such that* $w_\star = -\lambda_2(L + L^*)/2 > 0$*. Let* (BK)*,* (RC) *and* (SD) *hold for some* $\alpha \in [1,2]$ *and* $\beta \in (0,1]$*, respectively, and* $\mathrm{cl}(\mathrm{Im}(S_\pi)) = \mathcal{L}_\pi^2(\mathcal{X})$*. Given* $\delta \in (0,1)$ *and* $r \in [n]$*, let*

$$\gamma \asymp \left( \frac{\ln^3(n/\delta)}{n \, \Delta t \, w_\star} \right)^{\frac{1}{\alpha+\beta}}, \quad \varepsilon_n^\star(\delta) = \left( \frac{\ln^3(n/\delta)}{\mu^{\frac{2(\alpha+\beta)}{\alpha}} n \, w_\star} \right)^{\frac{\alpha}{2\beta+3\alpha}} \tag{21}$$

$\Delta t = \varepsilon_n^\star$ *and* $1/\ell = o(\varepsilon_n^\star)$*, then there exists a constant* $c > 0$*, depending only on* $\mathcal{H}$ *and* $\sigma_r(R_\mu S_\pi) - \sigma_{r+1}(R_\mu S_\pi) > 0$*, such that for large enough* $n \geq r$ *with probability at least* $1 - \delta$ *in the draw of* $\mathcal{D}_n$ *it holds that*

$$\mathcal{E}(\widehat{G}_{m,\gamma}^r) \lesssim \max\left( \widehat{\sigma}_{r+1}, \, \sigma_{r+1}(R_\mu S_\pi) \right) + c \, \varepsilon_n^\star(\delta)). \tag{22}$$

*Proof of Theorem 6.2.* Note first that under the gap condition $\sigma_r(R_\mu S_\pi) - \sigma_{r+1}(R_\mu S_\pi) > 0$ and by choice of $\gamma$ in (73), we have for $n$ large enough that $\sigma_r^2(Z_\mu) - \sigma_{r+1}^2(Z_\mu) - (c_\alpha/\mu)^2 \, c_{\mathcal{H}}^{\alpha/2} \, \gamma^{\alpha/2} \geq (\sigma_r(R_\mu S_\pi) - \sigma_{r+1}(R_\mu S_\pi))/2 > 0$.

Next we set $l = n/2$. By definition of the rates $\bar{\varepsilon}_n(\delta)$, $\bar{\varepsilon}_n^{(1)}(\gamma,\delta)$, $\bar{\varepsilon}_{n,l}^{(2)}(\gamma,\delta)$ and $\bar{\varepsilon}_{n,l}^{(3)}(\gamma,\delta)$, and since $\tau \in (\beta,1]$, we have for $n$ large enough, $\bar{\varepsilon}_n^{(1)}(\gamma,\delta/3) \in (0,1/2)$, $n\mu\Delta t \gg 1$ and

$$\bar{\varepsilon}_{n,\ell}^{(3)}(\gamma,\delta/3) + \sqrt{c_{\mathcal{H}}}\,\mu\,s_{\mu,\ell}(\Delta t) \lesssim \frac{\ln^2\left(\frac{n}{\delta}\right)}{\sqrt{\Delta t\,w_\star\,\gamma^\beta\,n}} + \Delta t \ll 1.$$

We note indeed that $l = n/2$ satisfies the condition $1/l = o(\varepsilon_n^\star)$ for $n$ large enough. Hence, recalling the decomposition of the operator norm error in (44) and Proposition C.13, we get with probability at least $1 - \delta$,

$$\mathcal{E}(\widehat{G}_{m,\gamma}^r) \lesssim \sigma_{r+1}(B) + \frac{1}{\mu}\big(\overbrace{\gamma^{\alpha/2} + \underbrace{\frac{\ln^2\left(\frac{n}{\delta}\right)}{\sqrt{\Delta t\,w_\star\,\gamma^\beta\,n}}}_{\textbf{(B)}} + \Delta t}^{\textbf{(A)}}\big). \tag{72}$$

By balancing **(A)** with respect to $\gamma$ first and then **(B)** with respect to $\Delta t$, we derive the upper bound on **(B)** $\leq \varepsilon_n^\star(\delta)$ for the choices of $\gamma$ and $\Delta t$ given in (73).

Finally, if $\sigma_{r+1}(B) = 0$ then we proved the result. Otherwise if $\sigma_{r+1}(B) > 0$, then an union bound combining (72) with (64) and (67) gives the final bound. $\qquad\square$

### C.5. LaRRR spectral operator bounds

First, recalling Proposition 6.1, essentially proven in (Kostic et al., 2022; 2023), with the only difference that instead of applying it to TO, we apply it to the resolvent of IG. This exposes the perturbation level such that the pseudospectrum of $R_\mu$ contains an eigenvalue of its estimator. The bound on the perturbation level comes in two forms: one specific to the targeted eigenvalue and one uniform over all leading $r$ eigenvalues. Since we have resolved the bound on the operator norm error, the only term that remains is the metric distortion for which we have the following

$$\eta(\widehat{h}_i) \leq \frac{\|\widehat{G}\|}{\sigma_r(S_\pi\widehat{G})} \quad \text{and} \quad \left|\widehat{\eta}_i - \eta(\widehat{h}_i)\right| \leq \left(\eta(\widehat{h}_i) \wedge \widehat{\eta}_i\right)\eta(\widehat{h}_i)\widehat{\eta}_i\left\|\widehat{C} - C\right\|.$$

Now, applying this to the LaRRR estimator, since we have that $|\sigma_r(S_\pi\widehat{G}_{m,\gamma}^r) - \sigma_r(S_\pi G_{\mu,\gamma}^r)| \leq \|S_\pi(\widehat{G}_{m,\gamma}^r - G_{\mu,\gamma}^r)\|$ and

$$\sigma_r(S_\pi G_{\mu,\gamma}^r) \geq \sigma_r(C_\gamma^{1/2}G_{\mu,\gamma}^r) - \sqrt{\gamma}\left\|G_{\mu,\gamma}^r\right\| \geq \sigma_r(B) - \sqrt{\gamma}\left\|G_{\mu,\gamma}\right\|,$$

using Propositions C.3, C.13 together with (71) we obtain that for every $i \in [r]$ metric distortion (in the worst case) is bounded by

$$\eta(\widehat{h}_i) \leq \frac{(c_\alpha/\mu)\,c_{\mathcal{H}}^{(\alpha-1)/2} + \gamma^{-1/2}\varepsilon_n^\star(\delta)}{\sigma_r(R_\mu S_\pi)} + \frac{\varepsilon_n^\star(\delta)}{\sigma_r^2(R_\mu S_\pi)},$$

which further ensures that $\eta(\widehat{h}_i) - \widehat{\eta}_i \lesssim \varepsilon_n^\star(\delta)$. Hence, checking that

$$\gamma^{-1/2}\varepsilon_n^\star(\delta) \asymp \left(\frac{\ln^3(n/\delta)}{n\,\Delta t\,w_\star}\right)^{\frac{-1}{2(\alpha+\beta)}}\left(\frac{\ln^3(n/\delta)}{\mu^{\frac{2(\alpha+\beta)}{\alpha}}\,n\,w_\star}\right)^{\frac{\alpha}{2\beta+3\alpha}} \asymp \left(\frac{\ln^3(n/\delta)}{n\,w_\star}\right)^{\frac{-1}{2(\alpha+\beta)}}\left(\frac{\ln^3(n/\delta)}{\mu^{\frac{2(\alpha+\beta)}{\alpha}}\,n\,w_\star}\right)^{\frac{\alpha}{2\beta+3\alpha}\left(1+\frac{1}{2(\alpha+\beta)}\right)}$$

that is

$$\gamma^{-1/2}\varepsilon_n^\star(\delta) \asymp \left(\frac{\ln^3(n/\delta)}{n\,w_\star}\right)^{\frac{2\alpha^2+2\alpha\beta-2\alpha-2\beta}{2(\alpha+\beta)(3\alpha+2\beta)}}\mu^{-\frac{1+2\alpha+2\beta}{3\alpha+2\beta}}$$

remains bounded w.r.t. $n \to \infty$ for $\alpha \geq 1$, we can apply Theorem 6.2 and Proposition A.2 to obtain the statement of Theorem 6.3. Alternatively, we could apply Proposition A.5 to obtain that for general (non-normal) sectorial operator the estimated eigenvalues satisfy

$$\frac{|\lambda_i - \widehat{\lambda}_i|}{|\mu - \lambda_i||\mu - \widehat{\lambda}_i|} \lesssim \|F\|\,\|F^{-1}\|\left(\widehat{\sigma}_{r+1}\widehat{\eta}_i \wedge \frac{\sigma_{r+1}(R_\mu S_\pi)}{\sigma_r(R_\mu S_\pi)}\right) + \varepsilon_n^\star(\delta),$$

where $F$ is the bounded operator with bounded inverse such that $F^{-1}LF$ is diagonal operator.

## C.6. LaRRR bounds in the misspecified setting

In this section, we briefly discuss the case where the eigenfunctions of the generator do not belong to RKHS, i.e., the misspecified learning setting when $\alpha < 1$. First, we note that the misspecified setting for the learning of transfer operators was considered in (Li et al., 2022) in terms of the excess risk (Hilbert-Schmidt norm) of kernel ridge regression (KRR), while the error (operator norm) of KRR, principal component regression (PCR), and reduced-rank regression (RRR) was addressed in (Kostic et al., 2023). In both works, risk/error could be made arbitrarily small for every $\alpha \in (0, 1)$, however at arbitrarily slow rate as $\alpha \to 0$. By adapting our proofs, notably in (68) by avoiding the splitting of $G_{\mu,\gamma}$ from the concentration result in Proposition C.10, one can also derives of the operator norm error for learning of $R_\mu$.

On the other hand, recalling Proposition 6.1, the main challenge of the misspecified setting arises when developing spectral learning rates. Namely, since given any eigenfunction $f \in \mathcal{L}_\pi^2(\mathcal{X}) \setminus [\mathcal{H}]$, for every sequence $(h_k)_{k\in\mathbb{N}} \subseteq \mathcal{H}$ of functions in $\mathcal{H}$ that approximates it, i.e., such that $\|f - S_\pi h_k\| \mathcal{L}_\pi^2 \to 0$, metric distortions explode, that is $\eta(h_k) \to \infty$. Consequently, with current approaches, it is impossible to derive learning bounds for an eigenvalue whose corresponding right eigenfunction lies outside of the RKHS.

## C.7. LaRRR bounds for multiple unevenly sampled trajectories

Here we only derive a bound on the operator norm excess risk when we have access to multiple iid trajectories sampled unevenly, noting that spectral bounds then readily follow in the same fashon as above. Here $n$ stands for the number of trajectories and $\ell$ is the number of samples on each trajectory.

**Theorem C.14.** *Let $L$ be sectorial operator such that $w_\star = -\lambda_2(L+L^*)/2 > 0$. Let **(BK)**, **(RC)** and **(SD)** hold for some $\alpha \in [1, 2]$ and $\beta \in (0, 1]$, respectively, and $\mathrm{cl}(\mathrm{Im}(S_\pi)) = \mathcal{L}_\pi^2(\mathcal{X})$. Given $\delta \in (0, 1)$ and $r \in [n]$, let*

$$\gamma \asymp \left( \frac{\ln^2(\ell\delta^{-1})}{n\, w_\star} \right)^{\frac{1}{\alpha+\beta}}, \quad \varepsilon_n^\star(\delta) = \left( \frac{\ln^2(\ell\delta^{-1})}{n\, w_\star} \right)^{\frac{\alpha}{2(\beta+\alpha)}}, \tag{73}$$

*then there exists a constant $c > 0$, depending only on $\mathcal{H}$ and gap $\sigma_r(R_\mu S_\pi) - \sigma_{r+1}(R_\mu S_\pi) > 0$, such that for large enough $n \geq r$ and $l \geq 1$ with probability at least $1 - \delta$ in the draw of $\mathcal{D}_n$ it holds that*

$$\mathcal{E}(\widehat{G}_{m,\gamma}^r) \leq \widehat{\sigma}_{r+1} \wedge \sigma_{r+1}(R_\mu S_\pi) + c\left( \frac{\varepsilon_n^\star(\delta)}{\mu} + \kappa^2 \Delta t \right), \tag{74}$$

*where $\Delta t = \max_{j\in[\ell+1]}(t_j - t_{j-1})$ is the maximal time-step, $t_\ell$ is time-horizon and conditioning number $\kappa = \max_{j\in[\ell+1]}(t_j - t_{j-1})/\min_{j\in[\ell+1]}(t_j - t_{j-1}) \in [1, +\infty)$ quantifies the unevenness of the sampling.*

*Proof.* We proceed with the same decomposition (30) of the excess as in the proof of Theorem 6.2. The steps **(0)**, **(I)** and **(II)** remain unchanged. The integration bias in **(III)** is again tackled with Proposition C.4. As for the estimator variance in step **(IV)**, we now use concentration result for multiple iid trajectories instead of the results we derived in App. C.3 for a single $\beta$-mixing trajectory. Specifically, Proposition C.7 is replaced by Proposition 14 in (Kostic et al., 2023). Proposition 11 is replaced by Proposition 13 in (Kostic et al., 2023) with $n$ replaced by $n(l+1)$. Proposition 14 is replaced by Proposition 17 in (Kostic et al., 2023). Hence instead of Proposition 15, we obtain the following control on the variance term. For $n$ and $\ell$ large enough, in view of (40), we have with probability at least $1 - \delta$,

$$\left\| S_\pi(\widehat{G}_{m,\gamma}^r - G_{\mu,\gamma}^r) \right\| \lesssim \frac{1}{\mu}\left( \frac{\ln\left(\frac{l}{\delta}\right)}{\sqrt{w_\star\, \gamma^\beta\, n}} \right) + \kappa^2\Delta t + \frac{e^{-\mu t_\ell}}{\mu} \tag{75}$$

Thus we get w.p.a.l. $1 - \delta$,

$$\mathcal{E}(\widehat{G}_{m,\gamma}^r) - \sigma_{r+1}(B) \lesssim \frac{1}{\mu}\left( \underbrace{\gamma^{\alpha/2} + \frac{\ln\left(\frac{l}{\delta}\right)}{\sqrt{w_\star\, \gamma^\beta\, n}}}_{(A)} \right) + \kappa^2\Delta t + \frac{e^{-\mu t_\ell}}{\mu}, \tag{76}$$

We can balance **(A)** as in the proof of Theorem 6.2.

Next set $\underline{\Delta t} = \min_{j\in[\ell+1]}(t_j - t_{j-1})$. Note we have the following conservative bound for $l$ large enough

$$\frac{e^{-\mu t_\ell}}{\mu} \leq \frac{e^{-\mu\underline{\Delta t}l}}{\mu\underline{\Delta t}}\Delta t \lesssim \Delta t.$$

The end of the proof follows from the same argument as in the proof of Theorem 6.2. $\square$

# D. Experiments

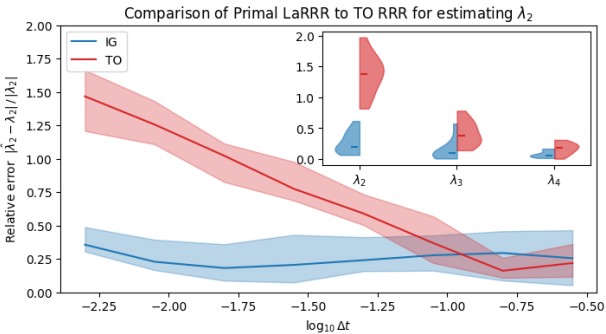

*Figure 4.* Comparison of the primal LaRRR algorithm to the TO RRR method for estimating the slowest timescales of the process in the basis of random Fourier features (Rahimi & Recht, 2007). As predicted by our theory, LaRRR (blue) remains stable as $t \to 0$, whereas the error of TO method (red) diverges. The main plot shows the three quartiles of relative error for $\lambda_2$ across 10 independent trajectories of a 1D Langevin process on a triple-well potential. The inset displays the distributions of the top three eigenvalues for $\Delta t = 0.005$.

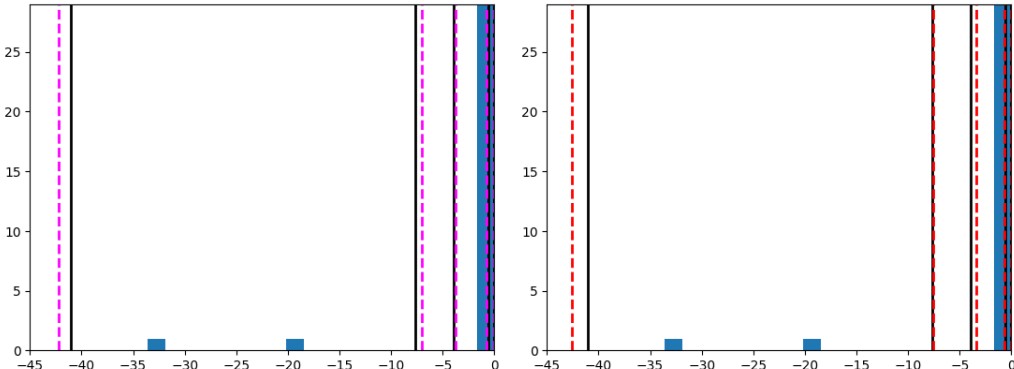

*Figure 5.* In the setting of experiment on Overdamped 1D Langevin dynamics, we show how LaRRR (presented in red) compares to two physics-informed IG baselines: energy based regression of Dirichlet form by Kostic et al. (2004) (magenta), and Galerkin projection by Hou et al. (2023) (blue). The eigenvalues are shown on the horizontal axis, and the vertical axis indicates the histogram hight of those estimated by Galerkin projection. While both the method of Kostic et al. (2004) (left) and LaRRR (right) recover the eigenvalues of IG (black lines), Galerkin projection (due to the unbounded nature of IG in $\mathcal{L}^2_\pi$) produces many spurious eigenvalues near the origin.

