# OpenReview forum: "Laplace Transform Based Low-Complexity Learning of Continuous Markov Semigroups"
_ICML.cc/2025/Conference — ICML 2025 poster_

### Official Review · Reviewer_iVn3 · 2025-02-18

**Overall Recommendation:** 3

**Summary:**

The paper presents theoretical developments to learning the infinitesimal generators of markov semigroups using Laplace transform.

**Claims And Evidence:**

The paper's claims all theoretically substantiated. Empirical evidence backs the claims, but the experiments are of very small scale.

**Essential References Not Discussed:**

No issues.

**Experimental Designs Or Analyses:**

The experiments are small scale, but seem reasonable.

**Methods And Evaluation Criteria:**

Ok

**Other Comments Or Suggestions:**

I want to start by stating that I’m not a theoretician or expert in the paper’s domain, but have knowledge on many subtopics of this paper. Hopefully my review can still be useful in showing how the manuscript opens up to an interested outsider.

The paper is very theoretical and technical with abundance of math. The paper does little to make the material accessible to wider audience. The presentation is technically excellent, although some parts show some looseness. I was able to follow maybe third of the material, and rest was too deep or cursory to be understood (which is ok).

However, one should be able to follow the higher-level story regardless: what problem are we solving, why math is introduced, and what does the math achieve. The paper is not particularly clear on these aspects, and there is insufficient motivation and introduction to the theoretical presentation to be even following the higher-level story. There is a lot of math’itis where math happens but it’s not clear why. The material is also not clearly contextualised into wider literature. The paper is presenting probably too much material (experiments start at last page), and the manuscript would have improved by prioritising some of the material (and some to appendix) to allow for room to breath.

I'm rating the paper weak accept due to its strong theoretical contributions, although the presentation has lots of room for improvement. I'm looking forward to hearing author comments and revise my review accordingly.

Questions:
- Is TO a standard wording for this concept in the literature? How does this relate to pushforwards, or to FNO’s or to operator learning?
- What’s the connection of IG to Fokker-Planck or to Ito’s lemma? Eq 7 looks like an expected Ito. Can you contextualise this a bit more?
- The scope of the work is abstract: what task are you solving when you "learn markovian semigroups"? I think this paper is about learning operators, especially in SDE context, but what problems can we then solve by these operators or by learning semigroups? Are we learning the solution operators of PDEs, or something more general?
- What kind of observations do we need?
- The open problem is quite vague. The introduction states that unboundedness of IGs makes designing “estimators” “challenging”. What specific problems does the unboundedness pose? Why does it make things challenging? It seems that earlier methods can learn PDE operators just fine: how are they deficient?
- In stochastic processes we usually operate within some boundary constraint, or within some bounded domain. Is the unboundedness of IG then a problem in those contexts? Can you give a bit more explanation what the unboundedness of IG means?

**Other Strengths And Weaknesses:**

S: The paper seems to present a strong theoretical advancement to operator learning

S: The theoretical presentation is convincing.

S: The empirical results support the contributions.

W: The presentation lacks exposition and the higher-level story gets drowned in all the technicalities. It was difficult to follow what was happening, and why. The technical presentation is dense and convoluted.

W: The results are consistent with the contributions, but they have small scale and are thus anecdotal. The method is only compared to one other method. The results show that one can learn the first two eigenfunctions from SDE dynamics. It is not demonstrated how is this beneficial, or what problem this solves. Why do we need these eigenfunctions? There are no benchmarks or result tables. I feel like the paper is not really realising its own potential, or had no room left for showing how the contributions lead to improved methods and results.

**Questions For Authors:**

- Why the $p_t$ has two arguments? What is $y$? Should this be perhaps interpreted as $p_{t,s}(X_t=x, X_s=y)$ for $t > s$?
- Is A_t a pushforward? Can you elaborate a bit more on what this means conceptually, and how it connects with IG
- What does A^* mean?
- The paper claims to study the transfer operators on a space of invariant measures. But in that space the TO seemingly does nothing: what is here to study?
- $f$ is first just a functional, and then later it’s from the space of invariant measures. Can you explain if these are two different $f$’s?
- In sec 2 L was an operator that turns functional f(x0) into a derivative of expected functional E[f(xt) | x0]. In sec 3 L is a mapping between two invariant measures. Are these two descriptions of L complementary, or did we just define a new, different L in sec 3?
- I don’t understand why the H and \calL have different metric structures. Surely if f,g are in both H and L, their inner products should be the same. Can you elaborate?
- Why can you assume that the RKHS is a subset of calL? The connection between the RKHS and stationary measures is not clear here. Does the kernel span only invariant measure functions? If so, how can you know such a kernel exists or how is it defined?
- It would be useful to expand example 2.2. to also describe it wrt A/f/L/etc
- I’m a bit confused of the setting of sec 3 first paragraph. We introduce a bunch fo stuff, but there is little motivation or introduction. For instance, it’s not clear what the \calL is, why we want the L, and why H is a subset of calL, and why we introduced kernels. All of this happens, but it's purpose is not described: what is the task/problem/context?
- Eq 9 has phi, but usually that is unknown. How do we handle this?
- In eq 9 we draw X0 from pi. But pi is a invariant distribution, which is where the process converges at some (late) time. Surely it’s not the initial time: so why do we then write X0?
- The eq 9 has a H-norm. But the square error seems to be over just scalar evaluations of measures: why do we need the Hilbert norm for scalars?
- Eq 10 shows that some function of X0 is an integral of Xt’s. What does this mean conceptually? Is this perhaps the solution operator in the hilbert space?
- I struggle to follow eqs 9 and 10. I don’t see much connection to earlier material since notation changes from A/f/L/S to phi/psi/pi. Can you elaborate a bit more here?
- The eq 10 has the exp form for the solution of the process. Is this a limitation? Does this apply to all kinds of SDEs or semigroups?
- Is eq 11 a quadrature?
- What are A_t_j? Where do these come from, and are they known?
- What is h or v?

**Relation To Broader Scientific Literature:**

The work is positioned sufficiently to its specific domain, but the broader picture is lacking.

**Theoretical Claims:**

I read the entire paper carefully, and understood some. I see no issues with correctness.

---

> ### Author Rebuttal · Authors · 2025-04-01
>
> Thank you for your insightful feedback. Due to space constraints, we address the key issues and questions briefly, committing to incorporate all suggestions in our revision. Further details are available upon request.
>
> # Empirical evidence:
> For our additional [results](https://green-chantal-92.tiiny.site), please see reply to __nvbC__ .
>
> # Presentation
> Following your suggestions, we'll emphasize motivations in Sec 3, address unboundedness effects, expand the app. on kernel-based operator learning for TOs and IGs, and add a notation table. If accepted, we'll use the extra page to enhance transitions between challenging concepts.
>
> # Questions
> - In stochastic process theory, TOs form the Markov semigroup associated with the process, acting as direct image functors in measurable spaces. When a measurable func pushes a measure forward, TO describes density transformations. Neural-network learning methods for TO relate to FNO but are distinct concepts, (Nakao and Mezic, 2020).
>
> - TOs are linear (evolution) operators defined on func spaces. We can say that $A_t$ acts as the pushforward of a function $f$ under the transition probability defined by $p_t$, making them crucial for understanding $(X_t)\_t$ dynamics. On the other hand, IGs, defined via Eq 25, are linear differential operators related to the eqs of motion via backward Kolmogorov eq. $\partial_t \mathbb{E}[f(X_t)|X_0=\cdot ]=L f$ for observable $f$ and forward Kolmogorov eq. (Fokker-Planck) $\partial_t q_t =L^* q\_t$ for probability density $q_t$ of $X_t$ (* being adjoint). Indeed, Eq 7 is obtained by applying Itô's lemma to $f(X_t)$ where $X$ is the solution of Eq 6.
>
> - Learning the spectral decomposition of TO/IG is equivalent to learning the solution of an associated SDE, so we can predict the evolution of distributions, see reply to __CrN8__. Working with IG allows us to do this reliably in continuous time.
>
> - Aș explained in Sec 5, the data is the observed trajectory of states.
>
> - Galerkin projection methods (empirical risk minimization) suffer from spectral pollution when applied to unbounded operators, (Kato 2012, Kostic et al. 2024). When the process evolves in a bounded domain $\mathcal X$, the IG are still typically unbounded, i.e. the IG doesn’t map all funcs $\mathcal{X}\to\mathbb{R}$ (observables of the process) in the $\mathcal{L}^2_\pi$ space to other funcs in that space - it’s only defined on a proper subset (domain of IG).
> # Q. for Authors
>
> [1] $(p_t)_t$ is a family of transition probability densities $p_t(x,y)dy=P(X_t \in dy|X_0=x)$ for all $x,y\in\mathcal{X}$. In this sense, $p_t(x,y)dy$ quantifies the probability that the process $(X_t)_t$ is in the infinitesimal set $dy$ at time $t$, given that it started at $x$.
>
> [2-3] See above.
>
> [4-5] While the action of TO can be defined for any measurable func $\mathcal{X}\to\mathbb{R}$, to characterize dynamics via semigroup, one needs the space of such funcs to be invariant under the action of TOs, and a typical choice is $\mathcal{L}_\pi^2$. Note that invariance of measure $\pi$ (both marginals of $P[X_t,X_0]$ are $\pi$), doesn't mean that TOs (conditional laws $P[X_t|X_0]$ ) are trivial.
>
> [6] In fact, for all $t$, $A_t$ is an operator that turns a func $f$ into a func $A_t f$ s.t. for any $x\in\mathcal{X}$, $A_t f(x)=\mathbb{E}[ f(X_t )|X_0 =x]$. While $L$ is an operator that maps a func $f\in dom(L)=\mathcal{W}\_\pi^{1,2}\subset \mathcal{L}\_\pi^2$ to a func $Lf \in\mathcal{L}\_\pi^2$, meaning that $\int_{\mathcal{X}} |L f(x)|^2\pi(dx)<\infty$.
>
> [7-8] In the RKHS theory it is standard to assume $\mathcal{H}\subset\mathcal{L}^2_{\pi}(X)$, $\pi$ being the probability of data samples (e.g. Gaussian kernel), and the main issue in the learning bounds is the difference between $\mathcal{H}$ (chosen) and $\mathcal{L}^2_{\pi}(X)$ (unknown) norms, e.g. (Steinwart&Christmann,2008).
>
> [9] For conciseness, the defs of $A$, $L$, and $\pi$ in the Ornstein-Uhlenbeck case are in App A.1. If space permits we can include them in the main text.
>
> [10-11] The funcs $\phi$ are not unknown but are provided when choosing the RKHS, e.g. $\phi(x)=e^{-\|x-\cdot\|^2/(2\sigma^2)}$ for Gaussian kernel.
>
> [12-15] In Eqs 9-10, $X_0$ is not the SDE's initial cond. but any random variable with the invariant measure, see reply to Q1 of __PJzG__ for extra clarity. Note that $\phi(X_t),\psi(X_0), G^*\phi(X_0) \in \mathcal{H}$ are funcs, not  scalars.
>
> [14 & 16-17] $\mathbb{E} [\psi(X_0)|X_0=\cdot]:\mathcal{X}\to\mathcal{H}$ is the Riesz representation of lin. operator $R_\mu$ in RKHS $\mathcal{H}$, and our learning objective. It is the regression func related to the excess risk, see Prop A.1, where the risk is given via the target feature $\psi(X_0)$. Since we cannot observe it precisely (due to the integral form) in Eqs 11-12 we show how to approximate it via quadrature.
>
> [18-19] $A_{t_j}$ is TO for time-delay $t_j$ (see Eq 2), $h$ was used as a generic func in RKHS $\mathcal{H}$ and $v$ as a vector in $\mathbb{R}^N$.

---

> > ### Comment · Reviewer_iVn3 · 2025-04-02
> >
> > I've read the rebuttal. I'm keeping my score. I don't recognise the paper demonstrating any practical impact from the theoretical treatment.

---

> > > ### Author Response · Authors · 2025-04-03
> > >
> > > We thank the reviewer for their feedback. In our rebuttal (due to the limited space) we have focused on answering all their questions on the mathematical framework we have studied and to clarifying potential misunderstanding of the theory. We hope that this was useful to the reviewer, as it was beneficial to us to improve the presentation. Since we couldn’t **provide a larger context and emphasize enough the broader impact** of our theoretical and empirical results, let us try to elaborate now.
> > >
> > > Understanding and accurately learning the spectral decomposition of the stochastic Koopman operator for continuous-time stochastic dynamical systems is **pivotal to a number of  machine learning (ML) and artificial intelligence (AI) applications**. This approach, as we briefly review, has significantly impacted diverse fields, including molecular dynamics, time-series clustering, computational neuroscience, and beyond.
> > >
> > > - **Molecular Dynamics** The field of molecular dynamics has particularly benefited from spectral decomposition methods of Markov semigroups. Research on AI-augmented molecular dynamics using statistical mechanics demonstrates the importance of accurate spectral gap identification—the separation between slow and fast modes—in molecular simulations, see (Schütte et al. 2001). Work that we extend (Kostic et al. 2024) was used in Devergne et al (2024) to demonstrate the impact on **IG based methods in accelerating simulations and unlocking practical identification of meta-stable states**. They stress that, unlike popular TO that have limited utility, _“ [...] the infinitesimal generator is the adequate tool to deal with dynamical information from biased data”_,. Further, in Devergne et al (2025), published in the journal of Chemical Physics, authors conclude that IG method _“[...] offers the exciting possibility of making close contact with experiments.”_ Note that our experiments show equally good results like (Kostic et al. 2024) but at the order of magnitude faster performance, which is particularly important in **scaling up IG methods to larger proteins**.
> > >
> > > - **Spectral Clustering.** Klus and Conrad (2023) introduced a Koopman-based spectral clustering method tailored for directed and time-evolving graphs. By leveraging TOs, their approach shows how to identify coherent sets within complex networks, enhancing the analysis of temporal data structures.  Further, Cabannes and Bach (2024), the results on Galerkin projections of IG, which we have directly generalized and improved, stress that learning spectral decomposition of IG _“[...] opens exciting follow-ups for any spectral-based algorithms, could it be spectral clustering, spectral embeddings, or spectral distances.”_. Note that in **the additional experiment we demonstrate that in contrast to this method our approach is reliable** in learning spectral decomposition.
> > >
> > > - **Computational Neuroscience.** Marrouch et al. (2020) applied data-driven Koopman operator techniques to analyze brain activity. Their work demonstrates the utility of Koopman spectral methods in capturing the spatiotemporal dynamics of neural signals, offering insights into brain function and potential applications in neurological disorder diagnostics. Further Ostrow et al. (2023) develop dynamical similarity analysis based on TOs that _“[...] opens the door to comparative analyses of the essential temporal structure of computation in neural circuits.”_, showing that TO based similarity metrics can distinguish learning rules in an unsupervised manner.
> > >
> > > Collectively, these **studies underscore the broad impact spectral decomposition of the TO (stochastic Koopman operators) and their IG across various ML and AI domains**, and we strongly believe that our contribution to **theoretical understanding and methodological approach to learning IG’s spectral decomposition** is critical for building **reliable** methods in such scientific applications.
> > >
> > >
> > > ### **Additional Ref.**
> > > - Schütte, C., Huisinga, W., & Deuflhard, P. (2001). Transfer operator approach to conformational dynamics in biomolecular systems. Springer Berlin Heidelberg.
> > > - Devergne, T., Kostic, V., Pontil, M., Parrinello M. (2024) From biased to unbiased dynamics: An infinitesimal generator approach, NeurIPS 2024
> > > - Devergne, T., Kostic, V., Parrinello M., Pontil, M. (2025) Slow dynamical modes from static averages. Journal of Chemical Physics. 162 (12)
> > > - Klus, S. and Conrad, N. D., (2023) Koopman-based spectral clustering of directed and time-evolving graphs. Journal of nonlinear science 33.1.
> > > - Marrouch, N., Slawinska, J., Giannakis, D., & Read, H. L. (2020) Data-driven Koopman operator approach for computational neuroscience. Annals of Mathematics and Artificial Intelligence, 88(11)
> > > - Ostrow, M., Eisen, A., Kozachkov, L., & Fiete, I. (2023). Beyond geometry: Comparing the temporal structure of computation in neural circuits with dynamical similarity analysis. NeurIPS 2023

---

### Official Review · Reviewer_CrN8 · 2025-03-12

**Overall Recommendation:** 4

**Summary:**

The authors present an approach for learning continuous Markov semigroups. Notably their approach comes with theoretical guarantees at any time-lag. In addition, their approach scales linearly in the state dimension opening the door to apply their methods on high-dimensional problems. Finally they demonstrate their approach on two test problems.

**Claims And Evidence:**

- *Novel approach for learning the spectral decomposition of the IG for Markov semigroups*: The proposed approach is novel as far as I'm aware. This is supported by a solid literature review and a detailed description of the approach. I personally found the algorithm blocks to be particularly helpful.
- *Statistical guarantees for their approach*: The authors prove tight error bounds for their approach which they validate with numerical studies.
- *Numerical stability for $\Delta t \to 0$*: One of the major claims the proposed approach is that it remains stable as $\Delta t \to 0$ (something which is not true for TO RRR). They show that this is indeed the case in the numerical study in Figure 1.

**Essential References Not Discussed:**

N/A

**Experimental Designs Or Analyses:**

- To reiterate, it wasn't clear to me what to read from Figure 2. The authors mention that the eigen functions have different constant values in the expected metastable states but some estimate for the accuracy of the approach would have been useful.

**Methods And Evaluation Criteria:**

The authors demonstrate their approach on two problems:
- Estimating the eigen values of the generator for a 1D triple well problem.
- Estimating the eigen functions of the alanine dipetide in water with a dimension of 45. It wasn't clear to me what was expected for the ground truth of the first and second eigenfunctions.

While I think these problems do a decent job of sketching out some of the main ideas and motivations for learning the IG, some more impressive numerical studies would have definitely strengthened the paper.

**Other Comments Or Suggestions:**

- under Section 3: "result holds true the minimizers of for the..."

**Other Strengths And Weaknesses:**

Strengths:
- Your introduction and review of the relevant literature was really well-written and easy to follow. I appreciate the effort and care that went into this section and I think it will make your work more broadly accessible to a wider ML audience.

Weaknesses:
- While I think your approach is a strong contribution some sections of your write-up were challenging to parse. While some of this comes with the territory and the nature of the problem you are tackling, I think adding an appendix on notation / simplifying some of your notation would make your work more accessible. For example, I had to dig quite a bit to understand the meaning of $[\cdot]_r$ and $\otimes$.

**Questions For Authors:**

- I'm wondering how much of challenge it might be to apply your approach for estimating latent dynamics (i.e. where not all relevant states are directly observed)?
- As a follow up to this question, might it be possible to apply your approach to estimate the generator for Markov-approximate fractional Brownian motion? [1]
- Can you discuss how you might apply your approach for forecasting?

[1] Daems, Rembert, et al. "Variational inference for SDEs driven by fractional noise." arXiv preprint arXiv:2310.12975 (2023).

**Relation To Broader Scientific Literature:**

This work relates to the broad problem of learning models of Markov processes from data.

**Theoretical Claims:**

- Theorem 6.2 & 6.3: The sketch of the proof seems correct but I haven't gone through all the details carefully.
- The discussion on the comparison between bounds from [Kostic'23a] and [Kostic'24] was really helpful for understanding the motivation of the present work.

---

> ### Author Rebuttal · Authors · 2025-04-01
>
> We appreciate the reviewer’s insightful evaluation and valuable comments. Below, due to space limitations, we briefly address the highlighted weaknesses and respond to the reviewer’s questions, committing to incorporating all feedback in our revision. If needed, we can elaborate further on each point.
>
> ## Additional empirical evidence:
> Following the reviewer's suggestion, we have expanded our empirical evaluation to include IG baselines and support the claims summarized in Table 1. For a brief discussion on the [results](https://green-chantal-92.tiiny.site) of our additional experiments, please see our reply to reviewer __nvbC__ .
>
> ## Presentation
> To help the reader, we will include a table of major notations at the beginning of the appendix, and expand our appendix with more context on the kernel based operator learning and derivation of the algorithms. Further, if accepted, we will use the extra page to include smoother transitions between challenging concepts. Finally, concerning the discussion of the Alanine-Dipeptide experiment, since there is no ground truth for this system, the only thing we could show is that the result aligns with the state-of-the-art expert knowledge in molecular dynamics, (Wehmeyer & Noe, 2018). In the revised manuscript we will discuss more on the interpretation of this figure.
>
> ## Questions
> We thank the reviewer for their questions, which suggest potentially interesting extensions of our framework.
>
> - Partially observed systems: The major problem in partially observed systems, in the context of TO/IG kernel-based learning, lies in the fact that one might lose the universality of representation, and hence encounter the difficulty to properly predict distributions. One possible way to overcome this is to rely on Takens-type theorems (deterministic and/or stochastic) (Sauer et al. ”Embedology”, 1991; ) Koltai and Kunde, “A Koopman–Takens theorem”, 2024), that essentially guarantee, under certain assumptions, that by augmenting the current measurements with the past ones we decrease the loss of information and improve estimation.
>
> - Fractional Brownian motion: Consider the SDE $dX_t = b(X_t) dt + \sigma(X_t) dB_{t}^H$ where $B^H =(B\_{t}^H )\_{t}$ is a fractional Brownian motion. When $H \neq 1/2$, the fractional Brownian motion (fBm) is no longer a semimartingale, which means Itô calculus does not apply directly. Although the equation can still be given a meaning—via the Stieltjes integral for $H > 1/2$ or the Skorokhod integral for $H \in (1/4,1/2)$—the resulting process is no longer Markovian. Our generator-based approach no longer applies in this setting.  However, as you suggest, we could adapt our method to estimate the generator for a Markovian approximation  associated with the autonomous version of the fractional SDE introduced in [1]:  $dX_t = b_\theta (X_t) dt + \sigma_\theta (X_t) dB_{t}^H.$  This process can be approximated using a finite linear combination of Ornstein-Uhlenbeck processes, $\hat{B}^H(t)$, leading to the so-called Markov-Approximate fractional SDE (MA-fBMSDE):  $dX_t = b_{\theta} (X_t) dt + \sigma_{\theta} (X_t) d\hat{B}\_{t}^H,$  where  $d\hat{B}\_{t}^H = \sum_{k=1}^{K} \omega_k dY_t^k, \quad \text{with} \quad dY_t^k = -\gamma_k Y_t^k dt + dW_t.$  By applying Proposition 2, the process $X_t$ can be augmented with a finite number of Markov processes $Y_t^k$ (which approximate $B^H$), forming a higher-dimensional state variable:  $Z_t = (X_t, Y_{t}^1, \dots, Y_{t}^K) \in \mathbb{R}^{D(K+1)}.$ The process $Z = (Z\_{t})\_{t}$ is Markovian and can be described by an ordinary SDE:  $dZ_t = h_{\theta} (Z_t) dt + \Sigma_{\theta} (Z_t) dW_t.$  The infinitesimal generator is given by:  $$\mathcal{L} f(z) =\left( b (x) - \sigma(x) \sum_{k=1}^{K} \omega_k \gamma_k y^k \right) \frac{\partial f}{\partial x} + \sum_{k=1}^{K} (-\gamma_k y^k) \frac{\partial f}{\partial y^k}+ \frac{1}{2} \sum_{i,j} (\Sigma_{\theta} \Sigma_{\theta}^\top)_{i,j} \frac{\partial^2 f}{\partial z_i \partial z_j}.$$ This satisfies our assumptions, allowing us to apply our procedure.
>
> ## Forecasting
> Given IG’s spectral decomposition, Eq (8) provides directly the solution since it enables forecasting of full state distributions beyond just the mean (e.g., $f$ as an indicator function). Hence, we can use our method to forecast $\mathbb{E}[h(X_t)\vert X_0 = x] \approx \sum_{i\in[r]} e^{\hat \lambda_i} \langle \hat{g}\_{i}, h \rangle\_{\mathcal H} \hat{f}\_{i}(x)$, for $h\in\mathcal{H}$, noting that $\hat{g}\_{i}, h \rangle\_{\mathcal H}$ can be computed on the training set via kernel trick, see (Kostic et al 2022). Further, note that this formula extends to all $\mathcal{L}^2_\pi$ functions, at the price of an additional projection error, and that we can, hence, predict evolution of distributions, see e.g. (Klus et al., 2019) and (Kostic at al. "Consistent long-term forecasting of ergodic dynamical systems", 2024). If useful, we can include an empirical example in the main body or appendix.

---

### Official Review · Reviewer_nvbC · 2025-03-13

**Overall Recommendation:** 4

**Summary:**

The paper deals with learning continuous-time Markovian dynamics. While existing methods focus on learning transfer operators, here the authors suggest to learn a spectral decomposition of the semigroup's generator, under some assumptions. This is done by finding a (finite-rank) approximation of the resolvent in an internal RKHS, thus exploiting its appealing properties (e.g. boundness).
This approach is applicable to a relatively broad class of processes, and gives raise to accurate and efficient data-driven algorithms.

## update after rebuttal

The paper and the author's answers are convincing, and my recommendation to accept the paper remains.

**Claims And Evidence:**

All claims made in the submission seem to be supported by a sound theoretical analysis and exact proofs (albeit naturally it was impossible to really get into details within the short review period).
The experimental section, however, does not show a clear case of superiority over existing methods. A comparison is made only on one example and only to one baseline, where even there the results are quite arguable.

**Essential References Not Discussed:**

The review of related work seems adequate.

**Experimental Designs Or Analyses:**

See my answer above.

**Methods And Evaluation Criteria:**

Basically yes, but more numerical experiments are needed. The inset frame in Fig.1 is unclear to me.

**Other Comments Or Suggestions:**

Although overall clarity of the paper is not bad, it is still very easy to get lost with all symbols and notations. It would be recommended to encapsulate notations and definitions into a single table  (in the Appendix, maybe).

It is crucial to add extra experiments and comparisons with more baselines.

**Other Strengths And Weaknesses:**

The submission seems original and novel, with strong theoretical contributions. The empirical contribution, however is not established in my opinion.

**Questions For Authors:**

None

**Relation To Broader Scientific Literature:**

Compared to existing literature, this work extend the analysis to a broad class of dynamics, and find solutions to priorly known drawbacks (e.g. short sampling intervals). The key idea of studying the resolvent seems to be new and mind opening.

**Theoretical Claims:**

I briefly went over proof, though the details are too deep to cover during the short review period.

---

> ### Author Rebuttal · Authors · 2025-04-01
>
> We appreciate the reviewer’s insightful evaluation and valuable comments. Below, due to space limitations, we briefly address the highlighted weaknesses and respond to the reviewer’s questions, committing to incorporating all feedback in our revision. If needed, we can elaborate further on each point.
>
> ## Additional empirical evidence (results can be found [here](https://green-chantal-92.tiiny.site) ):
>
> While we still feel that our work is mainly theoretical, following the reviewer's suggestion, we have expanded our empirical evaluation to include IG baselines and support the claims summarized in Table 1. In particular:
>
> ### 1D Langevin dynamics experiment:
> We extend the original experiment by comparing it to __two IG baselines__ that use prior knowledge of IG:
> -  **Galerkin projection estimate (Hou et al., 2023)**, and
> - **Energy based Dirichlet form regression (Kostic et al., 2024)**.
>
> In Figure 4 (see the above link) we plot true eigenvalues as black vertical lines, while the estimated eigenvalues are plotted as magenta (our dual method) and red (Dirichlet form regression) dashed lines. Further, since the Galerkin projection estimate has numerous spurious eigenvalues, we plot their empirical distribution in the form of histogram, stressing out that expert knowledge is needed for this estimator to extract good estimates from the spurious ones.  Finally, we report that **our results are comparable to the SOTA** (Kostic et al., 2024) estimator,  **despite not using the explicit IG knowledge**,  **while being one order of magnitude faster to train**.
>
> ### Non-normal sectorial IG:
> We further conduct an experiment using the 2D Ornstein-Uhlenbeck process with non-symmetric drift, estimating the leading nontrivial complex-conjugate eigenvalue pair. We compare our primal method based on random Fourier features with the corresponding TO, noting that **Kostic et al. (2024) is inapplicable** in this setting and that **Galerkin projection estimators (Hou et al., 2023) are inefficient**, since, as observed above, they typically result in over 50 eigenvalues in the zone of interest. In Figure 5 (see the link above) we show comparison for 10 random trials using 1000 random features, two different time discretizations $\Delta t=0.01$ and $\Delta t = 0.001$, and corresponding sample sizes $n=10^4$ and $n=10^5$. Note that, to obtain small errors, sample sizes are much higher than in the self-adjoint case (Langevin). This is due to non-normality of the generator, as predicted by our theoretical analysis, see Appendix C.5, where one can see that our method consistently has estimates in the tighter $\varepsilon$-pseudospectrum of the operator.
> ## Presentation
> As suggested, we will include a table of major notations at the beginning of the appendix.

---

### Official Review · Reviewer_PJzG · 2025-03-16

**Overall Recommendation:** 4

**Summary:**

This paper studies a new class of non-parametric learning algorithms for continuous-time Markov processes, specifically for learning the eigenfunctions and eigenvalues of their infinitesimal generator (IG) of the semigroup of transfer operators (TO). While existing methods tend to focus on learning the TO which share the same eigenfunctions to IG, the authors criticize their deteriorating spectral gap as the sampling frequency of the data increases (i.e. as the time-lag decreases). Then for more recent methods that directly learn the IG and its eigenstructure, the authors point out that the challenges coming from the unboundedness of the IG is not well addressed. Thus, the goal of the paper is to propose a learning algorithm for IG that can properly handle its unboundedness. To this end, the authors propose to leverage an auxiliary operator, called the resolvent operator, which has the same eigenfunctions as the IG, and can be obtained through the Laplace transform of the TO. For a class of Markov processes with sectorial IG (this includes all time-reversal-invariant Markov processes and some important non-time-reversal processes), the resolvent is uniformly bounded outside a sector containing the spectrum, which addresses the unboundedness problem. The authors then extend an established prior method (reduced rank regression; RRR) for operator learning in RKHS to learn the resolvent operator, and provide statistical learning guarantees. A particularly new contribution is bounding the integration bias, i.e., the error of Laplace transform under possibly irregular sampling intervals, using spectral perturbation theory. The authors demonstrate the proposed algorithm in two time-reversal-invariant processes. The first experiment demonstrates that while TO learning is sensitive to small time-lag, the proposed method learning the resolvent is robust. The second demonstrates that the proposed method successfully recovers the leading two eigenfunctions of IGs from a molecular dynamics data, in a setup where a previous IG learning method is intractable due to the high dimensionality.

## update after rebuttal

The authors provided detailed clarifications in response to my concerns. I think this is a solid work in the domain of nonparametric learning of dynamical systems and would like to retain my supportive rating.

**Claims And Evidence:**

Most of the claims are supported by theoretical or empirical evidence. Some claims listed below could benefit from additional clarification.

- In pages 2, 3, and Table 1, it is claimed that current methods that directly learn the IG are susceptible to spurious eigenvalues due to the unboundedness of the IG. This claim is not explicitly supported by theoretical results or experiments as far as I can confirm.
- As a related note to the above, while a proper empirical comparison against TO learning method is made in the first experiment, comparison against IG learning methods are only done theoretically, not empirically.
- In page 3, 8, and Table 1, the authors claim that the proposed method is structure-agnostic and apply to the broad class of sectorial IGs which covers not only self-adjoint IGs but also important non-self-adjoint ones. However the two experiments both concern self-adjoint IGs, as far as I understand.
- In page 5, the authors claim that the definition of the sampling operator and its adjoint implies that the empirical estimation of the covariance can be expressed using them. It would be better if the derivation is made more explicit.
- In page 4, the authors claim that while eigenvalues are informative about long-term behavior, they are static properties and fail to capture transient dynamics or the full time evolution of the process. Can the authors elaborate a bit on this? One might argue that eigenvalues decaying rapidly over time can provide information about the transient dynamics.
- In page 7, the authors claim that the eigenfunction learning bound for the considered TO learning method becomes vacuous as the time-lag goes to 0. Can the authors elaborate a bit more on why? I was not able to fully understand the description.

**Essential References Not Discussed:**

Important related works are properly discussed in the paper.

**Experimental Designs Or Analyses:**

Please see the Methods and Evaluation Criteria section.

**Methods And Evaluation Criteria:**

The proposed method is suitable for the problem setup considered in the paper. For the benchmark dataset, the scope of the experiment could be improved to include non-time-reversal-invariant Markov processes, which could add a strong supporting evidence to the paper; please see the Claims and Evidence section.

**Other Comments Or Suggestions:**

Minor typos or ambiguities:
- Page 1: trajecotry -> trajectory
- Page 1: In the introduction it is stated that current kernel methods for TO learning require kernel function selection, but this is a shared limitation with the proposed method.
- Page 2: "the space of functions on [?] that are square-integrable"
- Page 3: Below equation (4), which space is $L^2$? The same goes for the first line of page 13.
- Page 3: Below equation (5), should "the spectrum of $G$" be "the spectrum of $L$"?
- Page 3: In equation (8), the symbol $f$ refers to both the observable and right eigenfunctions of $L$
- Some parts in the appendix denote the resolvent operator by $L_\mu$, while the main text uses $R_\mu$
- Page 3: In equation (9), the reason why the adjoint $G^*$ is used is too implicit. As far as I understand, it is because technically we are regressing the embedded Perron-Frobenius operator (under the terminology of Klus et al. 2019), not the transfer operator (Koopman operator).
- Page 3: "an universal approximation result holds true the minimizers of for the minimizer (9)"
- Page 5: Below equation (18), which joint distribution does $\rho_{j\Delta t}$ represent?
- Page 5: "Ridge Regression solution of the regularized risk , that is without"
- Page 7: What does $\mathrm{gap}_j(R_\mu)$ precisely represent? I might have missed it but it is not defined in the main text.
- Page 13: "the RKHS associated to kernel $k\mathcal{X}\times\mathcal{X}\to\mathbb{R}$"
- Page 15: "(?)Theorem 2]Kostic2022"

Klus et al. Eigendecompositions of Transfer Operators in Reproducing Kernel Hilbert Spaces (2019)

**Other Strengths And Weaknesses:**

Strengths
- Originality: The paper is original in its use of the resolvent operator and its expression as Laplace transform of the Markov semigroup in the non-parametric operator learning context.
- Significance: The paper is significant in its generality (sectorial IGs, and possibly handling irregular sampling intervals) and addressing of the issues of the current TO and IG learning methods (sensitivity to small time-lag and spuriousness due to the unboundedness of IGs, respectively).
- Clarity: The paper thoroughly introduces the operator theory backgrounds necessary to understand the theoretical results.

Weaknesses:
- Clarity: The readability of the paper could be improved, especially in its description of the kernel learning algorithm in Sections 4 and 5. Currently it requires that the reader is already familiar with Kostic et al. (2022), Kostic et al. (2024), and Turri et al. (2023).
- For other weaknesses, please see the previous sections.

Kostic et al. Learning Dynamical Systems via Koopman Operator Regression in Reproducing Kernel Hilbert Spaces (2022)

Kostic et al. Learning the Infinitesimal Generator of Stochastic Diffusion Processes (2024)

Turri et al. A Randomized Algorithm to Solve Reduced Rank Operator Regression (2023)

**Questions For Authors:**

- In page 5, the authors assume stationarity $X_0\sim\pi$ to simplify the analysis. How restrictive is this, and what can be done if we were to remove this assumption?
- In page 1, the authors mention that understanding long-term behavior is essential for accurate forecasting and interpretation, and in page 4 that learning the IG alone is insufficient for forecasting the process and estimating the spectral decomposition is of greater interest. Indeed, the experiments focus on learning the eigenstructure. Can the authors comment on what is additionally needed to use the proposed method for forecasting?
- In Appendix C.2.4., I am not sure if I understood the first paragraph correctly. Am I right in understanding that the control of the integration term is a novel technical contribution of this paper, which was not established or considered in the previous TO and IG learning methods?
- Out of curiosity, can the authors comment on how the proposed approach is related to the use of Laplace transforms or similar integral operators in Mohr & Mezic, (2014) and Bevanda et al., (2023)?

Mohr & Mezic, Construction of Eigenfunctions for Scalar-Type Operators via Laplace Averages with Connections to the Koopman Operator (2014)

Bevanda et al. Koopman Kernel Regression (2023)

**Relation To Broader Scientific Literature:**

Resolvent operators of Markov semigroups and their expression as Laplace transform has been studied theoretically (Engel & Nagel, 1999) but has not been leveraged in machine learning context as far as I know. The closest ideas I am aware of are identifying Koopman eigenfunctions directly using Laplace transform or similar integral operators (Mohr & Mezic, 2014; Bevanda et al., 2023), which I think are related but not exactly the same to the approach in this work.

Engel & Nagel, One-Parameter Semigroups for Linear Evolution Equations (1999)

Mohr & Mezic, Construction of Eigenfunctions for Scalar-Type Operators via Laplace Averages with Connections to the Koopman Operator (2014)

Bevanda et al. Koopman Kernel Regression (2023)

**Theoretical Claims:**

I carefully checked the soundness of the problem setup, including Markov semigroups, transfer operator and their infinitesimal generator, the resolvent operator, and their spectral decomposition including the compatibility of eigenfunctions and they are sound. I did go over the derivation of the bound of the integration bias but could not carefully verify the correctness due to my limited expertise in spectral perturbation theory. I did not carefully check the proofs for the other bounds.

---

> ### Author Rebuttal · Authors · 2025-04-01
>
> We appreciate the reviewer’s insightful evaluation and valuable comments. Below, due to space limitations, we briefly address the highlighted weaknesses and respond to the reviewer’s questions, committing to incorporating all feedback in our revision. If needed, we can elaborate further on each point.
>
> ## Additional empirical evidence:
> Following the reviewer's suggestion, we have expanded our empirical evaluation to include IG baselines and support the claims summarized in Table 1. For a brief discussion on the [results](https://green-chantal-92.tiiny.site) of our additional experiments, please see our reply to reviewer __nvbC__ .
>
> ## Algorithms
> We will expand Appendix B with detailed derivations, including a discussion of the sampling operator $\hat{S}$. Briefly, by the definition of $\hat{S}$ for $h \in \mathcal{H}$, we derive $\hat{S}^* \hat{S} h = n^{-1/2} \hat{S}^* [h(x_1) \dots h(x_n)]^{\top} = n^{-1} \sum_{i \in [n]} \phi(x_i) h(x_i) = [n^{-1} \sum_{i \in [n]} \phi(x_i) \otimes \phi(x_i)] h$, concluding that the empirical covariance can be written as $\hat{S}^* \hat{S}$.
>
> ## Transient dynamics
> As discussed in Appendix A.3, when $L^* L \neq LL^*$, eigenvalues alone may not fully explain the evolution of linear dynamics. The norm of the resolvent relates to transient growth in stable systems via the pseudo-spectrum (Trefethen & Embree, 2020). We will expand this discussion in the revision.
>
> ## TO as $\Delta t \to 0$
> The core idea is that $A_{\Delta t} = e^{\Delta t L }\to I$, which implies that the spectral gap vanishes, thereby rendering the eigenfunction bounds vacuous.
>
> ## Minor issues
> We will correct all typos. Additionally, we remark:
> The presence of $G^*$ in the risk can be understood from two perspectives. One follows from TO’s risk formulation in the space of observables, which, due to the kernel trick, translates into vector-valued regression (Kostic et al., 2022). The other, as the reviewer notes, considers the Perron-Frobenius operator, (Klus et al., 2019), on probability distributions, linking the estimator to the MMD metric induced by a characteristic kernel. We will expand Section A.2 to introduce regression-based operator learning for both TO and IG methods.
> $\rho_t$ denotes the distribution of $(X_s, X_{s+t})$ and $\mathrm{gap}_i$ is introduced in the last line of Theorem 6.3. A table of major notations will be added at the beginning of the appendix.
>
> ## Questions
> [Q1] When $X_0$’s distribution is not invariant, variance analysis requires adapting the method of blocks for mixing with Bernstein inequalities in Hilbert spaces for independent (but non-identically distributed) variables. This complicates the effective dimension, as the covariance operator w.r.t. the invariant distribution is replaced by one w.r.t. the ergodic mean. While technically feasible, this adds complexity, so we opted for a simpler approach to highlight key contributions.
>
> [Q2]  Knowing IG implies knowledge of the SDE, requiring a numerical solution for process realizations. In contrast, given IG’s spectral decomposition, Equation (8) provides directly the solution, since it enables forecasting of full state distributions beyond just the mean (e.g., $f$ as an indicator function). See reply to __CrN8__.
>
> [Q3] To our knowledge, our technique for controlling Bochner integral approximation errors for linear operators is novel, possibly extending beyond TO/IG methods. We will emphasize this in the contributions paragraph.
>
> [Q4] As the reviewer rightly points out, the Laplace transform is a well-known analytical tool in the study of continuous operator semigroups. In the context of deterministic dynamical systems, it was used by Mohr & Mezic, (2014) to investigate the spectral decomposition of the Koopman operator (TO for deterministic dynamical systems). Their work underscores its potential but leaves efficient numerical methods as an open problem. In contrast, we consider SDEs and not only design numerical methods based on the Laplace transform, but also develop statistical learning theory, providing sharp bounds on spectral estimation. Additionally, we remark that the results of (Bevanda et al. 2023) are also limited to deterministic dynamical systems. There, the Laplace transform served more as inspiration than an explicit component of the method, as the authors built an RKHS via finite-horizon integration of kernel features over trajectories to formulate Koopman operator regression and study learning bounds.

---

> > ### Comment · Reviewer_PJzG · 2025-04-03
> >
> > Thank you for providing the detailed clarifications. I think this is a solid work, the main reason that keeps me from raising my score to 5 is related to the concern of reviewer iVn3--a good amount of additional effort would be necessary for the ideas in this paper to (practically) impact the general audience of ICML.

---

> > > ### Author Response · Authors · 2025-04-03
> > >
> > > We thank the reviewer for their feedback and appreciation of our work. We would just like to note that **we have additionally addressed concerns on the impact**, which, due to limited space in the rebuttal, we couldn’t emphasize enough in our reply to __iVn3__ while still answering all their questions on the mathematical framework we have studied.  We believe that this discussion, which will be included in the revised manuscript, shows how that our work can significantly impact the ML community and inspire more practically oriented follow-up interdisciplinary research.

---

### Decision · Program_Chairs · 2025-05-01

**Decision:**

Accept (poster)

**Comment:**

**Summary.**

The topic of this work is estimation of the Infinitesimal Generator (IG) of a continuous-time Markov processes in $\mathbb{R}^d$ from a trajectory of observations.

Specifically, the authors introduce a new learning algorithm for the problem (with theoretical guarantees), which estimates a spectral decomposition of the (typically unbounded) IG under additional assumptions (e.g. geometrically ergodicity, sectoriality).
The contribution contrasts with existing methods which focused on estimating transfer operators instead.
The authors demonstrate the applicability and robustness of their approach through experiments.

**Strengths.**

* The reviewers asked for numerous technical clarifications in their reviews, but the response was very effective and no soundness concerns were reported.
* Strong theoretical analysis (nvbC, iVn3, PJzG).
* The method overcomes a key drawback of previous methods: it remains stable even for short time-lags.
* The central idea of focusing on the resolvent of the IG was considered to be novel and insightful.
* The proposed approach is applicable to a broad class of Markov processes (including but not limited to self-adjoint IG) and to high-dimensional spaces (an experiment was conducted with d=45).

**Weaknesses.**

* There were mixed opinions about clarity: the paper is neatly written (CrN8) but is too dense (iVn3); its readability could be improved (PJzG, nvbC).
* Experiments fail to demonstrate superiority of the method (nvbC, CrN8, iVn3).
* Impact on the general audience of ICML is questionable (PJzG, iVn3).

**Discussion and reviewer consensus.**

The reviewers unanimously agree that this paper should be accepted. No major concern remains after the author response. The author response to the only remaining concern about the broader impact of the results (PJzG, iVn3) was sufficient to convince me.

**Overall evaluation.**

This is serious work; I recommend acceptance. In my opinion, the theoretical contribution warrants publication, and the authors have done an honest effort to back-up their claims experimentally; remaining experimental shortcomings should not disqualify the paper.

**Recommendation to the authors.**

If the paper is accepted, I recommend the authors to make good use of the additional space to increase the clarity of their writing; additional experiments may be added in the supplementary material (refer also to the suggestions of nvbC regarding adding a table of notations).